# The mechanism of DNA unwinding by the eukaryotic replicative helicase

Daniel R. Burnham [1], Hazal B. Kose[1], Rebecca B. Hoyle [2] & Hasan Yardimci [1]

Accurate DNA replication is tightly regulated in eukaryotes to ensure genome stability during cell division and is performed by the multi-protein replisome. At the core an AAA+ hetero-hexameric complex, Mcm2-7, together with GINS and Cdc45 form the active replicative helicase Cdc45/Mcm2-7/GINS (CMG). It is not clear how this replicative ring helicase translocates on, and unwinds, DNA. We measure real-time dynamics of purified recombinant *Drosophila melanogaster* CMG unwinding DNA with single-molecule magnetic tweezers. Our data demonstrates that CMG exhibits a biased random walk, not the expected unidirectional motion. Through building a kinetic model we find CMG may enter up to three paused states rather than unwinding, and should these be prevented, in vivo fork rates would be recovered in vitro. We propose a mechanism in which CMG couples ATP hydrolysis to unwinding by acting as a lazy Brownian ratchet, thus providing quantitative understanding of the central process in eukaryotic DNA replication.

[1] The Francis Crick Institute, 1 Midland Road, London NW1 1AT, UK. [2] School of Mathematical Sciences, University of Southampton, Southampton SO17 1BJ, UK. Correspondence and requests for materials should be addressed to H.Y. (email: Hasan.Yardimci@crick.ac.uk)

Healthy cell division requires accurate DNA duplication to maintain genome stability which is accomplished through complex coordination of proteins forming a replisome, travelling along and replicating parental DNA. To provide templates for synthesis, DNA strands are first separated by NTP hydrolysing helicase proteins, invariably forming a toroidal complex[1], a mechanism common across all domains of life. Eukaryotic replisome construction and disassembly is tightly regulated[2,3]; an AAA+ ATPase hetero-hexameric minichromosome maintenance complex (Mcm) 2–7 loads as a double hexamer on double-stranded DNA (dsDNA)[4] at origins of replication before additional firing factors aid formation of the active replicative helicase Cdc45/Mcm2-7/GINS (CMG)[2,5–7]. Further proteins are recruited to the replication fork, including polymerases, Tof1, Csm3, Mrc1, and PCNA, all required to achieve maximum rates of replisome progression[8]. A concerted effort is underway to understand the molecular mechanisms of origin firing, from Mcm2-7 recruitment, through CMG activation, to elongation[3,7,9,10].

CMG, thought to be conserved across archaea and eukaryotes[11], is a ring-shaped helicase formed by the superfamily six Mcm2-7 hexamer with distinct C- and N-terminal domains forming two tiers[12], and GINS and Cdc45 plugging a gate between Mcm2 and Mcm5[12,13], preventing the opening of the ring without additional factors[14]. Notably, other replicative helicases, including SV40 large T antigen, can load from a monomeric state and open the hexameric ring dynamically[15,16]. Translocation proceeds on single-stranded DNA (ssDNA) with 3′-5′ polarity, unwinding via steric exclusion[5,7,17,18]; excluding the lagging-strand template outside, and threading the leading-strand template through the helicase[13,19].

How CMG couples ATP hydrolysis to unwinding is unclear and an understanding of the translocation, hence unwinding mechanism, post activation, remains elusive, although analogous helicases give indications. Structural studies of papillomavirus DNA replicative ring helicase E1 suggest a cyclic escort translocation mechanism, where the helicase walks along ssDNA, through hand-over-hand movement of DNA-binding loops[20], possibly slipping as it does so[21]. The homohexameric bacterial replicative helicase DnaB is suggested to work similarly but the C-terminal domain of each monomer disengages to perform the hand-over-hand translocation[22]. For bacteriophage T7 replicative helicase, gp4, steps of various size have been correlated with NTP hydrolysis[23], and viral monomeric helicase NS3 was described as a Brownian motor[24]. Structural studies of CMG reveal only two distinct conformers, not the expected six[13] leading to proposals of more exotic motions, including a pumpjack[25] or spring-like coil, acting as an inch-worm[10].

Little is known regarding the dynamics or kinetics of CMG as it unwinds DNA[9,14]. Our work measures single CMGs unwinding dsDNA in real-time with high precision. We find linear unwinding rates low compared to replication fork rates measured in cells and the helicase exhibits unexpectedly complex dynamics. To explain our data we build a quantitative kinetic model to underpin a biophysical description of the unwinding mechanism and establish that CMG unwinds DNA via a biased random walk with propensity to pause. Without this understanding of the foundation of the eukaryotic replisome, further insight into how replication dysfunction can be a major source of genome instability will be slowed.

## Results

### Tracking DNA unwinding with magnetic tweezers. Helicase activity of purified recombinant *Drosophila melanogaster* (*Dm*CMG) (Fig. 1a) was confirmed with bulk model fork unwinding assays[18,26].

The substrate was incubated with *Dm*CMG in the presence of ATPγS and unwinding initiated by addition of ATP. *Dm*CMG demonstrated helicase activity (Fig. 1b), matching previous work[6].

We remove ensemble averaging of classical biochemical techniques by employing multiplexed single-molecule magnetic tweezers[27] (Fig. 1c, Supplementary Fig. 1a). A single 2.7 kilobase (kb) dsDNA molecule, with a polyT 3′ flap for high affinity binding[6,26], is tethered between a PEGylated glass surface via biotin/streptavidin binding and a super-paramagnetic microsphere via digoxigenin/anti-digoxigenin binding. CMG is drawn into the sample chamber in loading buffer, containing ATPγS, and given time to bind the 3′ flap. Next ~4–5 chamber volumes of temperature-equilibrated CMG-free running buffer, containing ATP, is drawn through the chamber (Fig. 1d). This ensures only previously loaded CMG remains in the sample chamber, preventing multiple helicases acting on single DNA substrates. Constant force is applied with a cuboid magnet pair such that conversion from dsDNA to ssDNA increases the microsphere vertical displacement (Supplementary Fig. 1b). After subtraction of the position of reference microspheres stuck to the glass coverslip[28] and low-pass filtering to 0.17 Hz, displacement is measured with $3.2 \pm 1.2$ nm ($n = 3$, all uncertainties standard deviation, unless otherwise stated) standard deviation over 91 mins (Supplementary Fig. 2). This displacement is converted to base pairs unwound; a proxy for helicase position. Using forces between 20 and 40 pN prevent re-annealing behind the helicase and permits spontaneous re-annealing from the ss-dsDNA junction, ahead of the helicase (Supplementary Note 1). Measuring this displacement through time gives a CMG unwinding trajectory. Upon analysis, ~10% of single DNA molecules show activity, demonstrating it is unlikely more than a single helicase acts (further detail in Supplementary Note 2).

The instrument precision is quantified by the Allan deviation[29] of an enzyme-free trajectory (Supplementary Fig. 2). Before low-pass filtering, the Allan deviation shows precision on the order of a nanometre across experimental timescales. The method and apparatus are validated by demonstrating that unwinding by homohexameric ring helicase SV40 large T antigen replicates the speed and features of previous work[30,31] and has features similar to other ring helicases[32–34] (Supplementary Fig. 3). The protocol remains identical to CMG assays except 110 nM monomer large T antigen was incubated with ATP at 37 °C for 20 mins before introduction to the sample chamber of tethered DNA, with no further chamber washes. In Supplementary Fig. 3a, we demonstrate a typical example of SV40 large T antigen unwinding a 1-kb DNA hairpin and in Supplementary Fig. 3b, unwinding the 3′ flap DNA template used for CMG experiments. The hairpin was not used in CMG experiments as in our hands no unwinding was observed, likely as CMG requires a free 3′ ssDNA tail to bind the DNA construct[14] (Supplementary Fig. 4 and Supplementary Note 3).

The measured trajectories are low-pass filtered to 0.17 Hz (Supplementary Fig. 5a). Following the protocol outlined in Fig. 1d, we observe typical CMG single-molecule unwinding trajectories as shown in Fig. 1e, with no unwinding observed in the absence of CMG or ATP (Supplementary Figs. 6a and 7).

### CMG slowly unwinds DNA, non-monotonically with heterogeneity. Figures 1e, f demonstrate the number of base pairs unwound by CMG is non-uniform in rate and does not solely increase. We observed variable unwinding rates and regions where the number of unwound base pairs decreases. This indicates unwinding of DNA by CMG is non-monotonic with heterogeneity in rate found both within, and between, individual enzyme unwinding trajectories.

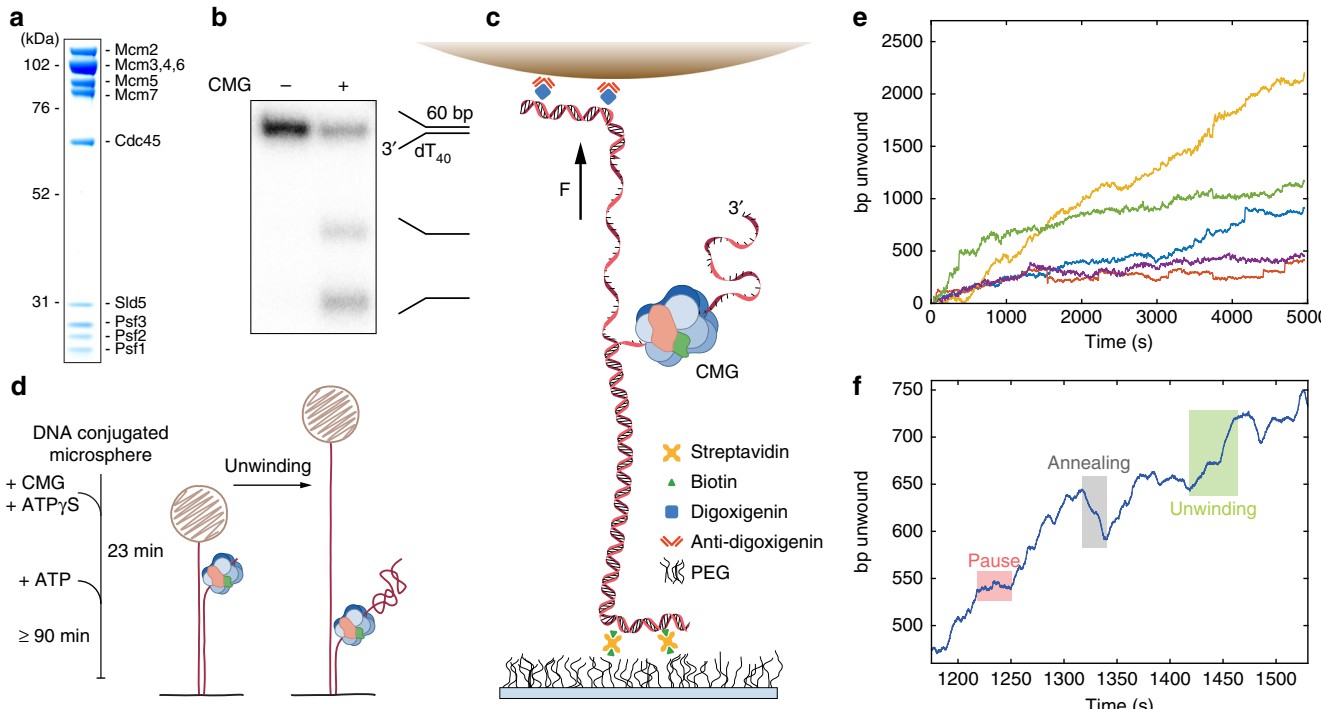

**Fig. 1** Purified recombinant *Dm*CMG, bulk unwinding, single-molecule assay principle, and observed single-molecule DNA unwinding trajectories. **a** Coomassie stained 4–12% SDS-PAGE gel of purified *Dm*CMG. Sld5, Psf1-3 forms GINS. **b** Example of bulk unwinding of duplex DNA, radiolabelled at 5′ ends, by CMG. The 60 bp duplex with 40 nt 3′ polyT tail is unwound into the two single strands. **c** To form the single-molecule assay a single 2.7 kilobase dsDNA molecule, with a polyT 3′ overhang to serve as a high affinity binding site for CMG, is tethered at one end to a PEGylated glass surface via biotin and streptavidin binding; and to a super-paramagnetic microsphere at the other end via digoxigenin/anti-digoxigenin binding. A force, *F*, is applied using a pair of neodymium cube magnets. Precise tracking of the microsphere vertical displacement gives the extension of each single DNA molecule. **d** Protocol for experiment. Surface-tethered DNA is incubated with CMG in the presence of ATPγS to aid helicase loading onto the 3′ ssDNA tail. ATP is then introduced through a 4–5 sample chamber volume wash, and unwinding begins, with data being collected for at least 90 min. **e** Typical examples of single-molecule DNA unwinding by CMG at 20pN and 4 mM ATP, filtered to 0.17 Hz using a mean running window. CMG is a slow helicase, acting non-monotonically with heterogeneity both within and between individual enzyme unwinding events. **f** Single-molecule unwinding CMG trajectories exhibit not only unwinding but also, what can be crudely described as, pause-like dynamics and annealing, which we attribute to reverse motion of the helicase. Example regions have been manually highlighted

We observed CMG unwinds DNA slowly compared to previous ring helicase studies[35]. Calculating the mean unwinding rates with linear fits to single-molecule unwinding trajectories, we measured rates between $0.10 \pm 0.08$ bps$^{-1}$ ($n = 19$) and $0.47 \pm 0.56$ bps$^{-1}$ ($n = 75$) over the ATP concentrations and forces applied in our experiments (Fig. 2). Although one should be careful not to heavily weight inferences from this naive approach, this is approximately one to two orders of magnitude slower than the ~10–50 bps$^{-1}$ replication fork rates observed in eukaryotic cells[36–38]. The replicative helicase does not attain the speeds necessary for timely genome duplication.

Our observations contrast starkly with the 5′ to 3′ translocating superfamily 4, RecA core, ring replicative helicases of bacteriophage[39–41] and *E. coli*[32]. These have predominantly exhibited monotonic, uniform, and unidirectional unwinding behaviour at ~100 bps$^{-1}$, interspersed with pauses in activity[32]. These rates are within one order of magnitude of replication fork rates observed in live cells and bulk biochemical assays[38,42–45]. CMG linear unwinding rates measured in this work are similar to those of other AAA+ helicases. For example, purified SV40 large T antigen shows relatively slow unwinding rates of ~1.5 bps$^{-1}$ (online refs. [30,31]) and ~3 bps$^{-1}$ in cell extract in vitro[15]. Furthermore, at high precision E1 shows non-monotonic behaviour, including pausing, forward, and reverse movement[21] as does archaeal MCM[33]. Similarities in speed and dynamics between these homohexameric helicases and CMG may result

from the common AAA+ ATPase motor and 3′ to 5′ translocation direction of superfamily three and six helicases.

Unexpectedly, CMG exhibited not only unwinding and pause-like dynamics but also reverse motion (Fig. 1f). With annealing behind the helicase prevented at high forces spontaneous annealing can only take place from the ss-dsDNA junction. Thus, the reduction in bp unwound indicates CMG has travelled backwards, allowing annealing ahead of the helicase. Reverse motion previously observed in ring helicases is either abrupt and monotonic and ascribed to slipping[21,33,46], a motion considered not to require ATP hydrolysis, or is on the order of single base pairs[23]. Our results showed CMG translocates backwards over prolonged periods, neither abruptly nor over single base pairs, thus cannot be slipping. In this work, rather than attribution to separate mechanisms, we incorporate the reverse motion within a single unifying description, thus the emergence becomes inherent (see below).

We investigated DNA unwinding by individual CMG molecules with varying forces applied to the linear duplex through the lagging-strand template. Passive helicases, which await spontaneous thermal fluctuations to open the duplex, demonstrate higher unwinding rates when larger forces are applied to DNA[47]. Naively calculating the linear unwinding rate (Fig. 2a) we observed a wider distribution and increase in mean from 30 pN to 40 pN, yet a decrease in mean and narrower distribution from 20 pN to 30 pN ($t(86) = 5.1$, $p = 2.2 \times 10^{-6}$ and $t(78) = 2.9$,

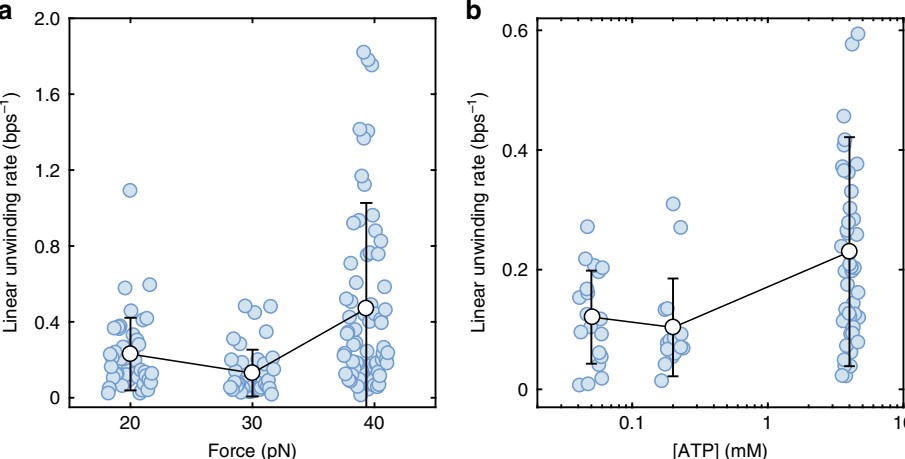

**Fig. 2** Mean linear DNA unwinding rate by CMG as a function of force and ATP concentration. **a** Forces of 20.0, 30.0, and 39.3 pN applied to the linear duplex through the lagging-strand template varies the unwinding rates and distribution with increases at high forces. **b** ATP concentrations of 0.05, 0.2, and 4 mM varies the unwinding rates and distribution with increasing ATP having higher rates. Error bars are standard deviation

$p = 0.004$, respectively. Welch's $t$-test statistics given in Supplementary Fig. 8). Despite the small decreases in mean between 20 pN and 30 pN, these results are broadly consistent with the trends observed for other hexameric helicases[32,41,47] and the expected force dependence from theory[47,48]. The linear unwinding rate is also affected by ATP concentration (Fig. 2b) with decreasing, sub-saturating[6,26] conditions leading to lower mean unwinding rates, but with similar distributions.

The processivity, number of base pairs unwound before activity ceases, is $827 \pm 642$ bp ($n = 197$). This lower limit arises as our experiments do not reach an end state so processivity is unwinding rate dependent (Supplementary Fig. 6b, c). Thus, the maximum number of bases unwound in each trajectory is used to report mean and standard deviation for all data.

**CMG has behaviour beyond that of a simple walker.** Activity and pausing are commonly discriminated between in single-molecule molecular motor trajectories, using instantaneous velocity thresholding techniques[49,50], to remove pauses[32] or study the kinetics of the pauses and unwinding bursts[49,51]. This technique is susceptible to bias from choice of derivative window size and mis-assignment of states due to a threshold determining pausing versus activity[52]. To characterise behaviour, this state-separated data is binned, a notoriously subjective process, before usually fit with a Gaussian model a priori. The low instantaneous velocity of CMG is indistinguishable from noise usually considered to be a pause and prevents standard techniques detecting states within the raw trajectory. Instead we employ first-passage time (FPT) analysis[53], extended from dwell time distributions[52,54–56], which also negates the analysis problems described. We measured the time taken for CMG to first unwind a specific number of base pairs (passage interval, Fig. 3a, 20 bp), and repeated along the trajectory. To prevent spurious detection of FPTs we choose the passage interval to be twice the standard deviation of an enzyme-free trajectory (Supplementary Fig. 5)[52]. Fast continuous unwinding appears as short FPTs, but annealing increases the times because the helicase must repeatedly unwind the same segment of DNA (Fig. 3a).

The distribution of measured FPTs informs on the behaviour of the helicase[52]. Taking the commonly held model of a unidirectional deterministic walker[20,57], the FPT distribution would be a Dirac δ function. Including the stochastic nature of biochemical processes, the distribution of FPTs would take the

form of a gamma distribution (Fig. 3b, green line), parameterised by the number of intermediate kinetic steps in the translocation process and corresponding rate[55]. Taking our experimental FPTs, we plot a histogram to recover experimental FPT distributions (Fig. 3b, blue circles, and Fig. 3c, blue squares and red circles). Undoubtedly, there is a substantial difference between the experimental data and expected gamma distributions.

The experimental FPT distribution for CMG unwinding DNA has distinct features. Firstly, the FPTs span several orders of magnitude in time and probability density. Secondly, we observed a broad peak at short timescales, indicative of kinetic processes with multiple intermediate steps. Finally, exponential decays at longer times, indicative of no intermediate steps. This is vastly different to the distribution of times expected from linear unidirectional mechanisms, such as coordinated escort[20], pump-jacks[25] or inch-worms[10]. The experimental FPT distribution shows CMG is unlikely to, deterministically, escort or pumpjack DNA.

**CMG is a biased random walker with multiple distinct pauses.** Current suggested mechanisms are unable to explain the observed FPT distributions and reverse motion. With absence of evidence indicating CMG acts unidirectionally (Supplementary Fig. 6d), we describe the observed motion with a one-dimensional hopping model discrete in space and continuous in time[58], as developed for helicases by Betterton and Jülicher[59]. Pictured in Fig. 3d, the helicase translocates on ssDNA, positioned at the junction between ss- and dsDNA, by hopping forwards at rate $r_f$ from $n$ to $n + 1$, and backwards at rate $r_b$. The resulting biased random walk is governed by the master equation,

$$\frac{\partial P(n,t|n_0)}{\partial t} = r_b P(n+1,t|n_0) + r_f P(n-1,t|n_0) - (r_f + r_b)P(n,t|n_0),$$

(1)

where $P(n,t|n_0)$ is the probability of the helicase being at base pair $n$ at time $t$, given it started at $n_0$ at $t = 0$.

The analytical solution for the FPT distribution of the biased random walk of Eq. 1 is given by the first term of Eq. 2[60] which describes a broader peak (Fig. 3b, orange line) in comparison to a stochastic unidirectional walker (Fig. 3b, green line). This arises from the possibility of backwards travel and thus increase in duration of FPTs. However, this description does not explain the appearance of long FPTs observed in our experimental data

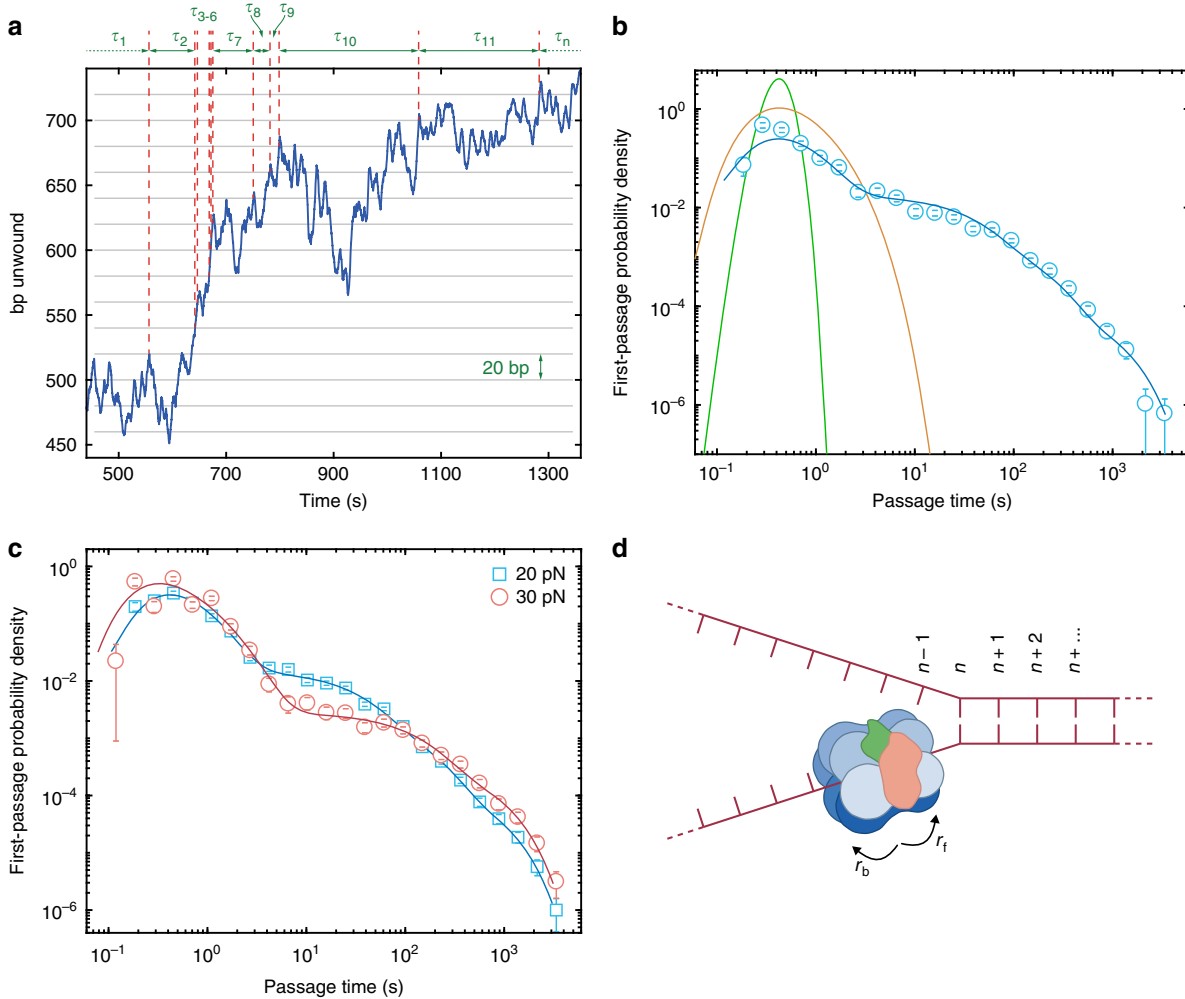

**Fig. 3** Extracting dynamic and kinetic information from CMG unwinding trajectories. **a** Portion of an example unwinding trajectory. Complex motion is observed but assigning any portion to a given state of unwinding, annealing, or pausing, leads to miscounting and is biased. Instead we measure the time taken, $\tau$, for the helicase to first unwind a set interval of basepairs (grey horizontal lines) along the whole trajectory, to give a list of first-passage times. $\tau_{3-6}$, for example, are short due to the fast unwinding present, $\tau_{10}$ is long due to re-unwinding of the same DNA and $\tau_{11}$ is long due to a crudely described pause. **b** Experimental first-passage time distribution for 4 mM ATP and 20 pN force (blue circles); bins are normalised by bin width and total counts. Blue line is maximum likelihood estimation for the model of Eq 2. Note, the data is not directly fit to the experimental histogram, MLE estimates the parameters best suited to describe the distribution of all first-passage times. Data from 21 molecules and 1009 first-passage times. Green line is the analytical first-passage time distribution solution for a unidirectional mechanism that includes the stochastic nature of biochemical processes, a gamma function, parameterised by the number of steps to cross the passage interval and the rate of unwinding. A biased random walk provides a broader peak, calculated with the same modal FPT, and is parameterised by the rate of forward and backwards hopping (orange line, first term Eq. 2). **c** Comparison of two FPT distributions at different forces and constant 4 mM ATP concentration. Blue squares are at 20 pN with a resulting model fit (blue line) which includes 3 pause states. Red circles are 30 pN with a resulting model fit (red line) which includes 2 pause states. Data is as reported in Supplementary Table 1. **d** The helicase is treated as translocating along ssDNA at the ss-dsDNA fork junction by hopping discretely in space, but continuously in time, forward or backwards one base pair at a time with rates $r_f$ and $r_b$ respectively. All error bars are standard deviation from 1000 bootstraps

(Fig. 3b). Given the exponential nature of these times and the propensity of translocating enzymes to pause[32,41,47,49,50,52,55], we attribute these events to one or more pause like states, and include them in the FPT probability distribution as the second term in Eq. 2[28].

Combining the biased random walk unwinding state and probable pause states, we describe the FPT distribution, $P(\tau)$, for CMG unwinding DNA by;

$$P(\tau) = w_{rw} \frac{m}{\tau} \left(\frac{r_f}{r_b}\right)^{\frac{m}{2}} e^{-(r_f + r_b)\tau} I_m \left[(r_f + r_b)\tau \left(1 - \left(\frac{r_f - r_b}{r_f + r_b}\right)^2\right)^{\frac{1}{2}}\right] + \sum_{i=1}^{q} w_i k_i e^{-k_i \tau}$$

(2)

where $m$ is the number of bases in the passage interval, $I_m$ is the modified Bessel function of the $m^{th}$ kind, $\tau$ is the FPT, $w_{rw}$ is the proportion of FPTs attributed to the biased random walk, $w_i$ describes the proportion of FPTs that are accounted for by pause states, and $k_i$ are the pause exit rates. The number of distinct pauses, $q$, is selected through comparison of the Bayesian information criterion (BIC) for increasing numbers of pauses. The mean unwinding velocity is $v_{mean} = (r_f - r_b)$ and the effective diffusion coefficient $D_{eff} = \frac{1}{2}(r_f - r_b)$[58].

Evidence from cryo-EM studies of both *Saccharomyces cerevisiae* (*Sc*CMG) and *Dm*CMG indicate all six Mcms may bind ssDNA in the central channel with two nucleotides per monomer[13,19]. This does not preclude the possibility of single

nucleotide hopping as used here. Furthermore, our model description can be adapted for arbitrary integer nucleotide hopping without qualitative alteration of the results and interpretation.

We observed up to 3 distinct pauses, $q \leq 3$, so for clarity we label the proportion of FPTs attributed to pauses at short, medium and long timescales as $w_{short}$, $w_{med}$ and $w_{long}$, replacing $w_i$, respectively. The corresponding exit rates are $k_{short}$, $k_{med}$ and $k_{long}$. Following Dulin et al.[28] we report probabilities of entering the biased random walk, $P_{rw}$, or short, medium, and long timescale pauses, $P_{short}$, $P_{med}$, and $P_{long}$, respectively.

The parameters of the model best describing our measured FPTs are determined via maximum likelihood estimation. For the data in Fig. 3b, we overlay the resulting FPT distribution model (blue line) on the experimental FPT distribution of multiple molecules from a single experiment (data combined from multiple experiments is presented in Supplementary Table 1). In the typical example of Fig. 3b we found $r_f = 136 \pm 22 \, bps^{-1}$ and $r_b = 109 \pm 27 \, bps^{-1}$, giving $v_{mean} = 27 \pm 35 \, bps^{-1}$, and $D_{eff} = 122 \pm 17 \, bps^{-1}$ at 20 pN and 4 mM ATP. For this example data set the proportion of FPTs contributed by biased random walk unwinding is $w_{rw} = 0.26 \pm 0.05$, while $w_{short} = 0.15 \pm 0.12$, $w_{med} = 0.45 \pm 0.10$ and $w_{long} = 0.14 \pm 0.06$. The probability of entering the biased random walk state, $P_{rw}$, is $0.935 \pm 0.002$, and short, $P_{short}$, medium, $P_{med}$, and long, $P_{long}$, long, timescale pauses, $0.02 \pm 0.01$, $0.04 \pm 0.02$, and $0.007 \pm 0.010$, respectively. Finally, the exit rates from the short, $k_{short}$, medium, $k_{med}$, and long, $k_{long}$ pauses are $0.034 \pm 0.013 \, s^{-1}$, $0.0078 \pm 0.0030 \, s^{-1}$, and $0.0014 \pm 0.0004 \, s^{-1}$, respectively. Full statistics and number of replicates are given in Supplementary Table 1. In Fig. 3c we plot further examples of FPT distributions at two forces demonstrating changes in the distribution at different timescales. At 20 pN, an increase in propensity to pause at short timescales correlates with a reduction in the FPT probability density of performing the biased random walk (the short time peak), when compared to 30 pN.

Remarkably, we can describe the dynamic motion observed in Fig. 1e, f using a relatively simple stochastic model. Elegantly, the periods of reverse translocation are an inherent consequence of the model, without an additional separate mechanism, due to the finite probability of the helicase hopping backwards.

The trajectories used to calculate the linear unwinding rates in Fig. 2 underwent FPT analysis and the resulting parameter estimations plotted in Fig. 4 and Supplementary Fig. 9. The proportion of FPTs ascribed to pausing ($w_{short} + w_{med} + w_{long}$) accounts for between $0.43 \pm 0.05$ and $0.84 \pm 0.04$ of FPTs observed. The proportion of FPTs attributed to CMG performing the biased random walk of unwinding, $w_{rw}$, ranges between $0.16 \pm 0.02$ and $0.57 \pm 0.02$. Thus, surprisingly, the FPT can be $\gtrsim 2$ times more likely to be dominated by a pause state rather than biased random walk unwinding. The probability of entering a random walk state, $P_{rw}$, is $\gtrsim 0.91$, and the likelihood of entering one of the three pause states is $\lesssim 0.09$, as displayed in Fig. 4a, b. The mean rate of exit from the pause states, over all observations, is $0.015 \pm 0.02 \, s^{-1}$, quantitatively demonstrating the helicase frequently enters long lived pauses.

The mean unwinding velocity, $v_{mean}$, may be interpreted as a pause-free unwinding rate, describing the peak in FPT distribution alone, measured between $18 \pm 13 \, bps^{-1}$ and $32 \pm 27 \, bps^{-1}$ for our data (Fig. 4c, d). Considered alone, these compare well with expected eukaryotic replication fork rates of ~10–50 bps$^{-1}$ in cells and the maximal in vitro replisome rates of 24 bps$^{-1}$ (online ref. [8]). However, the largely constant $v_{mean}$ does not explain the change in linear unwinding rates (Fig. 2a) as a function of force. The decrease in probability of entering a short pause (Fig. 4a and Supplementary Fig. 9) and increased rate of

exiting (Fig. 4e) as a function of force is most likely responsible for speed up. Similarly, for increasing ATP concentration the increase in pause exit rates is likely responsible for the increase in linear unwinding rate (Fig. 4f and Supplementary Fig. 9). Our data indicates that increasing force and ATP concentration do not increase the helicase random walk forward bias but do reduce the probability of pause entry and increase rate of exit, changing the average speed of the helicase.

The resulting estimates for the parameters of our model (Fig. 4 and Supplementary Fig. 9), show that CMG undergoes a diffusive random walk with a bias towards forward translocation acting to unwind DNA, with a proclivity to pause on at least two discrete timescales.

**CMG activity is an interplay between unwinding and pausing**. The probability of entering medium and long pause states is constant with force but entrance into the biased random walk or short pause are affected (Fig. 4a and Supplementary Fig. 9a, c, e, g). For decreasing ATP concentration, the long timescale pause disappears, together with a reduction in the likelihood of entering the medium timescale pause. The exit rate from a medium pause also decreases with decreasing ATP concentration (Fig. 4f and Supplementary Fig. 9l, n). The rate of short pause exit, $k_{short}$, increases with ATP concentration, while decreasing with force between 20 pN and 30 pN before increasing for 40 pN (Fig. 4e, f and Supplementary Fig. 9k, l). Force and ATP regulate the entrance and exit from pauses demonstrating the intricate interplay between mechanics and ATP hydrolysis.

## Discussion

We have demonstrated the molecular motor CMG exhibits dynamics of a biased random walk, with a propensity to pause. This more complex, yet elegant, description forgoes the necessity for linear or deterministic mechanisms, such as coordinated escort.

It remains unclear whether this is a universal feature of ring helicases or peculiar to CMG. The relatively slow linear unwinding rates of CMG may allow sufficient sampling of the activity to uncover the behaviour, but for ring helicases acting at 100 s bps$^{-1}$, such as DnaB[32], the activity may be under-sampled. Indeed, it has been suggested that backstepping is a universal feature of helicases[61]; a topic for further study.

The low linear unwinding rate observed is surprising and our work explains the origin of these long timescales. Firstly, the helicase undergoes a biased random walk; the magnitude of the bias governs the velocity of the helicase; 18 to 32 bps$^{-1}$. It is the relatively small bias, due to the finite backward hoping rate, that contributes to low speed. Secondly, the frequent entrance into long-lived pause states further reduces the overall speed.

Collaborative coupling between helicase and polymerase has been observed to speed up forks by 3–30 fold in prokaryotic[62] and bacteriophage[45,63] systems and single-molecule work has shown reduction in pausing[63]. Bulk biochemical studies of the eukaryotic minimal replisome observed rates of 4.4 bps$^{-1}$ in vitro[64] and inclusion of further replisome components increases this to 24 bps$^{-1}$ (online ref. [8]). Single-molecule studies of ScCMG within minimal replisomes have shown replication with polymerase ε occurs at 5.4 bps$^{-1}$, the inclusion of Mcm10 increases this to 11.9 bps$^{-1}$, and Mrc1-Tof1-Csm3 further to 21.1 bps$^{-1}$ (online ref. [9]). Given eukaryotic replication is more tightly regulated than viral and bacterial counterparts, it may not be surprising that eukaryotes need factors beyond a coupled polymerase to achieve endogenous replication fork rates.

We hypothesise additional eukaryotic replisome factors such as polymerase ε, RPA, Mrc1, Csm3, Tof1, or Mcm10, not present in

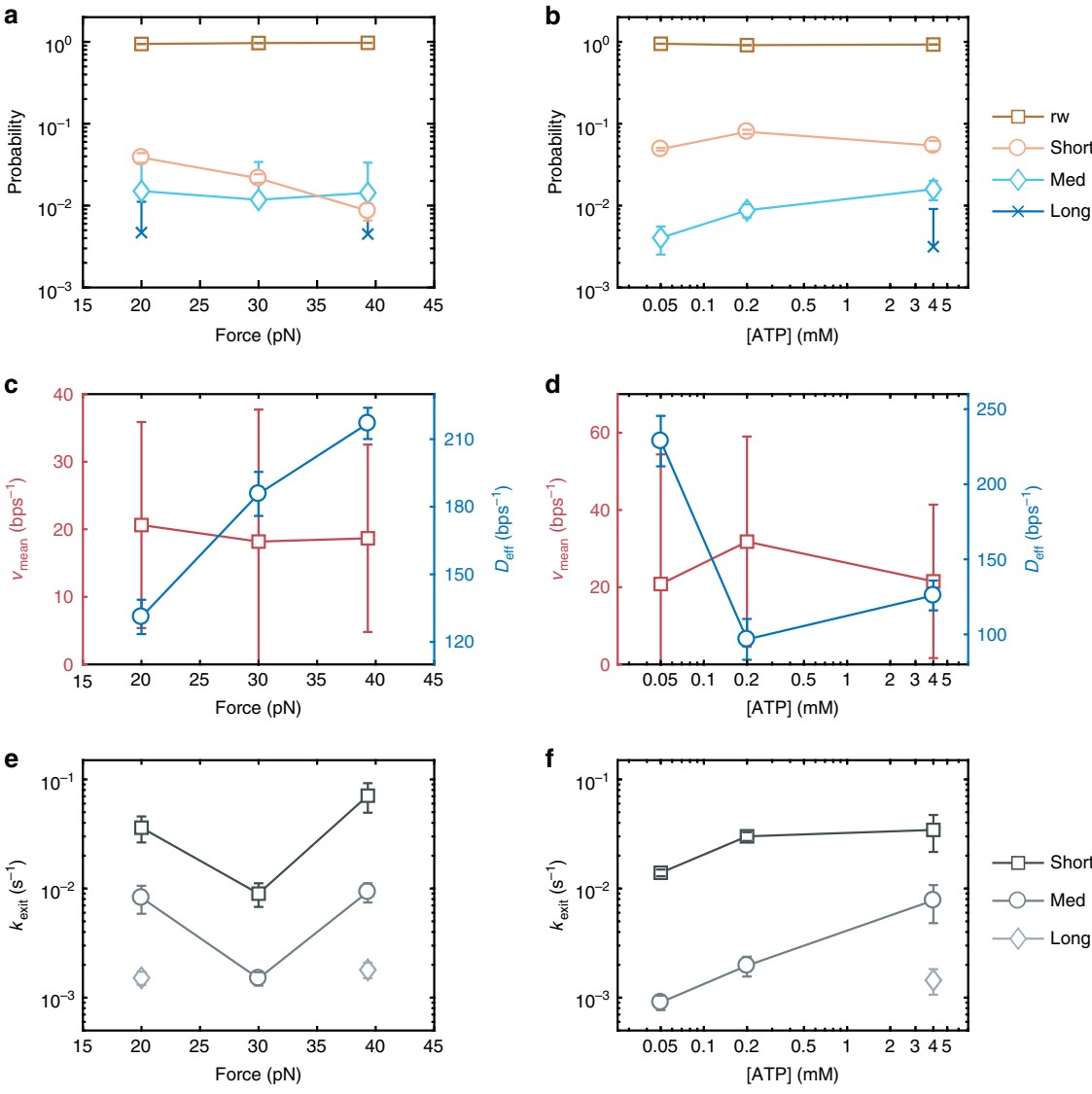

**Fig. 4** Model parameters of Eq. 2 extracted from maximum likelihood estimation of experimentally measured first-passage times for varying force and ATP concentration. Probability of entering the biased random walk state (brown squares), a short (orange circles), medium (cyan diamonds), or a long (blue crosses) timescale pause as a function of **a** force and **b** ATP concentration. Mean velocity (red sqaures) and effective diffusion (blue circles) as a function of **c** force and **d** ATP concentration. Rate of exit from short (dark grey sqaures), medium (grey circles) or long (light grey diamonds) timescale pause as a function of **e** force and **f** ATP concentration. All error bars are standard deviations from 1000 bootstraps

our study, increase the forward bias of the random walk, or reduce the probability of entering, or increase the exit rate from, a pause, or a combination of both. In particular, one would expect leading strand synthesis to severely impair any reverse motion currently observed. All would increase the linear fork rate measured.

Our quantitative kinetic model describing the behaviour of CMG as it translocates along ssDNA as a biased random walker with pausing explains well the heterogenous, non-monotonic and low speed unwinding observed. To better understand the molecular mechanism of CMG, we wished to establish a biophysically sensible description that explains how ATP hydrolysis is coupled to unwinding and gives rise to such dynamics. Our model must be consistent with unwinding occurring in a crowded Brownian environment[65,66]. We exclude the colloquial power-stroke, which describes a movement driven by a conformational change that releases free energy, as it has been shown to be irrelevant in determining the directionality and thermodynamic properties of all chemically driven molecular motors[67].

The two conformational states observed in CMG structures have been suggested to represent the compact and extended shape of an inch-worm like mechanism[25] or an open paused state and closed translocation state[13]. We propose these two conformational states alter the affinity for DNA (and the specificity for ATP) allowing the motor to act as a Brownian ratchet[68–70]. Such translocation mechanisms can be mapped onto the biased random walk master equation (Eq. 1) describing the motion observed here[71], and only two DNA affinity states are required[72].

The open ring configuration may correspond to a relaxed ATPase in a post ATP hydrolysis state with weak affinity for DNA. This is in agreement with the structural conformer showing the absence of DNA bound in the CMG central channel[13] and the poor affinity for DNA fork template in the presence of ADP[26]. When ATPγS is bound, CMG forms a closed ring and allows capture of template in the structural conformer[13] matching the observed high affinity for DNA substrate[26].

To facilitate a Brownian ratchet we propose, in the weakly bound state, open conformer, CMG may diffuse along DNA due

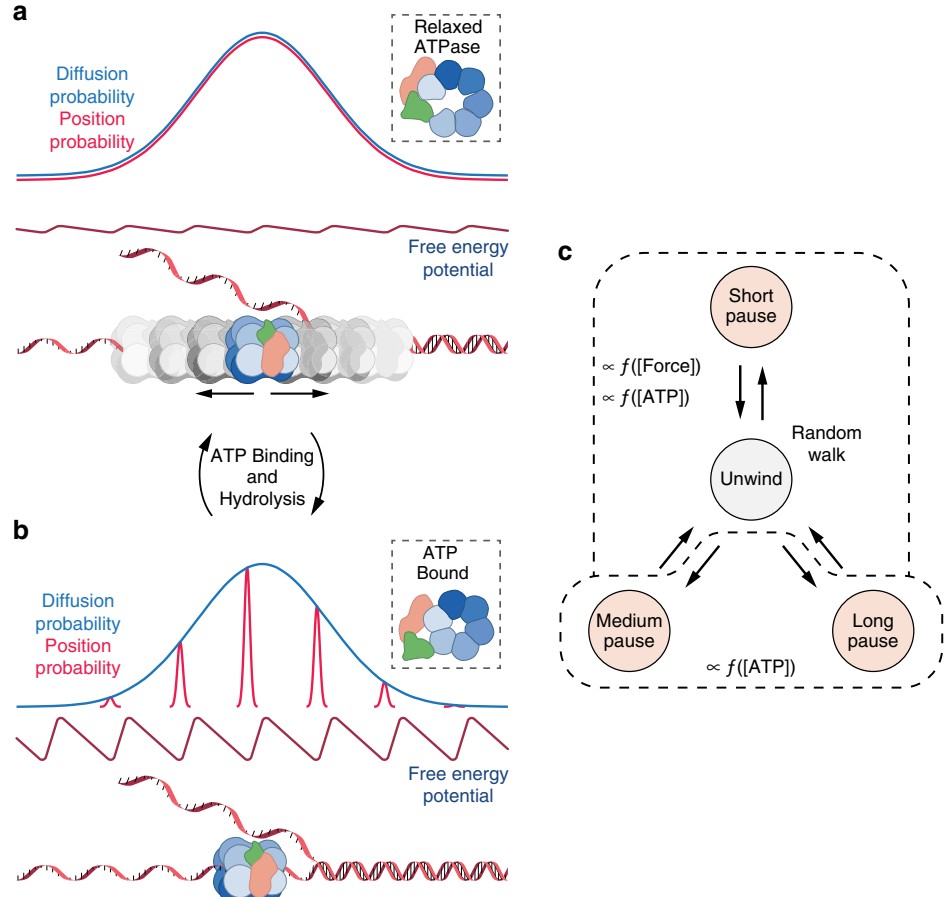

**Fig. 5** Proposed Brownian ratchet mechanism and most likely kinetic scheme. **a** In the weakly bound state, open conformer where the Mcm2,5 gate is open (dashed box). CMG can undergo unbiased diffusion (random walk) along the DNA due to a weak free energy potential. **b** Upon ATP binding/hydrolysis CMG undergoes an allosteric structural change that alters the affinity for DNA, increasing the free energy potential of interaction with the DNA. In this state, CMG remains at the bottom of the potential well, fixed in position. The asymmetry in the potential means that upon cycling between these two states through ATP hydrolysis CMG is more likely to be in the $(n+1)$th potential well representing the next base along, giving rise to a bias towards the unwinding direction. **c** The proposed kinetic scheme for DNA unwinding by CMG. The enzyme is in an active state, unwinding via a Brownian ratchet mechanism but may enter one of three pauses. The kinetics governing theses process are dependent on force applied to DNA and the ATP concentration

to a weak free energy potential. This motion is unbiased and depicted in Fig. 5a. Upon ATP binding or hydrolysis CMG undergoes allosteric structural changes that alter DNA affinity, increasing the free energy potential of interaction with DNA. In this state, thermal energy is insufficient for CMG to overcome the energy barriers and so remains in position. The asymmetric potential, likely provided by helicase position and state causing variations in ATP binding or release kinetics[73], means that upon cycling between the two states CMG is more likely to be in the $(n+1)$th potential well, representing the next base along; giving rise to a random walk with bias in the unwinding direction (Fig. 5b)[65]. Under this mechanism, we do not believe CMG directly catalyses ATP hydrolysis-dependent DNA annealing, but rather permits annealing upon reverse motion. ATP hydrolysis gating and thermal noise[67] produces a biased random walk with inherent backward motion, allowing the DNA to anneal.

As a Brownian ratchet requires no rotary nucleotide firing or coordinated escort, only alterations in DNA affinity, this model agrees with the observation of functional asymmetry in CMG, where ATPase activity of two subunits can be mutated without abrogating helicase activity[6]. Instead we suggest a stochastic/probabilistic principle, as proposed previously for AAA+ motors[74], where allosteric changes provide the required difference in DNA affinity. The lack of requirement for a sequential

mechanism may also explain the ability of Mcm4,6,7 to act as a helicase, although with relatively poor activity[75].

The inherent plasticity of Brownian ratchet mechanisms in ATP hydrolysis timing and robustness against major errors in the hexameric core would be beneficial for maintaining genome stability. For example, accurate DNA replication requires bypassing of DNA lesions and roadblocks[76] as no further helicase loading occurs after G1 phase of the cell cycle and only already loaded origins are fired in S phase[3]. Similarly, the ability of the replisome core to exhibit complex dynamics and move backwards seems essential for fork reversal and remodelling[77]. Our suggested mechanistic model does not require the existence of six conformational states, in agreement with the lack of multiple rotational states observed in Cryo-EM studies of *Dm*CMG and *Sc*CMG on various DNA templates[13,19,25].

The long timescale FPTs identified as pause states exist outside the Brownian ratchet described motion and hinder helicase progression at endogenous rates. Consequently, we label CMG a lazy Brownian ratchet and suggest the kinetic scheme for helicase action shown in Fig. 5c. There are several kinetic schemes with different pathways that may explain our data (Supplementary Fig. 10). Here, striking a balance with simplicity, we consider the scheme consists of a Brownian ratchet for unwinding that may enter one of the three pauses.

To the best of our knowledge there are no known causes of pausing during DNA unwinding by replicative helicases and the origin of those observed here cannot be established from our current work. We coarsely categorise them into either force and ATP concentration dependent or solely ATP concentration dependent (Fig. 5c). Cryo-EM of *Sc*CMG bound to fork template with roadblocks on both strands shows dsDNA can partially enter CMG[19]. Perhaps this represents a non-productive pause state, which must be exited before unwinding can resume, or is prevented by additional factors such as Mcm10[78]. Our data shows larger forces increase the likelihood of performing a biased random walk, hence unwinding, decreases the probability of entering a short pause, and increases the rate of exit (Fig. 4 and Supplementary Fig. 9), indicating a disruption to the lagging-strand template may inhibit pausing. This correlates with previous work demonstrating a decrease in time spent pausing by DnaB as increasing force is applied to the leading (non-tracking) strand[32]. Further studies to achieve higher precision and hidden Markov modelling[79] would help elucidate pause origins and kinetic pathways.

There are limitations to our study. Due to finite time resolution we miss shorter timescale events. We are also limited in spatial resolution as the efficiency with which we obtain DNA unwinding events means we must utilise low magnification to multiplex our measurements over 100 s of molecules to obtain sufficient statistics. Although we use purified recombinant *Dm*CMG, that binds to the DNA fork substrate in a non-physiological manner, it is unlikely that the dynamics of CMG activated from a fired origin would behave differently, as recombinant CMG can support full replication[64].

The eukaryotic replicative helicase unwinds DNA by executing a biased random walk with frequent entrance into one of three long lifetime pauses. We find a pause-free unwinding rate would recover in vivo replication fork rates in vitro, leading to speculation that additional factors not present in our study, for example, polymerase ε, Mcm10, Mrc1, or Csm3/Tof1, may alter unwinding mechanism kinetics, as found for leading strand synthesis[8,9].

We propose a lazy Brownian ratchet is the simplest model to explain our observations. Remarkably, the relatively simple stochastic model can recapitulate and elegantly explain the heterogeneous dynamics, without requiring careful sequential escort of the DNA.

It is far more plausible the replicative helicase evolved to take advantage of the stochastic and energetic nature of Brownian motion rather than fight against it to perform a perfectly deterministic and sequential walking mechanism. Future higher precision work may refine the model and provide a quantitative description of the free energy landscape along which CMG translocates[80]. Many details of replisome component operation remain to be understood. Our framework provides a means to investigate how these components interact and function to produce the dynamics and kinetics observed.

## Methods

**Microscopy**. The magnetic tweezers hardware was custom built. All apparatus was, at minimum, switched on several hours before experiments began and allowed to equilibrate to 23 °C, to minimise vertical drift in the experiment.

The brightfield microscope was built on a vibration isolated optical table (Thorlabs, B7590 Nexus Breadboard, PFA51505 Active Isolation Frame) surrounded by a light tight isolation chamber for temperature and air flow stability. Illumination was provided by a 530 nm LED (Thorlabs, M530D2) without Köhler illumination using only collimation by an aspheric $f = 20.1$ mm condenser lens (Thorlabs, ACL2520U-A). A Nikon 50× NA 0.90 microscope objective lens (Nikon, MRL01502) in conjunction with a 2″ $f = 200$ mm achromatic doublet tube lens (Thorlabs, AC508-200-A) imaged the sample plane onto a Falcon2 12 M camera (Teledyne DALSA, FA-80-12M1H) via a LabVIEW compatible frame grabber (National Instruments, NI PCIe-1433, 781169-01). The sample chamber was mounted on an x,y translation stage (Märzhäuser, 00-30-101-0000) and

clamped in position with homemade stage clips. The microscope objective was mounted in a z-axis piezo stage (Physik Instrumente, PD72Z1CAQ) for focus adjustment.

**Magnetic tweezers**. Two neodymium cube magnets (supermagnete.de, W-05-N50-G) are epoxyed onto a microscope cover slide (Fisher Scientific, 12332098) with magnetic moments pointing vertically in opposite directions using two aluminium spacers to provide the desired magnet gap. An optically clear window of glass is left to allow illumination light to pass through onto the sample chamber. This magnet pair is mounted on a vertical translation stage (Physik Instrumente, M-112.1DG) to allow force alteration. The stage mounted magnet pair and illumination source are mounted on an x,y translation stage (Thorlabs, MT1B/M) for alignment above the microscope objective centre.

**Fluidics**. The outlets of the pre-prepared sample chamber was connected to a six-way selection valve (VWR, 560-0166) via additional tubing (VWR, 554-2962) and hubless needles (Hamilton, 21 G, 22021-01). The six-way valve was connected to a glass syringe (Hamilton 1000 series, 26211-U) via a three-way stopcock (Cole-Parmer, WZ-30600-02). Using such a six-way valve allows facile switching of multiple sample chambers. The syringe was withdrawn using a syringe pump (Harvard Apparatus, 704504).

**Magnetic tweezers control software**. Control of the objective vertical position (focus) and magnet vertical position (force) was performed with custom written LabVIEW code. An interactive camera capture and microsphere tracking programme was used to perform the experiments. The salient features of this software are the tracking in real-time of several microsphere positions using the algorithm described by van Loenhout et al.[81], and the export of individual images at up to 58 fps directly to a hard drive. Code and software is available at github.com/danielburnham.

**Microsphere tracking software**. Tracking of the microspheres for final analysis was performed with custom written MATLAB (Mathworks, 2015a-2017a) code based on the algorithm described by van Loenhout et al.[81]. This is available for download at github.com/danielburnham

**Force calibration**. The force applied by the two neodymium cube magnets is calibrated as a function of displacement from the sample chamber using the equipartition method[27] and corrected for the finite exposure time[82]. We used two different sized gaps between the magnet pairs, one of 0.5 mm and one of 1.5 mm to provide low and high range forces.

**dsDNA to ssDNA extension calibration**. To convert the microsphere displacement, hence DNA extension, $z_{meas}$, to number of base pairs unwound we performed force extension measurements of the dsDNA unwinding template and a denatured ssDNA sample, in CMG running buffer. The results are plotted in Supplementary Fig. 1b, and values used with

$$n_{bp} = \frac{z_{meas}(F) - z_{ds}(F)}{z_{ss}(F) - z_{ds}(F)} N_{tot} \qquad (3)$$

to calculate base pairs unwound, where $z_{meas}(F)$ is the measured DNA extension, $z_{ds}(F)$ and $z_{ss}(F)$ are the extensions of fully ds- and ssDNA respectively, and $N_{tot}$ is the length of the template in base pairs.

ssDNA was formed by denaturing dsDNA. Stock DNA (~10 ng/μl) was diluted 5-fold in Milli-Q. One microlitre of this diluted sample was further diluted 10-fold into 500 mM NaOH, heated at 37 °C for 10 mins and kept on ice. The resulting ssDNA is used in place of stock template DNA in the conjugation of DNA to microspheres.

**Sample chamber**. The sample chamber consists of three parts; a gasket, a functionalised coverslip, and a coverglass with holes. All following steps at room temperature unless stated otherwise.

Gaskets were cut with a scalpel from double-sided self-adhesive sheets (TESA SE, TESA 4965) to 24 × 40 mm rectangles with three 3 × 30 mm rectangles cut out.

Glass coverslips were functionalized by first placing five 24 × 40 × 0.17 mm coverslips (Hecht-Assistent, 41014542) were placed in a staining jar (Sigma-Aldrich, S5641). The coverslips were immersed in ethanol and sonicated in an ultrasonic bath (Branson, M3800-E, 142-0133) for 30 minutes, rinsed with Milli-Q water (Millipore), then sonicated in 1 M KOH for 30 minutes, and again rinsed in Milli-Q. The ethanol, Milli-Q, KOH, Milli-Q steps were repeated before finally being left immersed in Milli-Q.

Milli-Q was decanted from the jar and the coverslips rinsed with acetone three times. Upon each rinse the acetone was carefully decanted over the inside of the jar lid to remove any traces of water. During the third rinse the coverslips were sonicated for 10 min in a water bath. Finally, the coverslips were immersed in fresh acetone.

Each coverslip was next treated to add silica reference microspheres permanently to the glass. Tracking of the position of such stuck microspheres

describes the motion of the coverslip. Three micrometer diameter silica microspheres (Bangs Laboratories, SS05001) were diluted 1:100 in ethanol and a 4-µl elongated bead of the suspension placed along the short edge of the coverglass before being swiped along the coverslip long axis with a pipette tip. The coverslip was heated on a hot plate for 10 mins at 70 °C and then placed back in acetone in the staining jar.

With haste; a 2% (v/v) silane solution was prepared by mixing 2 mL of 3-aminopropyltriethoxysilane (Sigma-Aldrich, A3648) in 100 mL acetone. The acetone from the staining jar was decanted as described above and the jar refilled with the 2% silane solution. The jars were then shaken horizontally in all directions for 120 seconds. Next the jar was rapidly filled with ~2 volumes of Milli-Q (without decanting the previous solution). The staining jar was then filled, shaken, and decanted 6 times with Milli-Q before leaving the coverslips immersed in Milli-Q.

The coverslips were placed on concertinaed aluminium foil and placed in an oven at 110 °C for 1 hour.

The backing paper of the previously made gaskets are removed and placed adhesive side up on a bench before placing the now dry coverslip silanised side down onto the gasket. Gentle pressure is applied to seal the gasket to the glass.

The coverslip/gasket combination is placed tape side up, leaving the backing plastic attached, in a closable chamber. With haste; to pegylate the remaining negative space made by the gasket a solution of 150 mg of mPEG-SPA (Laysan Bio, Inc., MW 5000) and 2 mg biotin-PEG-CO₂NHS (Laysan Bio, Inc., MW 5000) dissolved in 1 mL freshly made 100 mM NaHCO₃ (pH 8.2) was pipetted onto the channel created by the gasket to leave a tube of PEG solution. A small water reservoir is placed in the chamber before closing to prevent excess evaporation and the coverslips left for 3 hours.

Finally, each coverslip was rinsed thoroughly with Milli-Q and dried with a filtered compressed air or nitrogen gun. These pegylated coverslip/gaskets can be kept under vacuum for ~2 months.

The coverglass was made by creating holes in either a $50 \times 50 \times 0.4$ mm cover glass (GPD-5504, UQG Optics) or a $26 \times 70 \times 1$ mm cover glass (Fisher Scientific, 12332098) with either a dental sandblaster (Kent Express, Danville Microetcher Mark II, 89516) using aluminium oxide (Kent Express, Danville Aluminium Oxide Powder, 89517) or a Dremmel drill with diamond bit (UKAM Industrial Superhard Tools, 2030008), respectively, with spacings to match the ends of the rectangles cut to make the gasket. The inlet holes and outlet holes are made to snuggly fit Intramedic Polyethylene Tubing PE20 (Becton Dickinson, 427406) and PE60 (Becton Dickinson, 427416) respectively.

Finally, a pre-prepared pegylated coverslip/gasket is removed from vacuum and the plastic adhesive tape backing removed and placed adhesive side up on a bench. The coverglass with holes is aligned and placed on top. Gentle pressure is applied to seal the tape to the glass.

The Intramedic Polyethylene Tubing PE20 and PE60 are cut to 10 cm lengths and placed into opposite holes of what now spans each sample chamber. The tubing is fixed in place with epoxy (RS Components, 756–0102).

After the epoxy has cured ~20 µl of stock streptavidin (1 mg/mL Streptavidin) (Sigma-Aldrich, S4762) in phosphate-buffered saline (Thermo Fisher, 10010015) is drawn into the flow chamber manually with a syringe and left at room temperature for 30 mins. Finally blocking buffer is drawn into the chamber manually at speed to remove any bubbles in the tubing and air pockets in the chamber, before being kept at 4 °C until required.

**Anti-digoxigenin conjugated magnetic microspheres.** 2.8 µm diameter anti-digoxigenin super-paramagnetic microspheres were prepared by conjugating anti-digoxigenin fab fragments (1 mg/ml) (Sigma-Aldrich, 11214667001) to M280 tosyl-activated magnetic microspheres (Thermo Fisher, 14203) as per the manufacturers protocol.

**Calibration DNA template.** Oligos and primers for DNA templates are given in Supplementary Table 2. For force calibration, we used a lambda DNA template with biotin and digoxigenin handles at opposite ends. It was constructed from 3 parts.

A biotin incorporated handle was produced by amplifying a 574 bp duplex from within pUC19 with forward primer oligo 1 and reverse primer oligo 2, using Phusion polymerase (NEB, M0530L). The PCR was carried out in the presence of modified dUTP, biotin-16-dUTP (Enzo, ENZ-42811), at a molar ratio of 1:20 with unmodified dNTP mix. Each handle was PCR sample purified (Qiagen, 28104) into 10 mM Tris-HCl, pH 8.5. DNA was nicked with Nt.BspQI (NEB, R0644L) at 50 °C for 3 hours to generate a 12-nt 3′ overhang on the PCR template that is complementary to one end of lambda DNA. Nicked DNA was heated at 65 °C for 10 min to generate the desired ssDNA overhang in the presence of ~2 µM oligo 3 that prevents re-annealing of the overhang to the complementary strand. DNA was separated on 1.5% agarose and purified (Qiagen, 28704).

A digoxigenin incorporated handle was produced by amplifying a 574 bp duplex from within pUC19 with forward primer oligo 1 and reverse primer oligo 4, using Phusion polymerase (NEB, M0530L). The PCR was carried out in the presence of modified dUTP, digoxigenin-11-dUTP (Roche, 11093088910), at a molar ratio of 1:20 with unmodified dNTP mix. Each handle was PCR sample purified into 10 mM Tris-HCl, pH 8.5. After nicking with Nt.BspQI at 50 °C for 3

hours, oligo 5 was added and heated at 65 °C for 10 min, before separating on 1.5% agarose, extracting, and purifying.

The lambda DNA (NEB, N3011L) was phosphorylated with T4 Polynucleotide Kinase (NEB, M0201S) for 4 hours at 37 °C and then heated at 70 °C for 20 mins.

To ligate the biotin incorporated handle to lambda DNA ~10-fold molar excess of the biotin handle was heated with the lambda DNA to 65 °C and allowed to cool to room temperature on a heat block before ligation (T4 Ligase, NEB, M0202L) overnight at room temperature. Next, it was separated on a 0.5% agarose gel, excised, and purified using electroelution with 12–14 kDa MWCO dialysis tubing (VWR, 734-0672). Finally, DNA was concentrated with Vivaspin 500 10 kDa MWCO (Generon, VS0102).

To ligate the digoxigenin incorporated handle to the biotin handle/lambda DNA ~20-fold molar excess of digoxigenin handle was heated with the biotin handle/lambda DNA to 60 °C for 1 min and allowed to cool to room temperature before ligation (T4 Ligase, M0202L) overnight at room temperature. The DNA was separated on a 0.5% agarose gel, excised, and purified using electroelution with 12–14 kDa MWCO dialysis tubing (VWR, 734–0672). The gel slice was removed from the tubing and the sample dialysed against 10 mM Tris pH 8 overnight at 4 °C. Finally, DNA was concentrated with Vivaspin 500 10 kDa MWCO (Generon, VS0102).

**Unwinding template.** The 3′ flap template for unwinding studies was constructed from five parts.

A 2711-bp duplex was PCR amplified from within plasmid pUC19 using forward primer oligo 6 and reverse primer oligo 7 using Phusion polymerase (NEB, M0530L), followed by PCR sample purification into 10 mM Tris-HCl, pH 8.5. After nicking with Nt.BbvCI (NEB, R0632S) at 37 °C for 3.5 hours, oligo 8 and 9 were added and heated at 50 °C for 10 min and again purified. Finally, the sample was separated on 1% agarose before gel extraction and purification.

Two 568 bp duplex handles were amplified from within plasmid pUC19 using forward primer oligo 1 and reverse primer oligo 10 with Phusion polymerase (NEB, M0530L). The PCR was carried out in the presence of modified dUTP, digoxigenin-11-dUTP (Roche, 11093088910) or biotin-16-dUTP (Enzo, ENZ-42811), at a molar ratio of 1:20 with unmodified dNTP mix. Each handle was PCR sample purified in to 10 mM Tris-HCl, pH 8.5. After nicking with Nt.BbvCI (NEB, R0632S) at 37 °C for 3.5 hours, oligo 11 was added and heated at 50 °C for 10 min and again purified.

A 3′ fork spacer was formed by placing equimolar ratios of oligo 12, oligo 13, and oligo 14 in 10 mM Tris-HCl (pH 8), 100 mM NaCl, 1 mM EDTA and heating to 85 °C for 10 mins, then allowed to anneal by cooling to room temperature. Bands were extracted from an 8% PAGE gel and purified using electroelution with 12–14 kDa MWCO dialysis tubing (VWR, 734-0672). Finally DNA was concentrated with Vivaspin 500 3-kDa MWCO (Generon, VS0192).

Spacer 2 was formed by placing equimolar ratios of oligo 15, oligo 16 and oligo 17 in 10 mM Tris-HCl (pH 8), 100 mM NaCl, 1 mM EDTA were heated to 85 °C for 10 mins, then allowed to anneal by cooling to room temperature. Bands were extracted from an 8% PAGE gel and purified using electroelution with 12–14 kDa MWCO dialysis tubing. Finally, DNA was concentrated with Vivaspin 500 3 kDa MWCO.

The digoxigenin handle was ligated (T4 Ligase, M0202L) to the 3′ fork spacer at 10:1 molar ratio, overnight by cycling through the temperatures 16 °C, 20 °C and 25 °C in 1 hour repetitions. The same was repeated for the biotin handle and spacer 2. Each were separated on 1% agarose gel before extraction and purification. These two products were ligated (T4 Ligase, NEB, M0202L) to the 2.7-kb template at room temperature for 2 hours at equiweight ratio. The final product was separated on, and extracted from, a 0.8% agarose gel before being purified and stored at −20 °C.

**Bulk unwinding template.** Equimolar ratio of oligos 18 and 19 were mixed in STE buffer and incubated at 85 °C for 3 mins then allowed to cool to room temperature. After separation on 3% agarose gel the band corresponding to the fork template was excised and gel purified. Subsequent to purification, the template was radi-olabelled with [γ³²P]-ATP at 5′ ends using T4 PNK. To eliminate excess [γ³²P]-ATP the sample was passed through MicroSpin G50 columns (GE Healthcare, 27-5330-01) previously equilibrated with 10 mM Tris-HCl, pH 8.0, 20 mM NaCl.

**Hairpin template for large T antigen experiments.** The hairpin template for unwinding studies was constructed from five parts.

A 1046-bp duplex was PCR amplified from within plasmid pUC19 using forward primer oligo 20 and reverse primer oligo 21 using Phusion polymerase (NEB, M0530L), followed by PCR sample purification into 10 mM Tris-HCl, pH 8.5. After nicking with Nt.BbvCI (NEB, R0632S) at 37 °C for 3.5 hours, oligo 8 and 9 were added and heated at 50 °C for 10 min and again purified. Finally, the sample was separated on 1% agarose before gel extraction and purification.

Digoxigenin and biotin incorporated handles were prepared as described for the unwinding template.

An Upper Linker was constructed by heating equimolar ratios of oligo 22 and oligo 12 in 10 mM Tris-HCl (pH 8), 100 mM NaCl, 1 mM EDTA at 85 °C for 10 mins, then allowed to anneal by cooling to room temperature. Bands were

extracted from an 8% PAGE gel and purified using electroelution with 12–14 kDa MWCO dialysis tubing (VWR, 734–0672). Finally, DNA was concentrated with Vivaspin 500 3 kDa MWCO (Generon, VS0192).

A Lower Linker was constructed by heating equimolar ratios of oligo 17 and oligo 23 in 10 mM Tris-HCl (pH 8), 100 mM NaCl, 1 mM EDTA at 85 °C for 10 mins, then allowed to anneal by cooling to room temperature. Bands were extracted from an 8% PAGE gel and purified using electroelution with 12–14 kDa MWCO dialysis tubing. Finally, DNA was concentrated with Vivaspin 500 3 kDa MWCO.

The digoxigenin handle was ligated (T4 Ligase, M0202L) to the Upper Linker at 10:1 molar ratio, overnight by cycling through the temperatures 16 °C, 20 °C and 25 °C in 1 hour repetitions. The same was repeated for the biotin handle and the Lower Linker. Each were separated on 1.5% agarose gel before extraction and purification.

The digoxigenin/Lower and biotin/Upper ligation products were annealed at equimolar ratio in 10 mM Tris-HCl (pH 8), 100 mM NaCl, 1 mM EDTA. After heating at 50 °C for 5 mins the sample was heated at 40, 35, 30, and 25 °C, consecutively, each for 1 hour. The sample was separated on 1% agarose gel before extraction and purification into 10 mM Tris-HCl, pH 8.5, before adding NaCl to 10 mM.

The final hairpin was formed by ligating the 1 kb template, the handle/linkers, and oligo 24 (T4 Ligase, NEB, M0202L) at 1:3:30 molar ratio, overnight by cycling through the temperatures 16 °C, 20 °C, and 25 °C in 1 hour repetitions. The final product was separated on, and extracted from, a 1% agarose gel before being purified.

**Protein purification**. SV40 large T antigen was expressed in insect cells and purified using a monoclonal antibody as described previously[15]. Briefly, full length SV40 large T-antigen gene was cloned into pFastBac1 (ThermoFisher), which was used to make the baculovirus. Sf21 cells maintained in SF-900-III were infected with the virus ($2 \times 10^8$ pfu/ml) using an MOI of 0.1 for 72 hours. The 1 L Cultures grown in 5 L flasks.

Antibody PAb419 was coupled to Protein A sepharose beads (5 mg ml$^{-1}$) in PBS, incubated overnight, rotating in a cold room. Beads were washed with 15 ml 0.1 M sodium borate, pH 9.0, and re-suspended in 2 ml. 20 ml of 0.0125gml$^{-1}$ dimethyl pimelimidate was mixed with the beads and incubated, rotating, at room temperature for 1 hour. Coupling reaction was halted by incubating beads in 0.2 M ethanolamine, pH 8.0, rotating for 1 hr at room temperature.

Cell pellets were re-suspended in 10 pellet volumes of L-Tag re-suspension buffer and incubated on ice for 15 mins. The suspension was centrifuged at $25,000 \times g$ for 15 mins. 0.5 volumes of L-Tag neutralisation buffer was added and mixed. Sample was first loaded onto protein A-only column, equilibrated with L-Tag loading buffer. Flow-through was then loaded onto PAb419-conjugated protein A column equilibrated with L-Tag loading buffer. The column was washed with 50 ml L-Tag loading buffer, then 50 ml of L-Tag wash buffer, followed by 20 ml L-Tag EG buffer.

Large T-antigen was eluted with 5–10 ml of L-Tag elution buffer before dialysis overnight in L-Tag dialysis buffer.

11 subunits of *Dm*CMG (plasmids provided by Costa Lab) were co-expressed using the baculovirus expression system[13]. Expression and purification of the complex were carried out as described in Abid Ali et al.[13] and Ilves et al.[6]. With minor changes the method is outlined here.

Briefly, following bacmid generation for each subunit of *Dm*CMG, Sf21 cells (Structural Biology, STP, The Francis Crick Institute) were used for the initial transfection and in the subsequent virus amplification stage to make P2 stocks using serum-free Sf-900TM III SFM insect cell medium (Invitrogen, 10902-096). Virus was amplified to make a P3 stock by inoculating 100 ml of Sf9 cell cultures (Cell Services, STP, The Francis Crick Institute) ($0.5 \times 10^5$ ml$^{-1}$) with 0.5 ml of P2 stocks with MOI ≈ 0.1 for each subunit virus. The resulting cultures were incubated in 500 ml Erlenmeyer sterile flasks for 4 days at 27 °C on a cyclic shaker at 100 rpm. Supernatant was filtered, after centrifugation at ~$1000 \times g$ for 15 mins.

4 L of Hi five cells (Cell Services, STP, The Francis Crick Institute) cultured in Graces medium supplemented with 10% FCS ($1 \times 10^6$ ml$^{-1}$) were infected with the fresh 11 subunit virus P3 cultures with MOI ≈ 5. Cells were incubated at 27 °C and harvested after 60 hours. The resulting pellets were first washed with PBS (Gibco, ThermoFisher, 70011044) supplemented with 5 mM MgCl$_2$ and resuspended in 200 ml resuspension buffer C and frozen in 10 ml aliquots on dry ice before storage at −80 °C ready for purification.

Cell pellets were thawed and lysed in a Dounce homogeniser (Wheaton, 40 ml Dounce Tissue Grinder) for at least 50 strokes per 30 ml of re-suspended cell pellets. KCl was added to achieve final 100 mM concentration in lysed cell suspension. The lysate was centrifuged at $24,000 \times g$ for 10 mins. The supernatant was incubated with 2 ml buffer C equilibrated M2 agarose beads (Sigma Aldrich, F3165) for 2.5 hours with end-over-end mixing. The supernatant was discarded following a centrifugation at $200 \times g$ for 5 mins and the beads were washed with 30 ml of buffer C-100. The beads were incubated with 5 ml elution buffer, supplemented with 200 μg/ml peptide (DYKDDDDK, Peptide Chemistry, STP, The Francis Crick Institute), at room temperature for 15 mins with end-over-end mixing to elute bound proteins. Flow through was collected and the process

repeated with a further 4 ml elution buffer, supplemented with 200 μg/ml peptide, and 10 mins mixing. Finally both flow throughs were pooled. The eluate was passed through a 1-ml HiTrap SPFF column (GE Healthcare, 17-5054-01) equilibrated with buffer C-100. The flow-through was collected and pooled with a further 4 ml of buffer C-100 passed through the column.

*Dm*CMG complex was separated with a 20 ml 100-550 mM KCl gradient using 5/50GL MonoQ column (GE Healthcare, 17-5166-01). Fractions where *Dm*CMG was eluted included ~400-450 mM KCl. Fractions of CMG were pooled and diluted to ~150 mM KCl. To further concentrate the sample, pooled fractions were loaded onto MonoQ PC 1.6/5GL (GE Healthcare) column equilibrated with buffer C-150-No Tween. A 2 ml 150-550 mM KCl gradient was applied to separate *Dm*CMG complex and the fractions containing *Dm*CMG were pooled and dialysed against 1 L dialysis buffer for 2 hours. Aliquots were flash frozen using liquid nitrogen and kept in -80 °C. Protein purification was checked by SDS-PAGE – original gel image given in Supplementary Fig. 11a.

**Bulk unwinding assay**. The bulk unwinding fork templatehas 60 bp duplex DNA with a 40-bp-long polyT region at the 3′ end for CMG binding and a GC-rich region at the 5′ end. It was radiolabelled with [γ32P]-ATP at 5′ ends, using T4 PNK, and was incubated with CMG in reaction buffer (25 mM HEPES, pH 7.5, 10 mM magnesium acetate, 5 mM NaCl, 5 mM DTT, 0.1 mg/ml BSA) in the presence of ATPγS (500 μM) for 2 hours to achieve successful loading of the helicase on DNA. In each reaction, DNA:protein ratio was kept to a minimum of 1:50. Unwinding was initiated by adding 10-fold excess ATP (5 mM final concentration) and stopped after 5 mins by adding reaction stop buffer containing 0.5% SDS and 20 mM EDTA. Substrates were separated using 12% polyacrylamide native gel. Gels were mounted on Whatman paper, exposed to phosphor imager screen overnight and scanned using a Typhoon 9500 (GE Healthcare) before pixel value linearisation with ImageJ. Original gel is shown in Supplementary Fig. 11b.

**Single-molecule DNA unwinding assay**. To bind DNA constructs to magnetic microspheres it should be noted that values with * indicate parameters that need to be adjusted to obtain an optimal number of single DNA molecules tethered between glass surface and microsphere at an ideal density.

In total, 1 μl* of anti-digoxigenin conjugated magnetic microspheres were re-suspended in 100 μl blocking buffer and the supernatant removed after separation with a magnet. This step was repeated. Next the microspheres where re-suspended in 100 μl binding buffer and the supernatant removed after separation. After re-suspending in 4 μl* of binding buffer, 1 μl* of diluted stock DNA construct (~10 ng/μl stock diluted 500* fold with MilliQ) was added and left on a tube rotator (VWR, 445-2102, Stuart SB3) at room temperature and 10 rpm for 20 mins to prevent sedimentation.

The DNA conjugated microspheres are separated with a magnet and supernatant removed before re-suspension in 200 μl of blocking buffer. This buffer is removed in the same manner and replaced with 40 μl* blocking buffer.

To bind DNA-conjugated microspheres to the sample chamber surface the sample chamber is first removed from 4 °C, mounted on the microscope and allowed to reach temperature equilibrium. The outlet tubing is connected to the six-way valve and the inlet tubing placed in a 1-mL eppendorf tube of blocking buffer at room temperature. An initial high speed pulse of blocking buffer is drawn into the sample chamber to remove bubbles in the flow system. 200 μl of blocking buffer is drawn through the chamber at 20 μl min$^{-1}$ with judicious flicking of the inlet and outlet tubing for the first 100 μl to remove trapped air.

The magnetic force is set to ~0 pN and with the flow stopped and isolated the inlet tubing is placed in the DNA conjugated microsphere sample before flowing at 20 μlmin$^{-1}$ until a high density of microspheres is present in the chamber. The flow is stopped and isolated to ensure no microsphere movement due to leaks or capacitance in the fluidics and the inlet placed back in blocking buffer.

The flow is restarted at 4 μl min$^{-1}$ to prevent clustering of microspheres and over the space of ~15 mins increased to 10 μl min$^{-1}$ and then to 20 μl min$^{-1}$ until all non-bound microspheres are removed. This stage is open to adjustment with the aim of optimising the density of single-molecule DNA bound microspheres present, often with the help of careful outlet tube flicking to remove loose microspheres.

Finally, the force is increased and a field of view chosen with the maximum number of single DNA-microspheres at an ideal density for analysis, and with at least two reference microspheres. If necessary the procedure is repeated to increase the density of available DNA molecules.

A look up table (LUT) is created to allow sub-pixel vertical tracking of microspheres. Once a sufficient number of DNA conjugated microspheres are tethered to the glass surface, 100 μl of CMG loading buffer is drawn into the sample chamber at 20 μl min$^{-1}$. The force is set to that to be used in the experiment. The focus is set to just above a force stretched DNA molecule and microsphere and the objective stepped in 100 nm increments above the focus a distance greater than the contour length of the DNA construct. At each step a single image is taken and saved for later analysis.

**CMG unwinding assay**. With the flow turned off and isolated the force is set at 7 pN, then 0 pN, and back to 7 pN, each for ~25 s. This data is later used to both

screen for DNA molecules of the correct length and as a point of reference for the absolute position of the sample chamber bottom. The magnet force is set to 8 pN.

Stock CMG is taken directly from −80 °C freezer and diluted to 42 nM in CMG loading buffer. The sample is directly drawn into the flow chamber at 20 μl min⁻¹. After 8 mins at 8 pN the magnet force is set to 3.5 pN for a further 15 mins. The force is then set to that required and left for ~70 seconds to record a reference point. Next, 40–50 μl (equal to 4–5 flow chamber volumes) of temperature equilibrated CMG running buffer is drawn into the flow chamber and the flow turned off. The experiment is left for up to 2 hours before ending, having collected ~0.5 million image frames.

Any used sample chamber is not used again.

**SV40 large T antigen unwinding assay**. For the hairpin template assay the flow is turned off and isolated and force is set at 9 pN. The DNA tethers are identified as hairpins by varying the force between 4 pN and 25 pN to open and close the construct. The force is returned to 9 pN. For the linear template assay the flow is turned off and isolated and the force is set at 7 pN, then 0 pN, and back to 7 pN, each for ~25 s. The force is finally set to 20 pN.

For both constructs stock large T antigen is diluted to 110 nM (monomer) into 20 μl CMG running buffer. The sample is heated at 37 °C for 20 mins before being drawn into the flow chamber at 20 μl min⁻¹. The flow is again isolated and data recorded.

**Data analysis**. Raw images saved to hard disk go through several steps in an analysis workflow. The salient points are discussed here.

All microspheres that exhibit expected behaviour and have the expected length under a high to low force transition are tracked through time from the image frames exported during the experiment. The resulting z position trajectories have the z position of the mean of at least two reference microspheres subtracted point for point and set to the zero level of the pre-unwinding trajectory.

The raw z position data is low pass filtered using a 6-s moving mean average. This 6 s filter size is selected from studying the enzyme-free motion of the DNA-tethered microsphere at 20 pN force. The sum of the squared difference between raw and filtered data points is plotted as a function of filter size (Supplementary Fig. 5a). At a certain filter size the benefit of further increases has diminishing returns and only serves to smooth the signal rather than increasing the signal to noise ratio. As such a filter size is chosen from the point at which the sum of differences between raw and filtered data reaches a plateau.

Mean linear velocities of unwinding are found through a linear fit to the filtered data.

The passage interval is chosen by determining twice the standard deviation of the signal when filtered with the filter size chosen above. For illustration, we plot in Supplementary Fig. 5b how the standard deviation, $\sigma$, of an 80 s enzyme-free trajectory at 20 pN varies as a function of filter size. For the 6 s filter size chosen above $\sigma = 10$ bp. To avoid spurious interval crossing and hence first passage times the passage interval, m, is set to 20 bp $= 2\sigma$[28]. The time taken for the trajectory to first pass each consecutive interval was measured until the end of the trajectory was reached. All first-passage times under the same conditions were concatenated and used in the fitting procedure. The parameters extracted are resistant to changes in passage interval, $m$, beyond the associated uncertainties, as the choice of interval for raw data analysis is reflected in the model fitting with Eq. 2.

The model first-passage time probability distribution (Eq. 2) was used to find maximum likelihood estimators for the model parameters. The maximum likelihood estimation is performed using *fmincon* in MATLAB (Mathworks, 2014a-2017a). The choice of model, unidirectional walker or biased random walk, as well as the number of pauses to include was made by calculating the Bayesian Information Criterion (BIC)[83] for each and selecting the model with the minimum BIC as the best fit. This criterion penalises for model complexity, i.e., the more parameters, the larger the BIC. Greater detail on fitting multi parameter models can be found in references[56,84] and those within.

For display of the data the first-passage times were binned, normalised by bin width and total samples and plotted. The model parameters found through MLE are used to overlay the resulting analytical expression for the model best describing our data. We note that the model is not fit to the histogram.

The standard deviation error bars on the experimental first-passage time data bins and the standard deviation reported with estimated parameter values were calculated through bootstrapping with replacement 1000 times.

**Buffers**. A list of buffers and contents are given in the Supplementary Methods.

**Reporting summary**. Further information on research design is available in the Nature Research Reporting Summary linked to this article.

## Data availability
Data of this study are available from the authors upon reasonable request.

## Code availability
Custom computer code used in this work is available for download at github.com/danielburnham. LabVIEW (National Instruments, 2012) was used for instrument control and MATLAB (Mathworks, 2015a–2017a) for analysis.

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

## Acknowledgements

Additional thanks to Remigijus Lape, Sherry Xie, Justin Molloy, Ferdos Abid Ali, Alessandro Costa, Dean Astumian and Stuart Horswell for fruitful discussions. This work was supported by the Francis Crick Institute which received its core funding from Cancer Research UK (FC001221), the UK Medical Research Council (FC001221) and the Wellcome Trust (FC001221). Also, this work has received funding from the European Union's Horizon 2020 research and innovation programme under the Marie Skłodowska-Curie grant agreement No 657479.

## Author contributions

D.R.B. and H.Y. conceived and designed the study. D.R.B. built the apparatus, wrote the software, and performed experiments and ran analysis. H.B.K. purified *Dm*CMG and performed bulk unwinding assay. D.R.B., R.B.H., and H.Y. designed the FPT model. D.R.B. and H.Y. interpreted results and wrote the paper with input from H.B.K. and R.B.H.

## Additional information

**Competing interests:** The authors declare no competing interests.

