## [Peer Review File · Nature Communications]

Reviewers' comments:

Reviewer #1 (Remarks to the Author):

In this manuscript by Burnham et al. they investigate the unwinding dynamics of recombinant *Drosophila melanogaster* CMG helicase in vitro using a single-molecule magnetic tweezer DNA unwinding assay. In isolation under the conditions used in this assay, CMG exhibits a biased random walk, where forward translocation of the helicase is complicated by pausing and annealing events. The authors construct a kinetic model that suggests CMG might enter several distinct paused states during unwinding. The observed helicase activity is distinct from those observed previously, and the authors propose that XMG works via a 'Lazy Brownian Ratchet' model. The manuscript is timely and should be of interest to those working in DNA replication and repair although there are a number of concerns that need to be addressed.

Major Comments

(1) One of the more surprising results of this manuscript is the reported rates for CMG are exceedingly low. The authors argue that high force should enable faster unwinding rates but there are a few problems with this statement. First, the authors claim that the unwinding rate increases with force (as shown in Figure 2a) but the authors show no statistical tests that the distributions, which look fairly similar as plotted, are actually distinguishable. Second, the authors are applying high forces, at least from a biological context, and do not test that the unwinding rate may be faster at lower forces (for example 2 pN).

(2) The authors never clearly show what their spatial resolution is in this assay. SI Fig 2a shows a control experiment with CMG but no ATP that would answer this question but it is too zoomed out to see.

(3) Along the lines of Comment (2) it is unclear how the authors selected the passage interval. SI Fig 5 provides some insight but more would be helpful as this seems to be a critical part of the analysis.

(4) Are the kinetic parameters extracted from the FPT distribution resistant to changes in the passage interval?

(5) Intuitively I would have predicted that increasing force would bias against reannealing yet SI table 1 shows that r_b actually increases with force at matched ATP concentrations. Can the authors interpret this result?

(6) Could the authors add RPA or even SSB to rxns after loading helicases to determine whether the reverse translocation is slippage, disengagement or active annealing?

(7) Could dissociation and re-association of CMG be responsible for any of the pausing observed? It seems possible, but is overlooked as an explanation.

(8) Some of the conclusions drawn from the model parameters do not appear to be particularly rigorous (Fig. 4). The authors state "The probability of entering the medium and long pause states are constant as force varies but entrance into the biased random walk or short pause are affected (Fig. 4a and 4c)." I must be misreading the figure because "rw" in Fig. 4A looks flat to me.

(9) The authors have attempted to "establish a biophysically sensible description that explains how ATP hydrolysis is coupled to unwinding and gives rise to such dynamics." Unfortunately, I am confused by the model. Why, for example, does decreasing ATP concentration lead to decreased pausing where in Fig 2 it was shown that increased ATP leads to increased rates of unwinding?

Minor Comments

Page 5, paragraph 1, last sentence

'Super family 3 viral replicative helicases show relatively slow unwinding rates when observed as a purified system [54] or in cell extract in vitro [55] and at high precision show unusual non-monotonic behaviour [29], more closely matching our data.'

The authors should highlight the close relationship between SF3 viral helicases and the SF6 helicases (MCM) as there may be a common basis for the observed slower unwinding rates. These are both AAA+ ATPases and translocate 3' to 5'. The faster bacterial and phage helicases belong to SF4 and SF5, and translocate 5' to 3'.

Figure 3c

The authors directly compare the first passage time distributions for two data sets at different forces (20pN and 45pN), but the [ATP] is also different so any differences might not be solely due to the force difference, as they claim [ATP] is also a factor that alters the unwinding rates in Fig 2b. This comparison should be done so that force is the only difference between experiments.

Reviewer #2 (Remarks to the Author):

This manuscript describes a detailed analysis of the *Drosophila* CMG helicase using single-molecule approaches. The authors tether an artificial fork DNA to the surface of the slide at the dsDNA end and a magnetic bead to the non-translocating ssDNA strand of the replication fork. They monitor the extent of unwinding by measuring the increased distance between the slide and the bead as dsDNA is converted to (longer) ssDNA by the helicase. The authors use a modified version of a CMG assay in which they first bind the complex to the template in the presence of ATP γ S and then activate unwinding by addition of ATP. The authors find that the CMG complex alternates between unwinding of DNA, reannealing of DNA and multiple paused states. Mathematical modeling of data suggests that there are three different types of paused states. Using these models, the authors predict how various properties of the CMG will respond to changes in ATP concentration and force applied to the DNA.

The strength of these studies is their ability to explain the very poor CMG helicase activity previously observed in ensemble/bulk studies and the exclusion of very simple models of CMG function in which it never releases from the DNA (e.g. similar to the model proposed for E1 helicase function). It is noteworthy, however, that a different single-molecule assay for unwinding (that is less able to follow long movements of the helicase) also showed a distribution of pausing, forward movement and reverse movement for E1 (PNAS USA, 2014 Mar 4; 111(9):E827-35). Thus, it remains unclear how different the CMG is relative to other helicases. Although not essential, it would improve the paper for these authors to look at another hexameric helicase (DnaB, E1 or SV40) using the same assay to provide a contrasting set of observations. The large number of pauses and reversals observed by the authors provide strong evidence that the eukaryotic helicase in this form is inefficient. These findings suggest that there will be additional factors that impact the CMG to allow it to attain the speeds of unwinding observed in vivo (e.g. DNA polymerases, Mcm10 or the Mrc1-Csm3-Tof1 complex).

The authors suggest that this enzyme is an inefficient Brownian ratchet. Although some of the data fits this view, other data is less compelling. For example, it seems odd that there is only a minimal effect of the force on the rate of unwinding (Fig. 2). It would seem that pulling on the non-translocating strand would cause increased rates of breathing at the fork facilitating a Brownian

ratchet mechanism of DNA unwinding. In contrast, going from 20 to 30 pN has no effect on unwinding rates and a further increase to 40 pN results in no change in many of the data points with a smaller number of points that show an increased rate. The authors merely summarize these data as showing that force increases unwinding rates but this is clearly not always the case. A discussion of why this would not be true between 20 to 30 pN is needed. It would improve the study to include biological replicates of some of the key experiments to show that the unusual response to force and ATP concentration is confirmed (see specific points below).

In summary, this manuscript provides compelling evidence explaining the inefficiency of the CMG helicase. The authors also provide evidence restricting the type of mechanisms by which the CMG helicase functions, although, it remains unclear how different this mechanism is from other replicative DNA helicases due to the lack of comparative data.

Specific points:

1. Why does reannealing occur when the DNA is under relatively high amounts of force. Do the authors believe that this is catalyzed by the CMG? If so, is this activity ATP-dependent?
2. The data in Fig. 2 does not strongly support the authors contention that there is both an ATP concentration and force dependent increase in rates (only the 40 pN data is different and that data shows extensive variability). Additional experimental replicates and more statistical analysis would be appropriate.
3. Given the central nature of the mathematical modelling to the manuscript the authors should provide more evidence of the confidence of the fits of the proposed model. This is particularly true of the three paused states. This would strengthen the manuscript and stimulate discussion in the community. For example, in the fit parameters shown in Figure 4c, why would the "diffusion coefficient" (in bp/s) peak at 30 pN, and then decrease at higher and lower force? Similarly in Figure 4d, there is no constant trend in ATP-dependence on the "diffusion coefficient". Do authors expect that there is a dependence of diffusion coefficients or some other process is responsible at that force and ATP concentration? These results are not consistent with the statement from the authors that "With increasing force the overall trend is an increased effective diffusion (Fig. 4c)..." There are similar concerns for the exit rates in response to force (Fig. 4e).
4. The authors propose that there are interactions between the non-translocating strand and the CMG (Fig. 5d – note, there is no call out for this part of the figure in the paper) but the evidence for this conclusion from this paper is limited. If the authors want to make this proposal (especially in the form of an illustration in a figure) they should discuss the evidence in support of it directly. Do the authors mean to suggest that this interaction is important for the distinction between binding to n and $n+1$ states illustrated in Figs. 5a and 5b?
5. The data in Supplemental Table 1 indicates that r_b and r_f increase with lower ATP concentrations. How does this fit into the author's model? Similarly, the authors should address why there is a significant peak in r_f and r_b at 30 pN relative to 20 pN and 40 pN. Why would this be?

Other points:

1. The data discussed in the third full paragraph on page 9 do not match the corresponding data in Supplementary Table 1 (e.g. r_b is 115 in the table and the text gives $r_b = 109$). Since this difference changes v_{mean} substantially this should be clarified.

Reviewer #3 (Remarks to the Author):

This work addresses the dynamic of the eukaryotic replication helicase which is a very interesting subject and promises to help understand how the replisome works in eukaryote. The authors use magnetic tweezers to study the enzyme with a dsDNA template in an occluded strand configuration. They record unwinding events and propose a model to explain their observations.

Although I find that this is a very serious work, as the authors I find the very slow and erratic helicase unwinding very strange. The helicase traces are really weird, the rate is very slow, the back-and-forth motion is extreme compared with all helicases traces that I have seen so far. Before looking for a model to explain the results, I would really double check that everything is working as it should. Furthermore, I am in trouble because the paper does not present simple demonstrative facts that you expect in single molecule work that might be very useful here.

1) In single molecule work you need to observe sparse events that spans ~10% of the recording time, this ensures that you mostly observe one enzyme and that the probability of having two enzymes working together is very low. Here there is no evidence that the helicase is actually working in a single molecule regime: in all the traces shown in the paper or SI, the unwinding is always going on and you neither see no unwinding signal nor a dissociation event showing that the dsDNA is rezipping fast upon the helicase detachment.

2) The traces show a very slow unwinding displayed in bp which is fine, but the reader must work to understand to what extension changes these traces correspond to and how they compare with experimental drifts. The occluding strand assay offers a low sensitivity typically 0.15 to 0.2 nm/bp depending on the force, overall the change of extension measured by the authors, when the helicase unwind the entire molecule is in the range of 500 nm (note that these numbers are not stated in the article) and this distance is traveled in 90 min while the detection resolution is typically 2 nm (corresponding to ~10 bp). This slow motion can easily be confused with experimental drifts which are difficult to avoid (the authors do not state if the sample is temperature regulated). The authors present filtered data but they do not provide any element demonstrating their signal stability over 90 min, they subtract the signal of reference beads to that of active beads, but we have no idea of the extent of this correction. This subtraction process is never perfect and will remove typically 90% of the drift but what is the amount of unresolved drift in comparison to the backward motions observed? The authors should provide a trace with no unwinding over 90 min.

3) The erratic signal with frequent backward motion is compatible with a helicase not very tight on its substrate. However the enzyme remains on this substrate for more than one hour. This is really strange, if the enzyme is not holding its substrate firmly, it is likely to detach frequently. Could it be that the signal is not a real helicase signal but the formation of a filament where the helicase will be covering the ssDNA and slowly melting the dsDNA? Can the authors show that the dsDNA in some occasion rezip very fast over all its extension ruling out a multi-enzyme binding?

4) The results are strange since the rate of the enzyme appears two orders of magnitude smaller than the in vivo replisome rate. In prokaryote, using bulk assays the replicative helicases of T7, T4 and E-coli have been shown to work poorly alone but very efficiently when coupled with their polymerase partner. In single molecule assays, these helicases were observed to be passive (or weakly active) meaning that their rate varies strongly with the amount of work that the helicase must develop to open dsDNA. When no help is provided to the helicase the enzyme unwinds dsDNA ~10 times slower than the replication rate. This factor 10 is easily explained in an Arrhenius context: a passive helicase rely on DNA spontaneous fluctuations opening, one helps the helicase by applying a force destabilizing the dsDNA the helicase unwind faster reaching the replication rate basically when the helicase has very little work to deliver. The DNA hybridization energy of ~3 kBT per base leads to the factor 10. Explaining a factor 100 will require a different mechanism.

Obviously one of the replisome partners might restore the correct helicase rate. For prokaryote, it has been shown that the when polymerase follows tightly the replicative helicase, transforming this helicase in an active helicase traveling at the replication rate. It is surprising that this work is not cited here. If the present result are correct one needs to identify the mechanism restoring the replication rate.

Within the present form, it is really hard for me to be convinced that the authors are really looking to single molecule events, I believe that the authors may have the necessary data to convince the reader that this is indeed the case and I encourage them to do so before we can explore their model.

Response to Reviewers

For clarity the reviewers original comments are included, the indented text are our replies, the bold indented text are the reviewers comments split into separate replies, and red text within quotes is text that has been added to the manuscript.

Within the manuscript itself, additions are highlighted in red. Finally, the following figures have been updated to reflect additional data taken and alterations from reviewer comments;

Updated Figs. 2a and 2b

Updated Fig. 3c

Updated Figs. 4a-f

Re-arranged Fig. 5

Updated Supplementary Table 1

Updated Supplementary Table 2

New Supplementary Fig. 2

New Supplementary Fig. 3

Updated New Supplementary Figs. 5b and 5c

New Supplementary Fig. 6

Updated Supplementary Fig. 7

Reviewer #1 (Remarks to the Author):

In this manuscript by Burnham et al. they investigate the unwinding dynamics of recombinant *Drosophila melanogaster* CMG helicase in vitro using a single-molecule magnetic tweezer DNA unwinding assay. In isolation under the conditions used in this assay, CMG exhibits a biased random walk, where forward translocation of the helicase is complicated by pausing and annealing events. The authors construct a kinetic model that suggests CMG might enter several distinct paused states during unwinding. The observed helicase activity is distinct from those observed previously, and the authors propose that XMG works via a 'Lazy Brownian Ratchet' model. The manuscript is timely and should be of interest to those working in DNA replication and repair although there are a number of concerns that need to be addressed.

We thank the reviewer for the excellent summary of our work and agree that the manuscript is timely and of interest to the field.

We have used all the reviewers comments and answered their questions to produce a much more precise and clearer manuscript that includes stronger evidence for our conclusions. The additions include more detail and clarifications in protocols, additional text to place our work in literature context, further analysis to demonstrate the precision of our assay, and altered text to clarify meaning.

We include a point by point reply below.

Major Comments

(1) One of the more surprising results of this manuscript is the reported rates for CMG are exceedingly low. The authors argue that high force should enable faster unwinding rates but there are a few problems with this statement. First, the authors claim that the unwinding rate increases

with force (as shown in Figure 2a) but the authors show no statistical tests that the distributions, which look fairly similar as plotted, are actually distinguishable. Second, the authors are applying high forces, at least from a biological context, and do not test that the unwinding rate may be faster at lower forces (for example 2 pN).

One of the more surprising results of this manuscript is the reported rates for CMG are exceedingly low

The rates observed are low compared to previously studied prokaryotic and bacteriophage hexameric helicases but not 'exceedingly' low. We did not communicate this clearly enough in our manuscript. To correct this we now discuss the values measured in context of previous literature, demonstrating the rates are commensurate with previous studies. We include the following in the paragraph 1, page 6;

“Conversely, CMG linear unwinding rates measured in this work, are similar to those of other AAA+ helicases. For example, purified SV40 Large T antigen shows relatively slow unwinding rates of $\sim 1.5 \text{ bps}^{-1}$ [(Klaue 2012, Berghuis, Köber et al. 2016)] and $\sim 3 \text{ bps}^{-1}$ in cell extract *in vitro* [(Yardimci, Wang et al. 2012)]. Furthermore, at high precision E1 shows non-monotonic behaviour, including pausing, forward and reverse movement [(Lee, Syed et al. 2014)] as does archaeal Mcm [(Schermerhorn, Tanner et al. 2016)]. The similarities in speed and dynamics between these homohexameric helicases and CMG may be a consequence of the common AAA+ ATPase motor and 3' to 5' translocation direction of superfamily 3 and 6 helicases.”

And we now also include a contextual reference in paragraph 2, page 13, that reads;

“Bulk biochemical studies of the eukaryotic minimal replisome have observed rates of 4.4 bps^{-1} [(Georgescu, Langston et al. 2014)] *in vitro*...”

Our clarifications and comparisons to literature demonstrate that both single-molecule studies of the eukaryotic model system SV40 large T antigen and bulk biochemical assays of CMG observe commensurate rates.

The authors argue that high force should enable faster unwinding rates but there are a few problems with this statement.

We stated in our original manuscript that a “...passive helicase awaits spontaneous thermal fluctuations to open the duplex, thus higher unwinding rates would be observed at larger forces [(Lohman and Bjornson 1996, Manosas, Xi et al. 2010)]”.

We intended this sentence to be a general statement about helicase literature as it has been thoroughly established (theoretically and experimentally) that for passive helicases the unwinding velocity does increase with larger forces applied to the DNA duplex. We have altered the above sentence (paragraph 3, page 6) to now read;

“Passive helicases, which await spontaneous thermal fluctuations to open the duplex, demonstrate higher unwinding rates when larger forces are applied to the DNA [(Lohman and Bjornson 1996, Manosas, Xi et al. 2010)]”

First, the authors claim that the unwinding rate increases with force (as shown in Figure 2a) but the authors show no statistical tests that the distributions, which look fairly similar as plotted, are actually distinguishable.

This was an oversight on our part and we now include Welch's t-test statistics for this data so readers can come to an informed understanding. We include the following in the main text in paragraph 3, page 6 with an extra supplementary figure 6 that includes all the comparison statistics:

“Naively calculating the mean unwinding rate over the trajectory with a linear fit (Fig. 2a) we observe a larger distribution in rate and an increase in mean from 30pN to 40pN ($t(86) = 5.1$, $p = 2.2 \times 10^{-6}$), Welch's t-test statistics given in supplementary fig. 6). This is consistent with the trends observed for other hexameric helicases [(Manosas, Xi et al. 2010, Ribbeck, Kaplan et al. 2010, Ribbeck and Saleh 2013)] and the expected force dependence from theory [(Manosas, Xi et al. 2010, Pincus, Chakrabarti et al. 2015)].”

Second, the authors are applying high forces, at least from a biological context, and do not test that the unwinding rate may be faster at lower forces (for example 2 pN).

In order to test the unwinding rate at lower forces ($< \sim 9$ pN) a single-stranded binding protein (SSB) must be used to prevent re-annealing of the unwound strands.

Our initial experiments were performed at 3.5 pN by following the same protocol as described in the manuscript except *E. coli*. SSB was included with the addition of ATP. Below, we include an example of unwinding trajectories for such an experiment. Many of the distinct trajectory features remain. The problem we faced was the careful interpretation of such data. It is known that SSB can slide on DNA [(Zhou, Kozlov et al. 2011, Bell, Liu et al. 2015)] and that bound RPA exchanges with free RPA in solution [(Gibb, Ye et al. 2014)], both of which alter the extension of DNA. While it is interesting to measure unwinding velocity at low forces, dis-entangling the activity observed due to inclusion of SSB versus the helicase only is non-trivial.

While we observe that unwinding is indeed faster at this lower force we cannot say with any certainty if this is due to lower applied force or the presence of SSB.

Example trajectories of DNA unwinding by CMG in the presence of *E. Coli*. SSB. Here the base pairs unwound are indicated by a decrease in number due to the shortening of DNA extension when dsDNA is converted to ssDNA at 3.5 pN. Some homogeneity remains and the mean unwinding rate has increased ~ 10 fold to 2.1 ± 0.3 bps $^{-1}$.

After further thought, and discussion with leaders in the field, we settled on a high force SSB-free assay as described in the manuscript. This removed any ambiguity in attribution of activity to CMG alone and allowed higher precision measurements due to the larger force applied.

We have successfully achieved a detailed understanding and description of the helicase alone. In future work we will address the interesting question of how single-stranded binding proteins affect helicase activity. This will allow measurements of velocity at physiologically relevant forces. However, studying and describing the helicase at these forces is beyond the scope of our current manuscript.

(2) The authors never clearly show what their spatial resolution is in this assay. SI Fig 2a shows a control experiment with CMG but no ATP that would answer this question but it is too zoomed out to see.

This was an oversight on our part and we have now included a significant amount of new information and data regarding the precision of the assay.

Paragraph 1, page 4, of section '**Results**' now includes the sentence;

“After subtraction of the position of reference microspheres stuck to the glass coverslip [(Lansdorp, Tabrizi et al. 2013, Dulin, Vilfan et al. 2015, Huhle, Klaue et al. 2015)] and low-pass filtering to 0.17 Hz, the displacement can be measured with 3.2 ± 1.2 nm standard deviation over 91 mins, (Supplementary Fig. 2).”

Paragraph 2, page 4 now reads;

“The precision of our instrument can be described by considering the Allan deviation [(Gibson, Leach et al. 2008, Czerwinski, Richardson et al. 2009)] of an enzyme free trajectory (Supplementary Fig. 2). Before low-pass filtering the Allan deviation shows precision on the order of a nanometre across the timescales of our experiments. The experimental method and apparatus are validated by demonstrating a homohexameric ring helicase, SV40 Large T antigen, unwinds in a manner that replicates previous work [(Klaue 2012, Berghuis, Köber et al. 2016)] (Supplementary Fig. 3). The protocol remains the same as for CMG except 110 nM monomer Large T antigen was first incubated in the presence of ATP at 37°C for 20 mins before being introduced into the sample chamber of tethered DNA, with no further chamber washes. In supplementary fig. 3a we demonstrate a typical example of SV40 Large T antigen unwinding a 1 kb DNA hairpin [(Hodeib, Raj et al. 2016)] and in supplementary fig. 3b unwinding the 3' flap DNA template used for CMG experiments. The hairpin was not used in the CMG experiments as in our hands no unwinding was observed. All values quoted in this work are mean \pm standard deviation.”

Paragraph 1, page 5 now reads;

“The measured trajectories are low-pass filtered to 0.17 Hz (Supplementary information section F and supplementary Fig. 4a for more details). Following the protocol outlined in Fig. 1d, we observe typical CMG single-molecule unwinding trajectories as shown in Fig. 1e, with no unwinding observed in the absence of CMG or ATP (Supplementary Fig. 5a).”

The new supplementary figure 2a includes example trajectories of both a DNA tethered microsphere and a coverslip stuck reference microsphere, at timescale zooms of 91 mins, 1000 secs, 100 secs, 10 secs, and 1 sec. This indicates the stability of, and noise present in, our apparatus, in detail.

Furthermore, we carefully analyse our non-enzymatic data to understand and demonstrate the stability of our system in a manner more reliable than observing the raw trajectories alone. We have calculated the Allan Deviation of our data [(Gibson, Leach et al. 2008, Czerwinski, Richardson et al. 2009)] which calculates half the averaged squared mean of neighbouring intervals to reveal how the deviation is dominated over various timescales, whether it be Gaussian white noise, or drift.

In supplementary figure 2b we include typical examples of Allan deviations calculated from a 91 minute non-enzymatic trajectory of a DNA tethered microsphere, and a stuck reference microsphere. Low pass filtering the data to improve signal to noise ratio, as performed in the manuscript, before calculating the Allan deviation results in a decrease in noise for timescales below 6 seconds (Fig Supplementary 2c).

The reader can now see the spatial resolution of our assay over all timescales to inform their understanding of the data presented. In particular one can see the timescale of the random walk observed for the helicase, 0.1 - 2 seconds (from FPT probability densities), is comparable to the region of minimum Allan deviation and so highest precision of our assay. The noise on this timescale is well below the magnitude of the effects seen in figure 1f, and is further reduced with low-pass filtering.

In summary the readers now have the information necessary to understand the precision of our experiments.

(3) Along the lines of Comment (2) it is unclear how the authors selected the passage interval. SI Fig 5 provides some insight but more would be helpful as this seems to be a critical part of the analysis.

We agree, this is critical to the analysis, and so we spent considerable effort developing an objective approach. ‘**Supplementary Section F: Data Analysis**’ clearly does not communicate our method, thus, in the interests of clarity we have included in paragraph 1, page 8;

“To prevent spurious detection of first passage times we choose the passage interval to be twice the standard deviation of an enzyme free trajectory (more details in Supplementary information section F) [(Dulin, Vilfan et al. 2015)].”

The standard deviation of the trajectory depends on the filter size used and so to clarify the passage interval selection we also must clarify the choice of filter size. To the best of our

knowledge there are no robust arguments in the magnetic tweezers literature rationalising trajectory filter size choice. Here, we use a quantitative approach with the aim of removing human estimation subjectivity.

To clarify filter choice size ‘**Supplementary section F: Data Analysis - Filter size**’ now reads;

“The raw z position data is low pass filtered using a 6 s moving mean average [(Little and Jones 2012)]. This 6 s timescale is selected from studying the enzyme-free motion of the DNA-tethered microsphere at 20 pN force. The sum of the squared difference between raw and filtered data points is plotted as a function of filter size (Supplementary Fig. 4a). At a certain filter size the benefit of further increases has diminishing returns and only serves to smooth the signal rather than increasing the signal to noise ratio. As such a filter size is chosen from the point at which the sum of differences between raw and filtered data reaches a plateau.”

With this clarification on filter size selection we can alter ‘**Supplementary section F: Data Analysis - Passage interval selection**’ to now read;

“The passage interval is chosen by determining twice the standard deviation of the signal when filtered with the filter size chosen above. For illustration we plot in Supplementary Fig. 4b how the standard deviation, σ , of an 80 s enzyme-free trajectory at 20 pN varies as a function of filter size. For the 6s filter size chosen above $\sigma = 10$ bp. To avoid spurious interval crossing and hence first passage times the passage interval, m , is set to $20 \text{ bp} = 2\sigma$ [(Dulin, Vilfan et al. 2015)]. The time taken for the trajectory to first pass each consecutive interval was measured until the end of the trajectory was reached. All first-passage times under the same conditions were concatenated and used in the fitting procedure. The parameters extracted are resistant to changes in passage interval, m , beyond the associated uncertainties, as the choice of interval for raw data analysis is reflected in the model fitting with equation 2.”

We feel this describes in detail the procedure used to determine the filter size employed, and the resulting passage interval choice.

(4) Are the kinetic parameters extracted from the FPT distribution resistant to changes in the passage interval?

Yes. A benefit of the FPT analysis is the objective nature. Our model describes the FPT distribution by equation 2, which includes the parameter, m , the number of base pairs in the passage interval. If the passage interval size is altered for the analysis of raw data then the parameter m is set to the same number when performing maximum likelihood estimation (MLE) with equation 2.

As an example we performed a Gillespie simulation to produce a trajectory of CMG unwinding that includes two pauses states. This is then analysed with a passage interval of 20 bp or 40 bp. The resulting FPT distributions are shown below.

FPT distributions of simulated CMG trajectory when analysed with a passage interval of 20 bp (left) and 40 bp (right). We plot the experimental FPT distribution overlaid by the best estimated model from MLE. The extracted parameters for both passage intervals remain the same, within the bootstrapped error estimates (100 bootstraps). The small discrepancies in value are due to the finite simulation time. Obviously the proportion of first-passage times attributed to each ‘state’, w_i , is different between the two interval sizes as the number of intervals dominated by a given state changes with interval size.

Obviously, the FPT probability distribution shifts to the right when the passage interval is increased, due to the increase in time to pass the larger interval, but, this is accommodated by the altered functional form of equation 2 due to the change in m .

One may imagine the robustness will decrease when the number of passage times sampled is low. This is true, and appears when rare events are detected, such as long time scale pauses. This is quantified explicitly in the manuscript by the standard deviation associated with each parameter extracted from the MLE. This standard deviation is calculated by performing MLE on 1000 bootstrapped with replacement sets of FPT data.

As explained in our manuscript one must be careful to choose an appropriately large enough passage interval to prevent mis-counting of interval crossings from noise, as described by Dulin et al. [(Dulin, Vilfan et al. 2015)]. For clarity we now include a reference to this work in the sections '**CMG behaves as a biased random walker with multiple, distinct, pauses**' and '**Supplementary section F: Data Analysis - Passage interval selection**'.

In summary, with a passage interval selected to prevent mis-counting, the parameters extracted are resistant to change. Any uncertainty is quantified and reported through the associated standard deviation of the parameters.

We have included this summary in '**Supplementary section F: Data Analysis - Passage interval selection**' which reads;

"The parameters extracted are resistant to changes in passage interval, m , beyond the associated uncertainties, as the choice of interval for raw data analysis is reflected in the model fitting with equation 2."

(5) Intuitively I would have predicted that increasing force would bias against reannealing yet SI table 1 shows that r_b actually increases with force at matched ATP concentrations. Can the authors interpret this result?

We do not currently have a reasoned explanation for this increase. There are only a handful of pieces of literature that discuss models of helicase motion from first principles [(Betterton and Jülicher 2005, Garai, Chowdhury et al. 2008, Pincus, Chakrabarti et al. 2015, Chakrabarti, Jarzynski et al. 2018)]. Interpretation of how these models fit with the data observed is difficult, as discussed by Manosas *et al.* (Manosas, Xi et al. 2010). The often cited work of (Betterton and Jülicher 2005) does describe the position of the helicase-fork midpoint with a biased random walk master equation but includes enzyme free DNA opening and closing rates. This opening and closing occurs on timescales ~ 5 orders of magnitude faster than our time resolution and so it is impossible to compare the parameters in their model directly with those we measure.

Intuitively one would expect DNA to be less likely to re-anneal with increasing force, as shown when force is included in the Betterton and Jülicher model [(Pincus, Chakrabarti et al. 2015)]. Likewise the rate of opening increases, as evidenced by force induced melting of dsDNA.

The kinetic rates of opening and closing are modified by the presence of the helicase. How so, is non-trivial. Indeed, this modification is governed by the coupling potential between helicase and fork structure. Without extensive further experimental and theoretical work it is not possible to determine this coupling potential and as yet we can only speculate as to why the increase in r_b may occur. Perhaps the lagging strand interaction and its disruption plays an important role.

In summary, this is a difficult question and will take a concerted effort, both experimentally and theoretically, to answer. The result does not change the discussion points and conclusions of the manuscript. We anticipate having this quantitative and interesting data out in literature will stimulate debate and effort around these questions.

(6) Could the authors add RPA or even SSB to rxns after loading helicases to determine whether the reverse translocation is slippage, disengagement or active annealing?

We break down our answer to this interesting question in several parts below. However, we first would like to point out that it is unnecessary to determine whether the reverse translocation is active annealing, slippage, or disengagement. This is because we demonstrate the observations can be fully recapitulated by the simpler model of a biased random walk without a separate mechanism for the reverse translocation. The reverse translocation is an inherent consequence and is governed by the relative probabilities of discrete forward and backward hopping on the DNA template.

We thank the author for alerting us to our omission of a clear description in this regard and have now included paragraph 4, page 10, which reads;

“Remarkably, we can describe the dynamic motion observed in Figs. 1e and f using a relatively simple stochastic model. Rather elegantly, the periods of reverse translocation are an inherent consequence of the model, without the need for an additional separate mechanism, due to the finite probability of the helicase to hop backwards at rate r_b .”

Perhaps a pertinent question would be whether or not the backwards hopping is ATP driven? In our proposed hypothesis of a Brownian ratchet the reverse translocation is a consequence of the random walk in a weakly bound state. All biased random walk motion is a consequence of ATP hydrolysis gating the energy from the surrounding thermal bath. In this sense it is ATP driven.

Could the authors add RPA or even SSB after landing the helicase?

We refer the reviewer to our answer to their question (1) where we discuss performing these experiments at low force.

Can we determine if the reverse translocation is ‘slippage’?

Slippage refers to the backwards motion of the helicase without ATP hydrolysis. We disregard this possibility as any slippage previously observed in literature is abrupt and linear, whereas the backwards motion of CMG is over prolonged timescales.

The inclusion of RPA or SSB would not necessarily preclude slippage events given these proteins are known to behave dynamically on DNA [(Zhou, Kozlov et al. 2011, Gibb, Ye et al. 2014, Bell, Liu et al. 2015)].

Can we determine if the reverse translocation is ‘disengagement’?

Our assay is constructed such that any unbound CMG is removed from the sample chamber during ATP addition. Thus the concentration dependent rate of helicase association [(Kang, Galal et al. 2012)] is negligible and should the helicase disengage it will unlikely re-associate. To clarify this we include the following in paragraph 1, page 4;

“CMG is drawn into the sample chamber in loading buffer, containing ATP γ S, and given time to bind the 3' flap. Next, 40-50 μ l (~4-5 flow chamber volumes) of temperature equilibrated CMG free running buffer, containing ATP is drawn into the sample chamber. Only previously loaded CMG remains.”

Any disengagement would result in near immediate re-annealing of the separated strands and give rise to motion appearing as slippage - which we have already discounted.

Can we determine if the reverse translocation is ‘active annealing’?

Not through adding SSB after helicase loading. How CMG interacts with protein roadblocks on the leading and lagging strand as it translocates is hotly debated so using a single-stranded binding protein to determine if the helicase can in some way ‘actively’ remove them would require much further work outside the scope of this manuscript before such a mechanism can be interrogated.

Although there are known annealing helicases that perform active (ATP hydrolysis dependent) annealing; they are rare. Single-molecule assays to investigate this are non-trivial and require

the formation of some form of unwound DNA that cannot re-anneal [(Burnham, Nijholt et al. 2017)]. This cannot be accomplished with our current assay and would be difficult with the reconstituted recombinant CMG we use because it requires a 3' to 5' flap for loading, preventing torsionally constrained assays.

In short, the use of SSB or RPA would not help to clarify if the reverse translocation is slippage, disengagement or active annealing, although we can already discount slippage and disengagement, as explained above. The model that we demonstrate describes our data has no requirement for additional mechanisms. Rather elegantly, the reverse translocation is an inherent feature of the biased random walk, parameterised simply by r_f and r_b .

(7) Could dissociation and re-association of CMG be responsible for any of the pausing observed? It seems possible, but is overlooked as an explanation.

There is no other CMG in the sample chamber so the concentration dependent rate of helicase association [(Kang, Galal et al. 2012)] is negligible. Thus, should the helicase disengage it will unlikely re-associate and rapid re-annealing would occur. Also, pausing must only occur with the helicase retained in place otherwise, again, rapid re-annealing would result with no pause observed. To clarify this we include the following in paragraph 1, page 4;

“CMG is drawn into the sample chamber in loading buffer, containing ATP γ S, and given time to bind the 3' flap. Next, 40-50 μ l (~4-5 flow chamber volumes) of temperature equilibrated CMG free running buffer, containing ATP is drawn into the sample chamber. Only previously loaded CMG remains.”

We also believe the stability of CMG means it must ‘thread’ onto the 3' to 5' flap and so if a pause were to be caused by dissociation the CMG would have to find the end of the unwound strand before re-unwinding the DNA. Again, in this scenario one would expect a fast, abrupt re-annealing of the DNA due to the lack of helicase barrier at the fork.

The pausing is thus not caused by dissociation and re-association as it would no longer be a pause due to the rapid re-annealing that would occur.

(8) Some of the conclusions drawn from the model parameters do not appear to be particularly rigorous (Fig. 4). The authors state “The probability of entering the medium and long pause states are constant as force varies but entrance into the biased random walk or short pause are affected (Fig. 4a and 4c).” I must be misreading the figure because “rw” in Fig. 4A looks flat to me.

As the probability parameters span several orders of magnitude we have placed them on a log scale so all can be compared easily. Log scales can disorient the reader so we also included the same data on linear scales in supplementary figure 7. However, clearly, we have not indicated this in the necessary points of the manuscript. Indeed, the paragraph the reviewer mentions contains some erroneous references. These have now been corrected. We have now included a reference to the supplementary figures that the reviewer will agree demonstrate the trends we describe.

Paragraph 3, page 12 now reads;

“The probability of entering the medium and long pause states are constant as force varies but entrance into the biased random walk or short pause are affected (Figs. 4a and 4c and Supplementary Figs. 7 a,c,e, and g). For decreasing ATP concentration the long timescale pause disappears, together with a reduction in the likelihood of entering the medium timescale pause. The exit rates from pauses also decrease with decreasing ATP concentration (Fig. 4f and Supplementary Fig. 7l and 7n). The rate of short pause exit, k_{short} , increases with force and ATP concentration (Fig. 4e, 4f and Supplementary Fig. 7k and 7l). Force and ATP regulate the entrance and exit from pauses demonstrating the intricate interplay between mechanics and ATP hydrolysis.”

(9) The authors have attempted to “establish a biophysically sensible description that explains how ATP hydrolysis is coupled to unwinding and gives rise to such dynamics.” Unfortunately, I am confused by the model. Why, for example, does decreasing ATP concentration lead to decreased pausing where in Fig 2 it was shown that increased ATP leads to increased rates of unwinding?

Fig. 2 shows increased rates of linear unwinding velocity at higher ATP concentration. The velocities here remove all the stochastic nature by applying a naive linear fit. As the reviewer indicates, through the first passage time analysis we see the probability of entering a pause decreases with decreasing ATP concentration. i.e. why does an increased probability of pausing cause an increase in linear unwinding rate?

Firstly, resolution of the confusion lies in considering both the probability of entering and the rate of exit from a pause. As ATP concentration decreases so too does the rate of exit. The pause lifetimes become longer lived, effectively slowing down the helicase. Equivalently increasing ATP concentration increases rates of pause exit, decreasing pause lifetime, effectively speeding up the helicase. This correlates with the increased ATP leading to increased linear rates of unwinding as noted by the reviewer.

Secondly, as the reviewer agrees, one should not put too much weighting on data that has arisen from a naive linear fit to such trajectories.

Minor Comments

Page 5, paragraph 1, last sentence

'Super family 3 viral replicative helicases show relatively slow unwinding rates when observed as a purified system [54] or in cell extract *in vitro* [55] and at high precision show unusual non-monotonic behaviour [29], more closely matching our data.'

The authors should highlight the close relationship between SF3 viral helicases and the SF6 helicases (MCM) as there may be a common basis for the observed slower unwinding rates. These are both AAA+ ATPases and translocate 3' to 5'. The faster bacterial and phage helicases belong to SF4 and SF5, and translocate 5' to 3'.

We thank the reviewer for this helpful suggestion. We have re-written paragraph 4, page 5, continuing onto page 6, paragraph 1, to read;

"Our observations are in stark contrast to the 5' to 3' translocating superfamily 4, RecA core, ring replicative helicases of bacteriophage [(Johnson, Bai et al. 2007, Lionnet, Spiering et al. 2007, Ribeck and Saleh 2013)] and *E. coli* [(Ribeck, Kaplan et al. 2010)]. These have predominantly exhibited monotonic, uniform, and unidirectional unwinding behaviour at rates of ~ 100 bps⁻¹, interspersed with pauses in activity in single-molecule assays [(Ribeck, Kaplan et al. 2010)]. These rates are, at most, within one order of magnitude of replication fork rates observed in live cells and bulk biochemical assays [(Werner 1968, McCarthy, Minner et al. 1976, Jeong, Levin et al. 2004, Rocha 2004, Stano, Jeong et al. 2005, Méchali 2010)]. Conversely, CMG linear unwinding rates measured in this work, are similar to those of other AAA+ helicases. For example, SV40 Large T-antigen shows relatively slow unwinding rates of ~ 1.5 bps⁻¹ [(Klaue 2012), (Berghuis, Köber et al. 2016)] when observed as a purified system or ~ 3 bps⁻¹ in cell extract *in vitro* [(Yardimci, Wang et al. 2012)]. Furthermore, at high precision E1 shows non-monotonic behaviour, including pausing, forward and reverse movement [(Lee, Syed et al. 2014)] as does archaeal Mcm [(Schermerhorn, Tanner et al. 2016)]. The similarities in speed and dynamics between these homohexameric helicases and CMG may be a consequence of the common AAA+ ATPase motor and 3' to 5' translocation direction of superfamily 3 and 6 helicases."

Figure 3c

The authors directly compare the first passage time distributions for two data sets at different forces (20pN and 45pN), but the [ATP] is also different so any differences might not be solely due to the force difference, as they claim [ATP] is also a factor that alters the unwinding rates in Fig 2b. This comparison should be done so that force is the only difference between experiments.

Thank you for pointing out this oversight. This has been corrected and we now show a comparison between 20 pN and 30 pN both at 4mM ATP. We have also corrected the figure caption and corresponding main body text.

Reviewer #2 (Remarks to the Author):

This manuscript describes a detailed analysis of the *Drosophila* CMG helicase using single-molecule approaches. The authors tether an artificial fork DNA to the surface of the slide at the dsDNA end and a magnetic bead to the non-translocating ssDNA strand of the replication fork. They monitor the extent of unwinding by measuring the increased distance between the slide and the bead as dsDNA is converted to (longer) ssDNA by the helicase. The authors use a modified version of a CMG assay in which they first bind the complex to the template in the presence of ATP γ S and then activate unwinding by addition of ATP. The authors find that the CMG complex alternates between unwinding of DNA, reannealing of DNA and multiple paused states. Mathematical modeling of data suggests that there are three different types of paused states. Using these models, the authors predict how various properties of the CMG will respond to changes in ATP concentration and force applied to the DNA.

We thank the reviewer for the nice summary of our work and appreciate the careful attention with which they read the manuscript. We have mitigated all the reviewers comments and so produced a much more precise and clearer manuscript that includes stronger evidence for our conclusions. The additions include further biological replicates of the data, additional experiments on an analogous hexameric helicase, precise statistical analysis of the data, and alterations to the text to clarify meaning.

We include a point by point reply below.

- The strength of these studies is their ability to explain the very poor CMG helicase activity previously observed in ensemble/bulk studies and the exclusion of very simple models of CMG function in which it never releases from the DNA (e.g. similar to the model proposed for E1 helicase function). It is noteworthy, however, that a different single-molecule assay for unwinding (that is less able to follow long movements of the helicase) also showed a distribution of pausing, forward movement and reverse movement for E1 (PNAS USA, 2014 Mar 4;111(9):E827-35). Thus, it remains unclear how different the CMG is relative to other helicases.

We agree that explaining the poor helicase activity observed in ensemble studies is a strength of our work. We thank the reviewer for pointing out the reference to E1. Having cited it in our manuscript, in hindsight we should have placed more emphasis on the observations made within and how it relates to the helicase under study in our work. Paragraph 4, page 5, continuing onto page 6, paragraph 1, now reads;

"Our observations are in stark contrast to the 5' to 3' translocating superfamily 4, RecA core, ring replicative helicases of bacteriophage [(Johnson, Bai et al. 2007, Lionnet, Spiering et al. 2007, Ribeck and Saleh 2013)] and *E. coli* [(Ribeck, Kaplan et al. 2010)]. These have predominantly exhibited monotonic, uniform, and unidirectional unwinding behaviour at rates of ~ 100 bps $^{-1}$, interspersed with pauses in activity in single-molecule assays [(Ribeck, Kaplan et al. 2010)]. These rates are, at most, within one order of magnitude of replication fork rates observed in live cells and bulk biochemical assays [(Werner 1968, McCarthy, Minner et al. 1976, Jeong, Levin et al. 2004, Rocha 2004, Stano, Jeong et al. 2005, Méchali 2010)]. Conversely, CMG linear unwinding rates measured in this work, are similar to those of other AAA+ helicases. For example, SV40 Large T-antigen shows relatively slow unwinding rates of ~ 1.5 bps $^{-1}$ [(Klaue 2012), (Berghuis, Köber et al. 2016)] when observed as a purified system or ~ 3 bps $^{-1}$ in cell extract *in vitro* [(Yardimci, Wang et al. 2012)]. Furthermore, at high precision E1 shows non-monotonic behaviour, including pausing, forward and reverse movement [(Lee, Syed et al. 2014)] as does archaeal Mcm [(Schermerhorn, Tanner et al. 2016)]. The similarities in speed and dynamics between these homo-hexameric helicases and CMG may be a consequence of the common AAA+ ATPase motor and 3' to 5' translocation direction of superfamily 3 and 6 helicases."

- Although not essential, it would improve the paper for these authors to look at another hexameric helicase (DnaB, E1 or SV40) using the same assay to provide a contrasting set of observations.

We agree this is not essential, being beyond the scope of this work. It may be detrimental to the narrative of our work, which provides observations of unwinding by CMG, details a model to describe the observations, and proposes a mechanism of operation - not hexameric

helicases in general. Providing a contrasting set of observations, while interesting, would cloud the clarity of the manuscript. It is a worthwhile endeavour, however, and over the coming years we will invest time and effort in completing this task.

Furthermore, questions remain as to whether the methodology would reveal the true nature of fast hexameric helicases due to the rate at which we can sample data. Due to the 10-100 fold speed decrease for CMG compared to, for example DnaB, we would need to sample at 10-100 times the rate in DnaB experiments to obtain the same level of sampling over the timescale of the fluctuations we aim to measure. This is not currently possible over the large fields of view necessary in our experiments. We are limited by camera technology.

We plan to prepare a second manuscript that discusses the similarities and differences between hexameric helicases and the methods used to analyse them. The two stories can be communicated effectively individually but not together.

The large number of pauses and reversals observed by the authors provide strong evidence that the eukaryotic helicase in this form is inefficient. These findings suggest that there will be additional factors that impact the CMG to allow it to attain the speeds of unwinding observed in vivo (e.g. DNA polymerases, Mcm10 or the Mrc1-Csm3-Tof1 complex).

We are pleased to hear the reviewer sees that we provide strong evidence and agree with their premise that the findings suggest additional factors will impact CMG kinetics. We alluded to this in our paper but have now included an extra Discussion section to highlight why CMG is inefficient and how we think this may be mitigated by additional factors. Please see new manuscript section '**Why does CMG unwind at a low linear velocity?**', page 13.

The authors suggest that this enzyme is an inefficient Brownian ratchet. Although some of the data fits this view, other data is less compelling. For example, it seems odd that there is only a minimal effect of the force on the rate of unwinding (Fig. 2). It would seem that pulling on the non-translocating strand would cause increased rates of breathing at the fork facilitating a Brownian ratchet mechanism of DNA unwinding. In contrast, going from 20 to 30 pN has no effect on unwinding rates and a further increase to 40 pN results in no change in many of the data points with a smaller number of points that show an increased rate. The authors merely summarize these data as showing that force increases unwinding rates but this is clearly not always the case. A discussion of why this would not be true between 20 to 30 pN is needed. It would improve the study to include biological replicates of some of the key experiments to show that the unusual response to force and ATP concentration is confirmed (see specific points below).

- We do suggest that CMG is a Brownian ratchet. Although we have been careful not to label it as inefficient as we believe this would require some measurement of the ability of the helicase to step forward per ATP hydrolysed. We label it as 'lazy' as it frequently enters long lived pause states.

It is unfortunate that the reviewer does not find all our data compelling. This may be due to poor communication of the subtleties on our part. Here we discuss these subtleties.

- Firstly, given the naive nature of a linear fit to the unwinding trajectories to achieve figure 2 one should not place too much weight on the conclusions drawn. We have added to paragraph 3, page 5 the following;

“Although one should be careful not to heavily weight inferences from such a naive fitting approach, this is approximately one to two orders of magnitude slower than the replication fork rates of ~10 - 50 bps⁻¹ observed in eukaryotic cells [(Raghuraman 2001, Anglana, Apiou et al. 2003, Conti, Saccà et al. 2007, Méchali 2010, Sekedat, Fenyö et al. 2010, Gispan, Carmi et al. 2017)].”

- It is not odd there is only a minimal effect of force on the rate of unwinding. Force independence of helicase unwinding rate has been demonstrated previously in literature [(Manosas, Xi et al. 2010, Ribbeck, Kaplan et al. 2010, Ribbeck and Saleh 2013)].

- Pulling on the non-translocating strand does cause increased breathing of the DNA at the fork. However, this would not facilitate a Brownian ratchet but rather alter the kinetics.

To understand why force-independent unwinding rates do not negate the possibility of a Brownian ratchet one must study the discussion of active versus passive helicases.

At the extremes of helicase mechanisms it can be said that a passive helicase only unwinds DNA by awaiting thermal fluctuations to open the fork, whereupon it translocates forward, blocking the re-annealing. At the opposite extreme, active helicases can disrupt base pairs ahead of the helicase within the double stranded region. The application of force destabilises base pairs ahead of the fork allowing DNA breathing to increase and the likelihood of a thermal fluctuation opening a base pair increasing.

A passive helicase then, likewise, has a greater probability of stepping forward and an overall increase in velocity as a function of increasing force. For an active helicase, disrupting the base pairs ahead, the destabilising force will have negligible effect on the kinetics thus the velocity can remain constant over a range of forces.

The small increase in mean unwinding rates as a function of force in figure 2 (which now has associated statistical tests displayed in supplementary figure 6 for the reader to decide on the sample descriptions) is indicative of a passive helicase. But, as we indicate in our manuscript analysing the data in this manner is misleading. If we consider the velocity calculated from the objective first-passage time analysis we see the unwinding velocity is constant with force (fig. 4c), indicative of an active helicase.

In summary the lack of force dependence of the velocity, rather than not being compelling of a Brownian ratchet, is in fact crucial to understanding whether the mechanism of action is active, i.e. disrupting base pairs ahead of the helicase.

The precise 'active' or 'passive' nature is not trivial to establish [(Manosas, Xi et al. 2010)] and does not impact on our conclusions drawn. We thus decide to leave this type of investigation for future work.

- Rather than simply summarising the data in figure 2 we now also include supplementary figure 6 which gives values for Welch's t-test and the associated p-values for comparisons between data sets. This allows the reader to draw informed conclusions regarding the data presented. Taking $p < 0.05$ as statistically significant (we have not written this in the manuscript as the choice of 0.05 is arbitrary and we feel readers should use the descriptive statistics we have provided and their experience to place the data in context) then there is a statistically significant difference between population samples.
- We have also conducted biological replicates of some of the key experiments, specifically repeats at each different force. This data is now included throughout the manuscript in the relevant figures and supplementary tables and figures. These replicates have not altered the observations in any meaningful way and do not affect the discussion or conclusions drawn.
- In discussion of why the increase would be true for 30 to 40 pN but not 20 to 30 pN we consider other work in the field where the trend, both experimentally and theoretically, has an exponential nature. Thus appearing constant at low forces while diverging at higher. We now include the following in paragraph 3, page 6, of the main text to clarify;

“Naively calculating the mean unwinding rate over the trajectory with a linear fit (Fig. 2a) we observe a larger distribution in rate and an increase in mean from 30pN to 40pN ($t(86) = 5.1$, $p = 2.2 \times 10^{-6}$, Welch's t-test statistics given in supplementary fig. 6). This is consistent with the trends observed for other hexameric helicases [(Manosas, Xi et al. 2010, Riebeck, Kaplan et al. 2010, Riebeck and Saleh 2013)] and the expected force dependence from theory [(Manosas, Xi et al. 2010, Pincus, Chakrabarti et al. 2015)].”

In summary, we feel we have carefully addressed the reviewer's comment, with the addition of supplementary information, and additional text in the main manuscript where appropriate. This, gratefully, has created a stronger manuscript.

In summary, this manuscript provides compelling evidence explaining the inefficiency of the CMG helicase. The authors also provide evidence restricting the type of mechanisms by which the CMG helicase functions, although, it remains unclear how different this mechanism is from other replicative DNA helicases due to the lack of comparative data.

We are pleased to hear the reviewer finds the evidence compelling. We agree it remains unclear how different this mechanism is from other replicative helicases. To build on our work in future we plan to use our objective and quantitative framework to establish if there are commonalities between replicative helicases. Indeed it is our aim to provide an analysis tool for the field such that direct comparisons can be made between data sets by ensuring the same analysis is performed.

Performing the experiments and measurements to include such data in this manuscript is beyond the scope of our work. It would require expression and purification of hexameric helicases spanning several organisms, and advancements in technology to ensure correct sampling resolution across the spectrum of helicase velocities. This considerable amount of work will span many years and many people, and we look forward to being part of it.

Specific points:

1. Why does reannealing occur when the DNA is under relatively high amounts of force. Do the authors believe that this is catalyzed by the CMG? If so, is this activity ATP-dependent?

A classic single-molecule stretching result is that with increasing force double-stranded DNA 'melts' to become single stranded at ~ 65pN. Upon reduction in force there is some hysteresis in annealing but below ~50pN the single-stranded DNA is once again dsDNA [(Smith, Cui et al. 1996)]. It is possible for the DNA to re-anneal from the fork backwards as we use forces below ~50 pN. To clarify this paragraph 1, page 4, now includes;

"We used forces between 20 and 40 pN that prevent re-annealing behind the helicase [(Dessinges, Lionnet et al. 2004)] (Supplementary information Section E - Prevention of annealing behind helicase at high force) and permit spontaneous re-annealing from the ss-ds DNA junction [(Smith, Cui et al. 1996)], ahead of the helicase."

We stated in the manuscript that "CMG exhibits...reverse motion...observed from a reduction in bp unwound due to annealing ahead of the helicase." And "[o]ur results showed that CMG translocates backwards over prolonged periods".

As we have not clearly communicated why the DNA anneals we have altered this main text and included further supplementary information to address the reviewer's comment. Paragraph 2, page 6, now reads;

"Unexpectedly, CMG exhibits not only unwinding and pause-like dynamics but also reverse motion (Fig. 1f). With annealing behind the helicase prevented at high forces (Supplementary information Section E - Prevention of annealing behind helicase at high force) spontaneous annealing can only take place from the ss-dsDNA junction. Thus the reduction in bp unwound indicates CMG has travelled backwards, allowing annealing ahead of the helicase. Reverse motion previously observed in ring helicases is either abrupt and monotonic and ascribed to 'slipping' [(Sun, Johnson et al. 2011, Lee, Syed et al. 2014, Schermerhorn, Tanner et al. 2016)], a motion that is considered not to require ATP hydrolysis, or is on the order of single base pairs [(Syed, Pandey et al. 2014)]. Our results showed that CMG translocates backwards over prolonged periods, neither abruptly nor over single base pairs, thus cannot be 'slipping'. In this work, rather than attribution to a separate mechanism, we chose to incorporate the reverse motion within a single unifying description, thus the emergence becomes inherent, as detailed in the next section."

Where we have included detailed information about the kinetic barrier formed by the helicase and force stretching in Supplementary section E that reads:

"Prevention of annealing behind helicase at high force

To re-anneal partially unwound DNA behind the helicase the leading, tracked strand, must overcome an activation energy to match the stretched, excluded, lagging strand. Following Dessinges et al.

[[Dessinges, Lionnet et al. 2004]] we can estimate the thermal fluctuation required to achieve this, $\Delta E_a = N.F.\partial L(F)$ where N is the helicase binding footprint, F is the applied force, and $\partial L(F)$ is the change in extension between CMG bound ssDNA and dsDNA. To calculate the change in extension of DNA within CMG to that of stretched ssDNA we note that the ssDNA within the central channel has a structural form similar to that of a single strand extracted from B-form dsDNA [[Abid Ali, Renault et al. 2016]]. For CMG with a DNA footprint of $N \approx 34$ nt, a change in DNA extension of a single nucleotide gap between dsDNA to ssDNA at $F = 20$ pN is $\partial(20\text{pN}) = 0.198$ nm; giving $\Delta E_a = 29k_B T$, reducing the probability of hybridisation by 14 orders of magnitude. Therefore, DNA does not re-anneal behind the helicase even at the lowest force (20 pN) used in our experiments.”

These alterations and inclusions serve to clarify the location at which annealing is able to occur. Specifically, DNA cannot re-anneal behind the helicase due to the presence of the helicase and the activation barrier at the forces we employ. And the forces are low enough such that any exposed single-stranded DNA in front of the helicase is able to anneal spontaneously from the fork backwards.

We do not believe that CMG catalyses this annealing in the sense of ATP driven re-winding activity. In the sense that the backwards motion of the helicase permits the annealing then yes CMG does catalyse the annealing. However, there is no separate mechanism for this annealing, CMG travels backwards due to the finite backwards hopping rate observed - the motion is an inherent consequence of a biased random walk. In one sense, as ATP gates the energy available from the thermal bath to produce a Brownian ratchet, then yes this is ATP-dependent.

We have now included the following paragraph 4, page 10, to emphasise this point, which reads;

“Remarkably, we can describe the dynamic motion observed in Figs. 1e and f using a relatively simple stochastic model. Rather elegantly, the periods of reverse translocation are an inherent consequence of the model, without the need for an additional separate mechanism, due to the finite probability of the helicase to hop backwards at rate r_b .”

And also alter paragraph 2, page 14 (continues to page 15) to read;

“To facilitate a Brownian ratchet we propose, in the weakly bound state, open conformer, CMG is able to diffuse along the DNA due to a weak free energy potential. This motion is unbiased and depicted in Fig. 5a. Upon ATP binding or hydrolysis CMG undergoes an allosteric structural change that alters the affinity for DNA, increasing the free energy potential of interaction with the DNA. In this state, thermal energy is not sufficient for CMG to overcome the energy barriers and so is fixed in position. The asymmetry in the potential, likely provided by helicase position and state causing variations in ATP binding or release kinetics [100], means that upon cycling between the two states CMG is more likely to be in the (n+1)th potential well representing the next base along; giving rise to a random walk with bias in the unwinding direction (Fig. 5b) [89]. Under this mechanism, we do not believe that CMG directly catalyses ATP hydrolysis dependent DNA annealing, but rather permits annealing upon reverse translocation. ATP hydrolysis gating and thermal noise [92] produces a biased random walk with inherent backward motion, allowing the DNA to anneal.”

2. The data in Fig. 2 does not strongly support the authors contention that there is both an ATP concentration and force dependent increase in rates (only the 40 pN data is different and that data shows extensive variability). Additional experimental replicates and more statistical analysis would be appropriate.

More details to answer these points are given in reply to the reviewers general comments. Here we summarise them for ease of reading.

As in response to the reviewer’s earlier general comment we now include additional experimental replicates and also supplementary figure 6 which describes more statistical analysis comparing the data sets. This analysis shows there is a statistically significant difference between the population samples as ATP and force increases. This increase is not linear but considering other work in the field where the trend, both experimentally and theoretically, has an exponential nature, we see our data fits with observations in literature that use the same linear unwinding rate analysis technique.

3. Given the central nature of the mathematical modelling to the manuscript the authors should provide more evidence of the confidence of the fits of the proposed model. This is particularly true of the three paused states. This would strengthen the manuscript and stimulate discussion in the community. For example, in the fit parameters shown in Figure 4c, why would the "diffusion coefficient" (in bp/s) peak at 30 pN, and then decrease at higher and lower force? Similarly in Figure 4d, there is no constant trend in ATP-dependence on the "diffusion coefficient". Do authors expect that there is a dependence of diffusion coefficients or some other process is responsible at that force and ATP concentration? These results are not consistent with the statement from the authors that "With increasing force the overall trend is an increased effective diffusion (Fig. 4c)..." There are similar concerns for the exit rates in response to force (Fig. 4e).

Given the central nature of the mathematical modelling to the manuscript the authors should provide more evidence of the confidence of the fits of the proposed model. This is particularly true of the three paused states. This would strengthen the manuscript and stimulate discussion in the community.

We agree that the mathematical modelling is central to the manuscript and evidence of the confidence of the fits strengthens the manuscript. As such, we took a rigorous approach to ensure objectivity in the fitting and to reporting the results. Most importantly the confidence of the fits are fully described by reporting the standard deviations for all parameters that are included in Supplementary Table 1. If the reader wishes to consider confidence intervals, or standard error of the mean, they can be readily calculated. Likewise the reader may calculate the respective p-values comparing all the data reported. This can inform the reader on the statistical details of our work but does not alter the conclusions drawn.

There are also details in the manuscript the reader may use to further understand the 'soundness' of the fits;

- We do not fit to histograms in the manuscript, the standard method, because changing bin size can drastically alter the resulting fits.
- We do not assume *a priori* a Gaussian or Gamma function for the distribution of passage times as often occurs in literature.
- We use maximum likelihood estimation on all passage times not fits to a binned version of the data.
- Using the Bayesian Information Criterion objectively reports which of the models we test describe the data best. For example, in Supplementary Fig 5d, we fit both unidirectional and random walk helicase models, and the calculated BIC is smaller for the random walk and so this unambiguously tells us the data is best fit by this model. Furthermore, the number of pauses to include in the model is also informed by calculating the BIC for different numbers of pauses.
- The standard deviation error bars on the experimental first passage time distributions are calculated from bootstrapping with replacement and the relative error indicated is indicative of the variability in the data at the given timescales. For example, the relative errors are larger at long passage times because long time scale events are rare and the bootstrapping with replacement creates variability in this region. The reader can consider these error bars to inform on the confidence of the data.

We believe the care we have taken to use the most objective and reliable methods, along with the reporting of all associated statistics fully describes the confidence of our model fitting.

For example, in the fit parameters shown in Figure 4c, why would the "diffusion coefficient" (in bp/s) peak at 30 pN, and then decrease at higher and lower force? Similarly in Figure 4d, there is no constant trend in ATP-dependence on the "diffusion coefficient". Do authors expect that there is a dependence of diffusion coefficients or some other process is responsible at that force and ATP concentration?

There are similar concerns for the exit rates in response to force (Fig. 4e).

Additional experimental replicates have altered the trend of the effective diffusion in Fig. 4c, which no longer demonstrates a peak at 30pN, however, the remaining parameter trends remain unchanged.

Whether or not there is a linear trend the question is difficult to answer. As we noted with reviewer 1 interpreting the handful of helicase models from literature in context of experiments is challenging [(Betterton and Jülicher 2005, Garai, Chowdhury et al. 2008, Manosas, Xi et al. 2010, Pincus, Chakrabarti et al. 2015, Chakrabarti, Jarzynski et al. 2018)]. Especially predicting how the experimentally measured parameters should vary as a function of force and ATP concentration. For example, the diffusion coefficient reported here cannot be compared to that from theoretical models as we have a finite time resolution. Also, no existing theoretical model includes the occurrence of pauses and so the behaviour cannot be predicted in this manner. We use the data we present to hypothesise the origin of the states and to support a given model.

In short we do not expect any particular type of trend for the diffusion coefficient or pause rates as we have no theoretical framework on which to base this. Kinetic parameters of replication machinery measured in bulk [(Kang, Galal et al. 2012)] and at the single-molecule level have been shown to reach a maximum or minimum at certain regions of force and ATP concentration [(Manosas, Spiering et al. 2012, Manosas, Spiering et al. 2012)]. One may imagine there is a point at which efficiency of unwinding is optimum and we may be touching on this area. The high diffusion at low ATP concentrations may be due to poor coupling of ATP hydrolysis to thermal energy gating in the Brownian ratchet mechanism.

Not wishing to overly speculate we have held back on suggestions for the meaning of each parameter variation when there is no work in literature to support it. Our manuscript successfully demonstrates an elegant model can explain the observation of interesting dynamics. Extracting the kinetic parameters allows us to start down the path of determining the origin of the states we observe. However, further work is required to fully understand our observations and by reporting them we look forward to pursuing these interesting questions in collaboration with the field at large.

These results are not consistent with the statement from the authors that “With increasing force the overall trend is an increased effective diffusion (Fig. 4c)...”

We have removed this paragraph as we no longer feel this is a technically correct comparison to the literature cited.

4. The authors propose that there are interactions between the non-translocating strand and the CMG (Fig. 5d – note, there is no call out for this part of the figure in the paper) but the evidence for this conclusion from this paper is limited. If the authors want to make this proposal (especially in the form of an illustration in a figure) they should discuss the evidence in support of it directly. Do the authors mean to suggest that this interaction is important for the distinction between binding to n and $n+1$ states illustrated in Figs. 5a and 5b?

We may have been a bit over eager in including this sub-figure due to the speculative nature of that discussion. We have corrected this by removing the sub-figure.

5. The data in Supplemental Table 1 indicates that r_b and r_f increase with lower ATP concentrations. How does this fit into the author’s model? Similarly, the authors should address why there is a significant peak in r_f and r_b at 30 pN relative to 20 pN and 40 pN. Why would this be?

As the reviewer notes r_b and r_f increase with ATP concentration, while the velocity remains constant. Equivalently, the diffusion of our helicase has increased (the sum of r_b and r_f) while the bias in directed motion remains constant. We can speculate that lower ATP makes the helicase more likely to enter the weakly bound open conformational state and so is able to diffuse more freely. Or, is not able to as efficiently gate the energy in the thermal bath, increasing diffusion. However, without further evidence we feel this would be too much speculation for inclusion in the manuscript. We hope in future to tackle this question in more detail by addressing the expected change in these variables from theory, but currently we report the values for completeness, without them altering the discussion or conclusion of our work.

As noted above additional experimental replicates have altered the trend of the effective diffusion in Fig. 4c, which no longer demonstrates a peak at 30pN.

The remaining parameter trends, however, remain unchanged. This is discussed in more detail in reply to comment 3 but in summary these trends are not out of the norm. While it may not be possible to currently explain this data we include it as a starting point for the field.

Other points:

1. The data discussed in the third full paragraph on page 9 do not match the corresponding data in Supplementary Table 1 (e.g. r_b is 115 in the table and the text gives $r_b = 109$). Since this difference changes v_{mean} substantially this should be clarified.

In the table we present data combined from multiple measurements at the same parameters. In the main text we show data from a single measurement. This is now clarified in the text.

Reviewer #3 (Remarks to the Author):

This work addresses the dynamic of the eukaryotic replication helicase which is a very interesting subject and promises to help understand how the replisome works in eukaryote. The authors use magnetic tweezers to study the enzyme with a dsDNA template in an occluded strand configuration. They record unwinding events and propose a model to explain their observations.

Although I find that this is a very serious work, as the authors I find the very slow and erratic helicase unwinding very strange.

We are glad the reviewer finds this a very serious work as we have taken a great deal of time and effort to interpret the results carefully and consistently.

We thank the reviewer for the input. We have significantly improved the manuscript to mitigate the comments, clarify our work, and strengthen the evidence. This has included additional experiments, placing our work in better context of the literature, and more precisely explaining our protocols and the consequences.

We include a point by point reply below.

The helicase traces are really weird, the rate is very slow, the back-and-forth motion is extreme compared with all helicases traces that I have seen so far.

On initial sight we agree that intuition would suggest the helicase traces are weird. However, on considering the literature we see this work sits well within the field. In our original manuscript we alluded to this in passing but due to this comment and on the recommendation of reviewer 1 we have now emphasised this. The key points now discussed in the manuscript are;

- The rates observed are compatible with those in literature for similar hexameric helicases.
 - Purified SV40 Large T-antigen *in vitro* ~ 1.5 bps⁻¹ [(Klaue 2012)(Berghuis, Köber et al. 2016)]
 - SV40 Large T-antigen in *Xenopus* Egg extract *in vitro* ~ 3 bps⁻¹ [(Yardimci, Wang et al. 2012)]
- The back-and-forth motion is not uncommon in literature
 - Purified E1 *in vitro* shows non-monotonic behaviour including pausing, forward and reverse motion [(Lee, Syed et al. 2014)]
 - Purified Archaeal Mcm *in vitro* exhibits very dynamic motion in a flow stretching assay [Schmerhorn2016]

We also note it has been suggested the propensity for backward motion is likely higher than previously thought [(Chakrabarti, Jarzynski et al. 2018)] and large diffusion has been observed for CMG-Mcm10 on dsDNA [(Wasserman, Schauer et al. 2018)].

This information has now been more extensively discussed within the manuscript as per comment 1, reviewer 1.

Furthermore, the literature on single-molecule unwinding by ring helicases is always with much faster enzymes. There is the possibility we observe larger back-and-forth motions as we are sampling the process at a large enough rate. Perhaps for enzymes that complete the unwinding in a matter of seconds, the process is effectively being under sampled and the back-and-forth motion is hidden.

We have included a description of this possibility in the manuscript at paragraph 5, page 12.

“It remains unclear whether this is a universal feature of ring helicases or peculiar to CMG. The relatively slow linear unwinding rates of CMG may allow sufficient sampling of the activity to uncover the behaviour, but when studying ring helicases acting at 100s bps⁻¹, such as DnaB [(Ribeck, Kaplan et al. 2010)], the activity may be under-sampled. Indeed, it has been suggested that backstepping may be universal feature of helicases [(Chakrabarti, Jarzynski et al. 2018)]. This is a topic for further study.”

Before looking for a model to explain the results, I would really double check that everything is working as it should. Furthermore, I am in trouble because the paper does not present simple demonstrative facts that you expect in single molecule work that might be very useful here.

We have now included additional experiments, analysis and protocol clarifications to demonstrate that we do observe single-molecule events.

1) In single molecule work you need to observe sparse events that spans ~10% of the recording time, this ensures that you mostly observe one enzyme and that the probability of having two enzymes working together is very low. Here there is no evidence that the helicase is actually working in a single molecule regime: in all the traces shown in the paper or SI, the unwinding is always going on and you neither see no unwinding signal nor a dissociation event showing that the dsDNA is rezipping fast upon the helicase detachment.

Given the very slow rate of activity, it would be practically impossible to observe sparse events that span 10% of the recording time. Indeed, this requirement is often neglected in literature [(Berghuis, Köber et al. 2016, Lewis, Spenkelnik et al. 2017)]. Here, we equivalently use sufficiently small concentrations of CMG such that only 10% of molecules show activity. Considering each single-molecule trajectory (active and non-active) being concatenated through time we do have sparse events spanning ~10% of the total recording time.

Additionally, we have miscommunicated our experiment protocol and this has caused some confusion. Once the CMG binding to DNA in the presence of ATP_γS stage is complete, it is followed by a 4-5 sample chamber volume wash with CMG free, ATP running buffer. In effect there is no free CMG in the sample chamber during unwinding, so it must be single-molecule activity.

We thank the reviewer for pointing out this lack of detail in our manuscript and believe future readers will appreciate the clarity. We have now made additions to our manuscript to demonstrate we are acting in the single-molecule regime.

Paragraph 1, page 4 now includes the text;

“Upon analysis ~10% of all single DNA molecules show activity, demonstrating it is unlikely more than a single helicase acts at once in our assays (supplementary information E - CMG unwinding assay).”

Where supplementary section E - ‘**CMG unwinding assay**’ now includes a description of why observing only 10% of molecules have activity leads to single molecule events.

“Upon analysis ~10% of all single DNA molecules show activity. Considering the Poisson distribution, we see the probability of a single DNA molecule being unwound by x helicases, $P_{\mu}(x)$, when the ratio of molecules with activity to total number of molecules is μ is;

$$P_{\mu}(x) = e^{-\mu} \frac{\mu^x}{x!}$$

The probability for any given single molecule DNA to be unwound by $x = 0, 1, 2,$ or 3 helicases is then;

$$P_{0.1}(0) = e^{-0.1} \frac{0.1^0}{0!} = e^{0.1} = 0.90$$

$$P_{0.1}(1) = e^{-0.1} \frac{0.1^1}{1!} = 0.1e^{0.1} = 0.090$$

$$P_{0.1}(2) = e^{-0.1} \frac{0.1^2}{2!} = \frac{1}{2}0.01e^{0.1} = 0.0045$$

$$P_{0.1}(3) = e^{-0.1} \frac{0.1^3}{3!} = \frac{1}{6}0.001e^{0.1} = 0.00015$$

Thus, the likelihood of a single DNA molecule being unwound by 2 or more helicases is 1%. Of the molecules that exhibit activity, the likelihood of ≥ 2 helicases being responsible is 10% ($P_{0.1}(\geq 2)/(P_{0.1}(1) + P_{0.1}(\geq 2)) = 0.01/(0.09 + 0.01) = 0.1$). Notably in our assay there is unlikely to be ≥ 2 helicases per DNA molecule due to the length of the 3' flap and pre-loading of CMG. This calculated value can thus be treated as an upper limit.”

Finally we include in paragraph 1, page 4, to clarify there is no free CMG in the chamber;

“CMG is drawn into the sample chamber in loading buffer, containing ATP_γS, and given time to bind the 3' flap. Next, 40-50 μl (~4-5 flow chamber volumes) of temperature equilibrated CMG free running buffer, containing ATP is drawn into the sample chamber. Only previously loaded CMG remains.”

2) The traces show a very slow unwinding displayed in bp which is fine, but the reader must work to understand to what extension changes these traces correspond to and how they compare with experimental drifts. The occulting strand assay offers a low sensitivity typically 0.15 to 0.2 nm/bp depending on the force, overall the change of extension measured by the authors, when the helicase unwind the entire molecule is in the range of 500 nm (note that these numbers are not stated in the article) and this distance is traveled in 90 min while the detection resolution is typically 2 nm (corresponding to ~10 bp). This slow motion can easily be confused with experimental drifts which are difficult to avoid (the authors do not state if the sample is temperature regulated).

We did not state explicitly the extension change numbers as we have provided Supplementary Figure 1b which shows the extension difference between dsDNA and ssDNA at the forces used in these experiments. However, for clarification we have now included the precise changes in extension for the forces used in our work in the caption of Supplementary Figure 1.

We ensure the unwinding is not attributed to drift by subtracting the position of surface-stuck reference microspheres. Importantly, we do not observe significant changes in DNA extension in the absence of either CMG or ATP in our assays (Sup Fig 5a).

We have also taken a great care to ensure any drift in our apparatus is minimised. While we do not have any active temperature stabilisation we did include two statements on this in our supplementary information.

“The system is always allowed to equilibrate to 23 °C before experiments begin.”

and

“The microscope was built on a vibration isolated optical table (Thorlabs, B7590 Nexus Breadboard, PFA51505 Active Isolation Frame) surrounded by a light tight isolation chamber for temperature and air flow stability.”

We have now expanded our statement on the temperature stability of our apparatus;

“All apparatus was left switched on continuously, or switched on several hours before experiments began and allowed to equilibrate to 23 °C, to minimise vertical drift in the experiment.”

The authors present filtered data but they do not provide any element demonstrating their signal stability over 90 min, they subtract the signal of reference beads to that of active beads, but we have no idea of the extent of this correction. This subtraction process is never perfect and will remove typically 90% of the drift but what is the amount of unresolved drift in comparison to the backward motions observed? The authors should provide a trace with no unwinding over 90 min.

We are a little confused by the reviewers comments here. We do indeed subtract the signal of reference beads to that of active beads, and the extent is described in the manuscript methods; we subtract the reference bead from the active bead. The subtraction of a reference microsphere in this manner is standard within the field and can lead to Angstrom resolution at ~1 Hz [(Lansdorp, Tabrizi et al. 2013, Dulin, Vilfan et al. 2015, Huhle, Klaue et al. 2015)].

Previously we had written in section F '**Data Analysis**':

“The resulting z position trajectories have the z position of the reference microsphere subtracted and set to the zero level of the pre-unwinding trajectory.”

To ensure further clarity it now reads;

“The resulting z position trajectories have the z position of the mean of at least two reference microsphere subtracted point for point and set to the zero level of the pre-unwinding trajectory.”

Any non-enzymatic noise due to movement of the coverslip (i.e. flow cell, stage, etc.) will be removed from the enzyme trajectory.

We wonder if there has been confusion over what constitutes a reference bead. While an ‘active bead’ is interpreted to be one that is tethered with DNA and undergoing enzymatic activity, perhaps the reviewer thought we are referring to a DNA tethered bead, not undergoing enzymatic activity, as the reference bead; rather than a bead stuck on glass. We have now clarified this in the main text by including the following in paragraph 1, page 4;

“After subtraction of the position of reference microspheres stuck to the glass coverslip [(Lansdorp, Tabrizi et al. 2013, Dulin, Vilfan et al. 2015, Huhle, Klaue et al. 2015)] and low pass filtering to 0.17 Hz, the displacement can be measured with 3.2 ± 1.2 nm standard deviation over 91 mins, (Supplementary Fig. 2).”

To further answer the reviewers comments we refer them to our reply to comment 2 of reviewer 1 which discusses in detail additions we have made to address this comment. In short, we have included text in the main manuscript body describing in detail the precision of our system and also an additional supplementary figure 2. The supplementary figure displays both a trace with no unwinding over 90 mins as requested and a detailed noise analysis of our trajectories.

Importantly one can see the timescale of the random walk observed for the helicase, 0.1 - 2 seconds, is comparable to the region of minimum Allan deviation and so highest precision of our assay. The noise on this timescale is well below the magnitude of the effects seen in figure 1f, and is further reduced with low-pass filtering.

The reader can now come to an informed understanding of the precision of our experiments.

3) The erratic signal with frequent backward motion is compatible with a helicase not very tight on its substrate. However the enzyme remains on this substrate for more than one hour. This is really strange, if the enzyme is not holding its substrate firmly, it is likely to detach frequently. Could it be that the signal is not a real helicase signal but the formation of a filament where the helicase will be covering the ssDNA and slowly melting the dsDNA? Can the authors show that the dsDNA in some occasion rezipts very fast over all its extension ruling out a multi-enzyme binding?

The erratic signal with frequent backward motion is compatible with a helicase not very tight on its substrate. However the enzyme remains on this substrate for more than one hour. This is really strange, if the enzyme is not holding its substrate firmly, it is likely to detach frequently.

Our model agrees that the frequent backward motion is compatible with a helicase having a weak interaction with the DNA substrate. If we were looking at a monomeric helicase then we would agree this timescale on substrate is surprising. However, due to the hexameric ring nature of replicative helicases even when in a weak substrate interaction mode the likelihood of detachment is low. In fact a conformational change in CMG would have to occur for the tracked strand to be excluded from the helical central channel that requires an additional co-factor, Mcm10 [(Wasserman, Schauer et al. 2018)].

We do not agree that this is overly strange. In single-molecule literature model eukaryotic hexameric helicases remain on the substrate for similar times. [(Klaue 2012, Yardimci, Loveland et al. 2012, Berghuis, Köber et al. 2016)]. And, indeed, the times are of the same order of magnitude as when we run SV40 large T antigen test experiments in our apparatus as shown in new supplementary figure 3.

In bulk biochemical assays stalled ring helicases disassociate from DNA substrates with a half-life of 15 to 36 minutes. In particular Mcm467 has a half-life of 33 minutes [(Nakano, Miyamoto-Matsubara et al. 2013)]. However, this referenced work uses short DNA substrates and omits Mcm235, GINS, and Cdc45, which are known to stabilise leading strand engagement [(Petojevic, Pesavento et al. 2015)].

In literature context it is reasonable to observe the helicase being attached for as long as it is.

Could it be that the signal is not a real helicase signal but the formation of a filament where the helicase will be covering the ssDNA and slowly melting the dsDNA? Can the authors show that the dsDNA in some occasion rezipts very fast over all its extension ruling out a multi-enzyme binding?

We again point out that we have miscommunicated our experiment protocol and we refer the reviewer to our reply to their comment 1. This discusses the additions we have made to clarify that only a single-molecule is in action, negating the possibility of filament formation.

Furthermore, filament formation should follow the classic shape of cooperative or non-cooperative growth and reach completion [(van der Heijden, Seidel et al. 2007, van Loenhout, van der Heijden et al. 2009)]. In the absence of protein continuous disassembly occurs. Our trajectories do not follow these behaviour.

For filament formation one would also expect activity on the majority of single DNA molecules not the 10% we observe. This is because the 'single-molecule' is now the DNA rather than the protein/enzyme and protein concentrations sufficient to cause filament formation mean that all DNA molecules will have some interaction that can be observed. The Poisson distribution must predict non-sparse events for filaments to form.

Finally, under similar conditions and concentrations no filament formation is observed in single-molecule fluorescence experiments [(Wasserman, Schauer et al. 2018)]

For these reasons we believe the activity is not due to filament formation.

4) The results are strange since the rate of the enzyme appears two orders of magnitude smaller than the in vivo replisome rate.

In prokaryote, using bulk assays the replicative helicases of T7, T4 and E-coli have been shown to work poorly alone but very efficiently when coupled with their polymerase partner. In single molecule assays, these helicases were observed to be passive (or weakly active) meaning that their rate varies strongly with the amount of work that the helicase must develop to open dsDNA. When no help is provided to the helicase the enzyme unwinds dsDNA ~10 times slower than the replication rate. This factor 10 is easily explained in an Arrhenius context: a passive helicase rely on DNA spontaneous fluctuations opening, one helps the helicase by applying a force destabilizing the dsDNA the helicase unwind faster reaching the replication rate basically when the helicase has very little work to deliver. The DNA hybridization energy of ~3 kBT per base leads to the factor 10.

Explaining a factor 100 will require a different mechanism.

Obviously one of the replisome partners might restore the correct helicase rate. For prokaryote, it has been shown that the when polymerase follows tightly the replicative helicase, transforming this helicase in an active helicase traveling at the replication rate. It is surprising that this work is not cited here. If the present result are correct one needs to identify the mechanism restoring the replication rate.

We address the points raised here in parts below.

The results are strange since the rate of the enzyme appears two orders of magnitude smaller than the in vivo replisome rate.

In prokaryote, using bulk assays the replicative helicases of T7, T4 and E-coli have been shown to work poorly alone but very efficiently when coupled with their polymerase partner. In single molecule assays, these helicases were observed to be passive (or weakly active) meaning that their rate varies strongly with the amount of work that the

helicase must develop to open dsDNA. When no help is provided to the helicase the enzyme unwinds dsDNA ~10 times slower than the replication rate. This factor 10 is easily explained in an Arrhenius context: a passive helicase rely on DNA spontaneous fluctuations opening, one helps the helicase by applying a force destabilizing the dsDNA the helicase unwind faster reaching the replication rate basically when the helicase has very little work to deliver. The DNA hybridization energy of ~3 kBT per base leads to the factor 10.

Explaining a factor 100 will require a different mechanism.

On initial viewing we would agree that the rate is 'strange' being so low. However, as we discuss in our reply to the first comments by the reviewer when compared to the detailed literature our work matches quite well. This has been included explicitly in the manuscript at paragraphs 4-5, page 5 (continues onto page 6), which reads;

"We observed that CMG unwinds DNA very slowly when compared to previous studies on other ring helicases [(Hodeib, Raj et al. 2017)]. Calculating the mean unwinding rate with a linear fit to the single-molecule unwinding trajectories, we observe rates between 0.10 ± 0.08 bps⁻¹ and 0.40 ± 0.48 bps⁻¹ over the range of ATP concentration and force applied in our experiments (Fig. 2). **Although one should be careful not to heavily weight inferences from such a naive fitting approach, this is approximately one to two orders of magnitude slower than the replication fork rates of ~10 - 50 bps⁻¹ observed in eukaryotic cells [(Raghuraman 2001, Anglana, Apiou et al. 2003, Conti, Saccà et al. 2007, Méchali 2010, Sekedat, Fenyö et al. 2010, Gispan, Carmi et al. 2017)]. The core replicative helicase does not attain the speeds necessary for timely genome duplication.**

Our observations are in stark contrast to **the 5' to 3' translocating superfamily 4, RecA core, ring replicative helicases of bacteriophage [(Johnson, Bai et al. 2007, Lionnet, Spiering et al. 2007, Ribeck and Saleh 2013)] and *E. coli* [(Ribeck, Kaplan et al. 2010)]. These have predominantly exhibited monotonic, uniform, and unidirectional unwinding behaviour at rates of ~100 bps⁻¹, interspersed with pauses in activity in single-molecule assays [(Ribeck, Kaplan et al. 2010)]. These rates are, at most, within one order of magnitude of replication fork rates observed in live cells and bulk biochemical assays [(Werner 1968, McCarthy, Minner et al. 1976, Jeong, Levin et al. 2004, Rocha 2004, Stano, Jeong et al. 2005, Méchali 2010)]. **Conversely, CMG linear unwinding rates measured in this work, are similar to those of other AAA+ helicases. For example, SV40 Large T-antigen shows relatively slow unwinding rates of ~1.5 bps⁻¹ [(Klaue 2012), (Berghuis, Köber et al. 2016)] when observed as a purified system or ~ 3 bps⁻¹ in cell extract *in vitro* [(Yardimci, Wang et al. 2012)]. Furthermore, at high precision E1 shows non-monotonic behaviour, including pausing, forward and reverse movement [(Lee, Syed et al. 2014)] as does archaeal Mcm [(Schermerhorn, Tanner et al. 2016)]. The similarities in speed and dynamics between these homohexameric helicases and CMG may be a consequence of the common AAA+ ATPase motor and 3' to 5' translocation direction of superfamily 3 and 6 helicases."****

The helicase rates we observe are 1-2 orders of magnitude slower than replication fork rates *in vivo*; the precise value depending on the exact literature read. Accounting for a 100 fold difference is not necessary in the context of the variability of *in vivo* rates found in literature, it is only necessary to explain a 10 fold difference.

There is a 3-29 fold increase in rates observed due to coupling of additional factors in prokaryote and bacteriophage replisomes [(Kim, Dallmann et al. 1996, Stano, Jeong et al. 2005, Manosas, Spiering et al. 2012)]. Eukaryotic replisomes can be sped up by an order of magnitude via additional factors [(Lewis, Spenkelink et al. 2017, Yeeles, Janska et al. 2017)]. This further demonstrates that the rates we observe are not unusual when compared to *in vivo* work, and the speed up is achievable with the factors discussed in literature.

We have now clarified the manuscript to show that an explanation of a factor of 10 speed up is required and that both the relatively slow speeds and the speed up required are compatible with current literature.

Paragraph 2, page 13, has now been included, and reads;

"**Collaborative coupling between helicase and polymerase that speeds up 'forks' by 3 to 30 fold has been observed in prokaryotic [(Kim, Dallmann et al. 1996)] and bacteriophage [(Stano, Jeong et al. 2005, Manosas, Spiering et al. 2012)] systems and single-molecule work has shown reduction in pausing [(Manosas, Spiering et al. 2012)]. Bulk biochemical studies of the eukaryotic minimal replisome have**

observed rates of 4.4 bps⁻¹ [(Georgescu, Langston et al. 2014)] *in vitro* and the inclusion of further replisome components increase this to 24 bps⁻¹ [(Yeeles, Janska et al. 2017)]. Single-molecule studies of yeast CMG within a minimal replisome have shown that replication with polymerase ϵ occurs at a rate of 5.4 bps⁻¹, the inclusion of Mcm10 increases this to 11.9 bps⁻¹, and the inclusion of the replication protection complex MTC further increases this to 21.1 bps⁻¹ [(Lewis, Spenkelink et al. 2017)]. Given eukaryotic replication machinery is much more tightly regulated compared to viral and bacterial counterparts, it may not be surprising that eukaryotes need factors beyond a coupled polymerase to achieve *in vivo* replication fork rates.”

Explaining a factor 100 will require a different mechanism.

We have now included a new section (**Why does CMG unwind at a low linear velocity?**) in the discussion highlighting the reviewer’s comments to place our work in context. We explicitly state how our work discovers reasons for the slow unwinding observed, and our hypothesis as to how eukaryotic replication components can alter the kinetics we observe to attain *in vivo* rates.

Obviously one of the replisome partners might restore the correct helicase rate. For prokaryote, it has been shown that the when polymerase follows tightly the replicative helicase, transforming this helicase in an active helicase traveling at the replication rate. It is surprising that this work is not cited here.

We have expanded our discussion on the collaborative coupling by including the section ‘**Why does CMG unwind at a low linear velocity?**’ discusses this collaborative coupling. We remedy our oversight by now citing, what we believe to be, the paper(s) suggested [(Stano, Jeong et al. 2005, Manosas, Spiering et al. 2012)]. And also expand the discussion to include the known coupling of eukaryotic replisome components. We go on to explore how the quantitative understanding we have brought can be used to hypothesise as to how the additional components can interact to produce the speed up.

If the present result are correct one needs to identify the mechanism restoring the replication rate.

We agree that now we have identified reasons for the slow unwinding rate the mechanism to restore the replication rate must be identified. However, given the substantial body of work that was necessary to construct this manuscript finding the mechanism is well beyond the scope of our work. Indeed, with the at least seven additional protein replication factors that contribute to replisome speed up in eukaryotes this question will require the dedication of many scientists over many years to establish and is a highly active area of research.

Some articles demonstrate a speed up that would account for the discrepancy we observe [(Georgescu, Langston et al. 2014, Lewis, Spenkelink et al. 2017, Yeeles, Janska et al. 2017)] but none provide a mechanism for this speed up, only speculation as to what the mechanism may be. We have gone one step further; demonstrating the reason part of the replisome is ‘slow’. Furthermore, we quantitatively demonstrate dynamic and kinetic parameters of the system that could be altered by the remaining components. We feel this is very impactful for the field and constitutes a deep and detailed scholarly work that the journal readership would value.

Within the present form, it is really hard for me to be convinced that the authors are really looking to single molecule events, I believe that the authors may have the necessary data to convince the reader that this is indeed the case and I encourage them to do so before we can explore their model.

To further allay the reviewers concerns in terms of the single-molecule nature, drift, noise correction, and speed of helicase, and we perform experiments using a eukaryotic model system, SV40 Large T-antigen (supplementary figure 7). This is a homohexameric replicative helicase that cooperates with eukaryotic host cell replication proteins.

We performed 2 types of experiments;

1. **Hairpin assay.** Here we use a 1 kb hairpin DNA construct where there is no leading strand flap, as per many studies in literature, summarised in the review Hodeib *et al.* [(Klaue 2012, Hodeib, Raj *et al.* 2016)]. In these types of experiments one expects the helicase to first unwind the DNA construct then allow the DNA to re-anneal in the wake of the helicase translocating on ssDNA.
2. **Linear assay.** Using the same 2.7 kb construct with 3' flap that we use for the CMG experiments [(Dessinges, Lionnet *et al.* 2004, Ribeck, Kaplan *et al.* 2010, Ribeck and Saleh 2013)].

The protocol was identical to the CMG studies except the following differences;

- SV40 Large T-antigen was heated in CMG running buffer (i.e. in the presence of ATP) at 37°C for 20 minutes.
- The helicase was immediately added to the sample chamber with no ATP_γS pre-loading step.
- Free Large T-antigen was not removed from the sample chamber with a sample chamber wash.

There are several important conclusions from the observations.

- We can recapitulate the hairpin unwinding assay results of both bacteriophage and prokaryotic replicative helicases, albeit at lower speeds [(Dessinges, Lionnet *et al.* 2004, Lionnet, Spiering *et al.* 2007, Ribeck, Kaplan *et al.* 2010, Ribeck and Saleh 2013)], and previous results for Large T-antigen unwinding a hairpin [(Klaue 2012)], including the speed of ~1.5bps⁻¹.
- We can recapitulate the linear unwinding assay results of prokaryotic and bacteriophage helicases [(Ribeck, Kaplan *et al.* 2010, Ribeck and Saleh 2013)].

Both these results demonstrate that;

- Our assay has sufficient precision, sensitivity, and stability for the signals we observe to be unwinding.
- Distinct unwinding and re-annealing 'peaks' were observed, indicative of the single-molecule nature. Thus, even with no sample chamber washes (which we do include in the CMG assays) to remove free large T antigen, it is possible to observe 'sparse' events.
- The characteristic trajectories we observe are not indicative of filament formation.
- The speed matches that in literature, thus our assay provides results consistent with previous work.
- Large T antigen is a hexameric helicase that is able to undergo ring opening without additional factors [(Yardimci, Wang *et al.* 2012)]. It is likely less stable in this sense than CMG. However, it is able to remain on the DNA unwinding substrate for a time similar to that which we observe for CMG. Thus, in context, it is not strange for CMG to remain on a substrate for the times we observe.

These additional experiments thus provide further strong evidence of the single-molecule nature of our work and the ability of our assays to capture unwinding of hexameric helicases.

To inform the reader of this validation paragraph 2, page 4 (and paragraph 1, page 5) now reads;

“The experimental method and apparatus are validated by demonstrating a homohexameric ring helicase, SV40 Large T antigen, unwinds in a manner that replicates previous work [(Klaue 2012, Berghuis, Köber *et al.* 2016)] (Supplementary Fig. 3). The protocol remains the same as for CMG except 110 nM monomer Large T antigen was first incubated in the presence of ATP at 37°C for 20 mins before being introduced into the sample chamber of tethered DNA, with no further chamber washes. In supplementary fig. 3a we demonstrate a typical example of SV40 Large T antigen unwinding a 1 kb DNA hairpin [(Hodeib, Raj *et al.* 2016)] and in supplementary fig. 3b unwinding the 3' flap DNA template used for CMG experiments.”

Also included in the main text, paragraph 2, page 4 is;

“The hairpin was not used in the CMG experiments as in our hands no unwinding was observed, likely as CMG requires a free 3' ssDNA tail to bind to the DNA construct”

This is because we were unable to observe any unwinding by CMG on a hairpin substrate. Obviously, absence of evidence is not evidence of absence but we believe purified recombinant CMG requires a 3' flap to bind on to the DNA substrate which is not possible in the hairpin assay.

Clearly we did not communicate some important details about our work precisely enough, leading to the reviewer's comments. The manuscript has been made stronger with the reviewers input and now includes a great deal more information, including new experiments. Given the infrequent events observed, the protocol of pre-loading CMG before removal of any free CMG, the non-filament forming nature of the trajectories, and the ability to recapitulate literature results for a second, eukaryotic model, hexameric helicase, we believe the evidence strongly asserts we are indeed looking at single-molecule events with high precision.

References

- Abid Ali, F., L. Renault, J. Gannon, H. L. Gahlon, A. Kotecha, J. C. Zhou, D. Rueda and A. Costa (2016). "Cryo-EM structures of the eukaryotic replicative helicase bound to a translocation substrate." *Nature Communications* 7: 10708.
- Anglana, M., F. Apiou, A. Bensimon and M. Debatisse (2003). "Dynamics of DNA replication in mammalian somatic cells: nucleotide pool modulates origin choice and interorigin spacing." *Cell* 114(3): 385-394.
- Bell, J. C., B. Liu and S. C. Kowalczykowski (2015). "Imaging and energetics of single SSB-ssDNA molecules reveal intramolecular condensation and insight into RecOR function." *Elife* 4: e08646.
- Berghuis, B. A., M. Köber, T. van Laar and N. H. Dekker (2016). "High-throughput, high-force probing of DNA-protein interactions with magnetic tweezers." *Methods* 105: 90-98.
- Betterton, M. D. and F. Jülicher (2005). "Opening of nucleic-acid double strands by helicases: Active versus passive opening." *Physical Review E* 71(1).
- Burnham, D. R., B. Nijholt, I. De Vlaminck, J. Quan, T. Yusufzai and C. Dekker (2017). "Annealing helicase HARP closes RPA-stabilized DNA bubbles non-processively." *Nucleic Acids Research* 45(8): 4687-4695.
- Chakrabarti, S., C. Jarzynski and T. D. (2018). "Processivity, velocity and universal characteristics of nucleic acid unwinding by helicases." *BioRxiv*.
- Conti, C., B. Saccà, J. Herrick, C. Lalou, Y. Pommier and A. Bensimon (2007). "Replication fork velocities at adjacent replication origins are coordinately modified during DNA replication in human cells." *Molecular Biology of the Cell* 18(8): 3059-3067.
- Czerwinski, F., A. C. Richardson and L. B. Oddershede (2009). "Quantifying noise in optical tweezers by Allan variance." *Optics express* 17(15): 13255-13269.
- Dessinges, M.-N., T. Lionnet, X. G. Xi, D. Bensimon and V. Croquette (2004). "Single-molecule assay reveals strand switching and enhanced processivity of UvrD." *Proceedings of the National Academy of Sciences of the United States of America* 101(17): 6439-6444.
- Dulin, D., Igor D. Vilfan, Bojk A. Berghuis, S. Hage, Dennis H. Bamford, Minna M. Poranen, M. Depken and Nynke H. Dekker (2015). "Elongation-Competent Pauses Govern the Fidelity of a Viral RNA-Dependent RNA Polymerase." *Cell Reports* 10(6): 983-992.
- Garai, A., D. Chowdhury and M. D. Betterton (2008). "Two-state model for helicase translocation and unwinding of nucleic acids." *Physical Review E* 77(6).
- Georgescu, R. E., L. Langston, N. Y. Yao, O. Yurieva, D. Zhang, J. Finkelstein, T. Agarwal and M. E. O'Donnell (2014). "Mechanism of asymmetric polymerase assembly at the eukaryotic replication fork." *Nature Structural & Molecular Biology* 21(8): 664-670.
- Gibb, B., L. F. Ye, S. C. Gergoudis, Y. Kwon, H. Niu, P. Sung and E. C. Greene (2014). "Concentration-dependent exchange of replication protein A on single-stranded DNA revealed by single-molecule imaging." *PLoS One* 9(2): e87922.
- Gibson, G. M., J. Leach, S. Keen, A. J. Wright and M. J. Padgett (2008). "Measuring the accuracy of particle position and force in optical tweezers using high-speed video microscopy." *Opt Express* 16(19): 14561-14570.
- Gispan, A., M. Carmi and N. Barkai (2017). "Model-based analysis of DNA replication profiles: predicting replication fork velocity and initiation rate by profiling free-cycling cells." *Genome Research* 27(2): 310-319.
- Hodeib, S., S. Raj, M. Manosas, W. Zhang, D. Bagchi, B. Ducos, J.-F. Allemand, D. Bensimon and V. Croquette (2016). "Single molecule studies of helicases with magnetic tweezers." *Methods* 105: 3-15.
- Hodeib, S., S. Raj, M. Manosas, W. Zhang, D. Bagchi, B. Ducos, F. Fiorini, J. Kanaan, H. Le Hir, J.-F. Allemand, D. Bensimon and V. Croquette (2017). "A mechanistic study of helicases with magnetic traps: Helicases Studied with Magnetic Tweezers." *Protein Science* 26(7): 1314-1336.
- Huhle, A., D. Klaue, H. Brutzer, P. Daldrop, S. Joo, O. Otto, U. F. Keyser and R. Seidel (2015). "Camera-based three-dimensional real-time particle tracking at kHz rates and Ångström accuracy." *Nature Communications* 6: 5885.

Jeong, Y.-J., M. K. Levin and S. S. Patel (2004). "The DNA-unwinding mechanism of the ring helicase of bacteriophage T7." *Proceedings of the National Academy of Sciences of the United States of America* 101(19): 7264-7269.

Johnson, D. S., L. Bai, B. Y. Smith, S. S. Patel and M. D. Wang (2007). "Single-Molecule Studies Reveal Dynamics of DNA Unwinding by the Ring-Shaped T7 Helicase." *Cell* 129(7): 1299-1309.

Kang, Y. H., W. C. Galal, A. Farina, I. Tappin and J. Hurwitz (2012). "Properties of the human Cdc45/Mcm2-7/GINS helicase complex and its action with DNA polymerase in rolling circle DNA synthesis." *Proceedings of the National Academy of Sciences* 109(16): 6042-6047.

Kim, S., H. G. Dallmann, C. S. McHenry and K. J. Mariani (1996). "Coupling of a replicative polymerase and helicase: a τ -DnaB interaction mediates rapid replication fork movement." *Cell* 84(4): 643-650.

Klaue, D. (2012). *DNA Unwinding by Helicases Investigated on the Single Molecule Level*. PhD, Technische Universität Dresden.

Lansdorp, B. M., S. J. Tabrizi, A. Dittmore and O. A. Saleh (2013). "A high-speed magnetic tweezer beyond 10,000 frames per second." *Review of Scientific Instruments* 84(4).

Lee, S.-J., S. Syed, E. J. Enemark, S. Schuck, A. Stenlund, T. Ha and L. Joshua-Tor (2014). "Dynamic look at DNA unwinding by a replicative helicase." *Proceedings of the National Academy of Sciences* 111(9): E827-E835.

Lewis, J. S., L. M. Spenkelink, G. D. Schauer, F. R. Hill, R. E. Georgescu, M. E. O'Donnell and A. M. van Oijen (2017). "Single-molecule visualization of *Saccharomyces cerevisiae* leading-strand synthesis reveals dynamic interaction between MTC and the replisome." *Proceedings of the National Academy of Sciences*: 201711291.

Lionnet, T., M. M. Spiering, S. J. Benkovic, D. Bensimon and V. Croquette (2007). "Real-time observation of bacteriophage T4 gp41 helicase reveals an unwinding mechanism." *Proceedings of the National Academy of Sciences* 104(50): 19790-19795.

Little, M. A. and N. S. Jones (2012). "Signal processing for molecular and cellular biological physics: an emerging field." *Philosophical Transactions of the Royal Society A: Mathematical, Physical and Engineering Sciences* 371(1984): 20110546-20110546.

Lohman, T. M. and K. P. Bjornson (1996). "Mechanisms of Helicase-Catalyzed DNA Unwinding." *Annual Review of Biochemistry* 65(1): 169-214.

Manosas, M., M. M. Spiering, F. Ding, D. Bensimon, J. F. Allemand, S. J. Benkovic and V. Croquette (2012). "Mechanism of strand displacement synthesis by DNA replicative polymerases." *Nucleic Acids Res* 40(13): 6174-6186.

Manosas, M., M. M. Spiering, F. Ding, V. Croquette and S. J. Benkovic (2012). "Collaborative coupling between polymerase and helicase for leading-strand synthesis." *Nucleic Acids Research* 40(13): 6187-6198.

Manosas, M., X. G. Xi, D. Bensimon and V. Croquette (2010). "Active and passive mechanisms of helicases." *Nucleic Acids Research* 38(16): 5518-5526.

McCarthy, D., C. Minner, H. Bernstein and C. Bernstein (1976). "DNA elongation rates and growing point distributions of wild-type phage T4 and a DNA-delay amber mutant." *Journal of molecular biology* 106(4): 963-981.

Méchali, M. (2010). "Eukaryotic DNA replication origins: many choices for appropriate answers." *Nature Reviews Molecular Cell Biology* 11(10): 728-738.

Nakano, T., M. Miyamoto-Matsubara, M. I. Shoukamy, A. M. Salem, S. P. Pack, Y. Ishimi and H. Ide (2013). "Translocation and stability of replicative DNA helicases upon encountering DNA-protein cross-links." *J Biol Chem* 288(7): 4649-4658.

Petojevic, T., J. J. Pesavento, A. Costa, J. Liang, Z. Wang, J. M. Berger and M. R. Botchan (2015). "Cdc45 (cell division cycle protein 45) guards the gate of the Eukaryote Replisome helicase stabilizing leading strand engagement." *Proc Natl Acad Sci U S A* 112(3): E249-258.

Pincus, David L., S. Chakrabarti and D. Thirumalai (2015). "Helicase Processivity and Not the Unwinding Velocity Exhibits Universal Increase with Force." *Biophysical Journal* 109(2): 220-230.

Raghuraman, M. K. (2001). "Replication Dynamics of the Yeast Genome." *Science* 294(5540): 115-121.

Ribeck, N., D. L. Kaplan, I. Bruck and O. A. Saleh (2010). "DnaB Helicase Activity Is Modulated by DNA Geometry and Force." *Biophysical Journal* 99(7): 2170-2179.

Ribeck, N. and O. A. Saleh (2013). "DNA Unwinding by Ring-Shaped T4 Helicase gp41 Is Hindered by Tension on the Occluded Strand." *PLoS ONE* 8(11): e79237.

Rocha, E. P. C. (2004). "The replication-related organization of bacterial genomes." *Microbiology* 150(6): 1609-1627.

Schermerhorn, K. M., N. Tanner, Z. Kelman and A. F. Gardner (2016). "High-temperature single-molecule kinetic analysis of thermophilic archaeal MCM helicases." *Nucleic Acids Research* 44(18): 8764-8771.

Sekedat, M. D., D. Fenyö, R. S. Rogers, A. J. Tackett, J. D. Aitchison and B. T. Chait (2010). "GINS motion reveals replication fork progression is remarkably uniform throughout the yeast genome." *Molecular Systems Biology* 6.

Smith, S. B., Y. Cui and C. Bustamante (1996). "Overstretching B-DNA: the elastic response of individual double-stranded and single-stranded DNA molecules." *Science* 271(5250): 795-799.

Stano, N. M., Y.-J. Jeong, I. Donmez, P. Tummalapalli, M. K. Levin and S. S. Patel (2005). "DNA synthesis provides the driving force to accelerate DNA unwinding by a helicase." *Nature* 435(7040): 370-373.

Sun, B., D. S. Johnson, G. Patel, B. Y. Smith, M. Pandey, S. S. Patel and M. D. Wang (2011). "ATP-induced helicase slippage reveals highly coordinated subunits." *Nature* 478(7367): 132-135.

Syed, S., M. Pandey, Smita S. Patel and T. Ha (2014). "Single-Molecule Fluorescence Reveals the Unwinding Stepping Mechanism of Replicative Helicase." *Cell Reports* 6(6): 1037-1045.

van der Heijden, T., R. Seidel, M. Modesti, R. Kanaar, C. Wyman and C. Dekker (2007). "Real-time assembly and disassembly of human RAD51 filaments on individual DNA molecules." *Nucleic Acids Res* 35(17): 5646-5657.

van Loenhout, M. T. J., T. van der Heijden, R. Kanaar, C. Wyman and C. Dekker (2009). "Dynamics of RecA filaments on single-stranded DNA." *Nucleic Acids Research* 37(12): 4089-4099.

Wasserman, M. R., G. D. Schauer, M. E. O'Donnell and S. Liu (2018). "Replisome preservation by a single-stranded DNA gate in the CMG helicase." *BioRxiv*.

Werner, R. (1968). "Distribution of growing points in DNA of bacteriophage T4." *Journal of Molecular Biology* 33(3): 679-692.

Yardimci, H., A. B. Loveland, A. M. van Oijen and J. C. Walter (2012). "Single-molecule analysis of DNA replication in *Xenopus* egg extracts." *Methods* 57(2): 179-186.

Yardimci, H., X. Wang, A. B. Loveland, I. Tappin, D. Z. Rudner, J. Hurwitz, A. M. van Oijen and J. C. Walter (2012). "Bypass of a protein barrier by a replicative DNA helicase." *Nature* 492(7428): 205-209.

Yeeles, J. T. P., A. Janska, A. Early and J. F. X. Diffley (2017). "How the Eukaryotic Replisome Achieves Rapid and Efficient DNA Replication." *Molecular Cell* 65(1): 105-116.

Zhou, R., A. G. Kozlov, R. Roy, J. Zhang, S. Korolev, T. M. Lohman and T. Ha (2011). "SSB functions as a sliding platform that migrates on DNA via reptation." *Cell* 146(2): 222-232.

REVIEWERS' COMMENTS:

Reviewer #1 (Remarks to the Author):

Burnham et al have significantly improved the manuscript entitled, 'The mechanism of DNA unwinding by the eukaryotic replicative helicase'. In general, they have removed irrelevant text and added and edited existing text. This has provided the manuscript with more focus and clarity and has more effectively put their results into the context of replicative helicases. They have gone to great lengths to validate their assay and data processing methods by replicating previous results other groups have generated for the SV40 Large T antigen helicase (Supp. Fig 3). The resolution and limits of their assay are now clear and easily understood thanks for additional text in the results and methods sections and a new figure (Supp. Fig 2). Statistical tests have now been applied to appropriate data sets and are visualized in Supp. Fig 6.

They have provided data in their reviewer response to explain why they did not include SSB or RPA in these experiments and why they must use a relatively high force during the unwinding assay. They have also clarified how their assay data and model can inform on the observed backwards motion of the helicase to distinguish between several causes of these motions.

I believe that the manuscript is timely and the data is now well supported. I therefore believe the manuscript is suitable for publication. My one comment that I think the reviewers should consider is that it still comes across like the authors have cherry-picked which trends to discuss to support their model while ignoring others. The manuscript would be strengthened by addressing these other trends more explicitly.

Reviewer #2 (Remarks to the Author):

The revised manuscript by Yardimci and colleagues addresses the mechanism of the CMG eukaryotic replicative DNA helicase using a single-molecule approach. The authors provide strong evidence that the mechanism of function of the isolated CMG complex shows, in addition to the expected forward movement, long pauses and reverse motions. The authors have done a nice job of improving the manuscript in response to the reviewer's comments. They have put their observation in context of other helicases more clearly. More importantly, they have clarified the basis of their conclusions to make the manuscript much more accessible to the non-expert. That being said, the modeling portion of the paper will still be difficult for many to understand. Nevertheless, the basic conclusion, that isolated CMGs do not move continuously on the DNA, is easily understood.

The importance of these data is that it provides a starting point to understanding what changes in CMG/replisome function occur in the context of a complete replisome. As indicated by the authors, an initial understanding of the enzyme on its own (which has been studied extensively in bulk assays) is critical as a starting point. I fully support publication of the current manuscript in Nature Communication.

Specific points:

1. The statement that "The rate of short pause exit, k_{short} , increases with force and ATP concentration (Fig. 4e, 4f and Supplementary Fig. 7k and 7l)..." still seems too broad given that this quantity goes down from 20-30 pN and up from 30-40 pN. The authors should describe this data more clearly.

Reviewer #3 (Remarks to the Author):

The authors have carefully answered my questions but I am still not at all convince by the helicase traces shown and even more by the analysis provided:

The authors have now explained how they calibrate their experiment and added some missing details that the single molecule community is eager to have, they have also added to the analysis of the SV40 helicase which I really appreciate. For this helicase they report two types of experiment: a first one using a hairpin substrate where the force is applied in an unzipping configuration and a second one in the unpeeling configuration that they have used for the CMG helicase. The signal from SV40 obtained in the unzipping configuration (Supp. Fig. 3a) is just "perfect," the signal displays first no helicase activity until an enzyme loads on the molecule and starts unwinding the hairpin, a second phase of the signal corresponds to the situation where the helicase has passed the apex and the hairpin reannealed with some delay between the helicase position and the re-hybridization fork position. Then the helicase has finished her travel. Although the speed is low (and quite similar to that of the CMG), it is really easy to detect no activity versus activity. The data of Supp. Fig. 3b is also not too bad since we see no activity in the beginning and a start of a burst at $t \sim 700$ s, but already there are some surprising facts: first we do not see the end of the burst and the rapid reannealing that we expect when the helicase detaches, and then we have two strange very fast opening events at $t \sim 900$ s and $t \sim 1200$ s. This would correspond to very fast unwinding events which are presumably not compatible with the SV40 helicase low rate. Now when we consider the CMG signals they were not obtained with the same experimental process: they are obtained using the unpeeling configuration which is OK, but the authors first load the CMG with ATPyS and then trigger the unwinding by flushing ATP after rinsing any unbound CMG. What is really strange is why stopping the recording after 90 min when the helicase has not yet finished unwinding. In single molecule conditions, ATP consumption is extremely small and the helicase velocity does not decrease in time after 24 hours, study of slow helicase (1 bp/s) have been done with long recordings, where unwinding bursts last more than one hour so that the beginning and the end of the signal is clearly seen. Thus all the traces analysed are peculiar since they are always going up, no helicase start nor unbinding are visible which really help identify single molecule traces. Using the simple process of having the buffer containing the helicase and ATP where the helicase loads randomly and unbinds and falls has revealed extremely simple and efficient providing nice bursts why the authors did not try this simple approach? The authors explain that the CMG helicase did not display any activity on the hairpin substrate (whereas SV40 did), this sounds extremely difficult to explain, this assay is typically six times more sensitive than the unpeeling configuration, having no signal here is a real issue what model can explain this?

Finally, using noisy traces with no indication of helicase unbinding, the authors fit a model with 9 parameters! Having done a lot of data analysis, I would not dare trying to publish such a model: there is a joke which is unfortunately true saying that with 5 parameters you can fit an elephant, nine are really a too much. To be more quantitative, the authors say that "Naively calculating the mean unwinding rate over the trajectory with a linear fit (Fig. 2a) we observe a larger distribution in linear unwinding rates and an increase in mean from 30pN to 40pN ($t(86) = 5.1$, $p = 2.2 \times 10^{-6}$, Welch's t-test statistics given in supplementary fig. 6). This is consistent with the trends observed for other hexameric helicases [58,59,69] and the expected force dependence from theory [69,70]." This statement is right but the authors consider only the helicase rate change between 30 and 39 pN to establish their statement, while they have measured a decrease in rate going from 20 pN to 30 pN which ought to contradict their statement completely (with a solid statistical test). In the same spirit, most helicases display Michaelis-Menten behaviour for their unwinding rate versus ATP concentration which is not the case here.

For me, the authors are far too confident in their helicase traces quality which, in my opinion, are not sufficient, I recommend having better traces and better statistics before publishing.

Point by point response to reviewer comments

Reviewer comments and questions are quoted below for clarity and we discuss each one in turn. Inset blue text are our responses. Quotes surround text of our manuscript and bold text within are new additions.

Detailed Replies to Reviewer #1 comments

Burnham et al have significantly improved the manuscript entitled, 'The mechanism of DNA unwinding by the eukaryotic replicative helicase'. In general, they have removed irrelevant text and added and edited existing text. This has provided the manuscript with more focus and clarity and has more effectively put their results into the context of replicative helicases. They have gone to great lengths to validate their assay and data processing methods by replicating previous results other groups have generated for the SV40 Large T antigen helicase (Supp. Fig 3). The resolution and limits of their assay are now clear and easily understood thanks for additional text in the results and methods sections and a new figure (Supp. Fig 2). Statistical tests have now been applied to appropriate data sets and are visualized in Supp. Fig 6.

They have provided data in their reviewer response to explain why they did not include SSB or RPA in these experiments and why they must use a relatively high force during the unwinding assay. They have also clarified how their assay data and model can inform on the observed backwards motion of the helicase to distinguish between several causes of these motions.

I believe that the manuscript is timely and the data is now well supported. I therefore believe the manuscript is suitable for publication.

We are pleased the reviewer agrees we have significantly improved the manuscript and they are happy with our revisions. We thank them for their help in refining our work ready for publication.

My one comment that I think the reviewers should consider is that it still comes across like the authors have cherry-picked which trends to discuss to support their model while ignoring others. The manuscript would be strengthened by addressing these other trends more explicitly.

This is a fair point and we have changed the manuscript text to now read differently in the following paragraphs;

"Naïvely calculating **the linear unwinding rate** (Fig. 2a) we observed a **wider** distribution and increase in mean from 30pN to 40pN, **yet a decrease in mean and narrower distribution from 20pN to 30pN** ($t(86) = 5.1$, $p = 2.2 \times 10^{-6}$ and $t(78) = 2.9$, $p = 0.004$, respectively. Welch's t-test statistics given in **Supplementary Figure 8**). **Despite the small decreases in mean between 20 pN and 30 pN, these results are broadly** consistent with the trends observed for other hexameric helicases [¹⁻³] and the expected force dependence from theory [^{3,4}]."

"CMG activity is an interplay between unwinding and pausing

The probability of entering medium and long pause states **is constant with** force but entrance into the biased random walk or short pause are affected (Figs. 4a and **Supplementary Figures 9 a,c,e, and g**). For decreasing ATP **concentration**, the long timescale pause disappears, together with a reduction in the likelihood of entering the medium timescale pause. The exit rate from a **medium** pause also decreases with decreasing ATP concentration (Fig. 4f and **Supplementary Figures 9l and 9n**). The rate of short pause exit, k_{short} , increases **with** ATP concentration, **while decreasing with force between 20 pN and 30 pN before increasing for 40 pN** (Fig. 4e, 4f and **Supplementary Figures 9k and 9l**). Force and ATP regulate the entrance and exit from pauses demonstrating the intricate interplay between mechanics and ATP hydrolysis."

In the manuscript we do refer to all the trends observed from which a relevant conclusion can be drawn and include the all the data so as not to 'hide' anything. For example, it is difficult to infer anything from the trends of the measured w/s (supplementary figure 9i and 9j), or the exit rates from the long timescale pause (supplementary figure 9o and 9p), and so we include them in the figure but not refer to them.

Detailed Replies to Reviewer #2 comments

The revised manuscript by Yardimci and colleagues addresses the mechanism of the CMG eukaryotic replicative DNA helicase using a single-molecule approach. The authors provide strong evidence that the mechanism of function of the isolated CMG complex shows, in addition to the expected forward movement, long pauses and reverse motions. The authors have done a nice job of improving the manuscript in response to the reviewer's comments. They have put their observation in context of other helicases more clearly. More importantly, they have clarified the basis of their conclusions to make the manuscript much more accessible to the non-expert. That being said, the modelling portion of the paper will still be difficult for many to understand. Nevertheless, the basic conclusion, that isolated CMGs do not move continuously on the DNA, is easily understood.

We are pleased to hear the reviewer thinks we have improved our manuscript, with their help, for which we thank them. And we agree that the modelling portion is more complex than for many single-molecule publications but is both necessary and more objective. In our citations we have included many relevant publications and books on this topic to assist the reader. We now also include references⁵⁻⁷ within the manuscript that give a broader background to the use of this type of modelling.

The importance of these data is that it provides a starting point to understanding what changes in CMG/replisome function occur in the context of a complete replisome. As indicated by the authors, an initial understanding of the enzyme on its own (which has been studied extensively in bulk assays) is critical as a starting point. I fully support publication of the current manuscript in Nature Communication.

We are delighted the author acknowledges the importance of this work in providing a foundation for future work on the replisome and that they fully support publication.

Specific points:

1. The statement that "The rate of short pause exit, k_{short} , increases with force and ATP concentration (Fig. 4e, 4f and Supplementary Fig. 7k and 7l)..." still seems too broad given that this quantity goes down from 20-30 pN and up from 30-40 pN. The authors should describe this data more clearly.

This is a fair point and we have changed this statement to read;

"The rate of short pause exit, k_{short} , increases **with** ATP concentration, **while decreasing with force between 20 pN and 30 pN before increasing for 40 pN**"

Detailed Replies to Reviewer #3 comments

As a general response to reviewer 3's comments we would like to note the majority can be addressed with reference to our manuscript and previous response to reviewer reports.

For example, many concerns are solved with the knowledge that eukaryotic helicase CMG loading onto and unloading from DNA is tightly regulated [⁸⁻¹⁰] and the ring is unable to open without the presence of additional factors [¹¹]. Conversely, other replicative helicases, including SV40 large T antigen, can load from a monomeric state to form a hexamer on DNA, and are to open the hexameric ring dynamically [¹²]. It is also important to note that unlike other replicative helicases CMG has a heterohexameric Mcm2-7 core with the addition of Cdc45 and GINS effectively locking the ring closed [¹³]. For these reasons, the DNA template must be of a particular construction, i.e. a linear duplex with a 3' free end, for CMG to thread onto. This inability to engage with non-free end DNA also likely prevents disengagement as the ring cannot open. This stability on DNA is critical for genome stability because eukaryotes must duplicate a large amount of DNA and replication cannot restart as helicase loading is inhibited outside G1 phase of the cell cycle.

To clarify the particular nature of CMG loading and disassembly we have added references [⁹] and [¹⁰] into the introduction which now reads;

"Eukaryotic replisome construction and disassembly is tightly regulated [^{14, 15}]; an AAA+ ATPase hetero-hexameric minichromosome maintenance complex (Mcm) 2-7 loads as a double hexamer on double-stranded DNA (dsDNA) [¹⁶] at origins of replication before additional firing factors aid formation of the active replicative helicase Cdc45/Mcm2-7/GINS (CMG) [^{8,14,17,18}]"

We have also added

“, preventing the opening of the ring without additional factors [¹¹]. Notably, other replicative helicases, including SV40 large T antigen, can load from a monomeric state and open the hexameric ring dynamically [^{12,19}].”

We now respond to each comment in turn below.

The authors have carefully answered my questions but I am still not at all convince by the helicase traces shown and even more by the analysis provided: The authors have now explained how they calibrate their experiment and added some missing details that the single molecule community is eager to have, they have also added to the analysis of the SV40 helicase which I really appreciate.

We are pleased the reviewer appreciates our additions which address both their previous and current concerns.

We will address each point in turn below to explain why the concerns have no merit and no additional work is needed.

...the SV40 helicase which I really appreciate. For this helicase they report two types of experiment: a first one using a hairpin substrate where the force is applied in an unzipping configuration and a second one in the unpeeling configuration that they have used for the CMG helicase.

This is correct.

The signal from SV40 obtained in the unzipping configuration (Supp. Fig. 3a) is just “perfect,” the signal displays first no helicase activity until an enzyme loads on the molecule and starts unwinding the hairpin, a second phase of the signal corresponds to the situation where the helicase has passed the apex and the hairpin reannealed with some delay between the helicase position and the re-hybridization fork position. Then the helicase has finished her travel. Although the speed is low (and quite similar to that of the CMG), it is really easy to detect no activity versus activity.

The reviewer describes our signal for homohexameric SV40 large T antigen on the hairpin DNA substrate as “perfect”. We agree it is a nice example of this assay. Importantly it does not make the other signals we observe any less “perfect”. The eukaryotic CMG helicase just does not conform to an idealised trajectory.

It is great to hear the reviewer is no longer concerned that CMG is “very slow” because we have demonstrated a second hexameric helicase travelling at similar speeds. We are also grateful that the reviewer agrees our assay demonstrates it is “really easy to detect no activity versus activity”.

In summary, we are pleased the reviewer unambiguously acknowledges we can recapitulate previous work and that our experimental apparatus has sufficient sensitivity to perform the measurements we describe.

The data of Supp. Fig. 3b is also not too bad since we see no activity in the beginning and a start of a burst at $t \sim 700$ s, but already there are some surprising facts: first we do not see the end of the burst and the rapid reannealing that we expect when the helicase detaches, and then we have two strange very fast opening events at $t \sim 900$ s and $t \sim 1200$ s.

Rather than “not too bad” we believe this data perfectly demonstrates that SV40 large T antigen, unwinding the same DNA substrate we use for CMG, behaves as previously studied hexameric helicases do. Explicitly;

- The end of the burst is not observed because the whole substrate had not unwound before the experiment ceased.
- The speed we observe for large T antigen matches previous literature [20,21].
- The features observed, that the reviewer points to at $t \sim 900$ and $t \sim 1200$ s, are also observed in literature across organisms, for example;
 - Large T antigen in Fig. 4 e in Berghuis, Köber et al. [21] demonstrates similar features.
 - *E. coli* DnaB in supplementary Fig. S3 of Ribeck et al. [2] demonstrates a variety of features.
 - Archaeal MCM in Fig. 3a in Schermerhorn et al. [22] again has similar features and more.
 - The viral monomeric helicase NS3 in Dumont et al. 2006 [23] displays similar features.
 Far from being surprising the features are ubiquitous in literature.

We demonstrated our additional work on SV40 large T antigen, in terms of speed and features, matches that already shown in literature. The old adage of a "typical result" shown in literature often being the best example obtained must be taken into consideration when expecting data the same as published work. In our work, we do not carefully select the data on display.

For clarity, we now include in the manuscript

"The method and apparatus are validated by demonstrating **that unwinding by** homohexameric ring helicase SV40 large T antigen replicates **the speed and features** of previous work [20,21] **and has features similar to other ring helicases** [2,22,23] (Supplementary Figure 3)."

This would correspond to very fast unwinding events which are presumably not compatible with the SV40 helicase low rate.

Helicases do not travel at a single, constant, rate.

Literature in the field is clear that unwinding is not monotonic or uniform for many other helicases. It is perfectly compatible for the rate to alter during the course of the experiment. Moreover, variations in speed or pausing are used extensively to ascertain details about molecular motor kinetics [24]. Frankly, it is naïve to think only a single rate would be observed in a stochastic environment and that none of the distribution of instantaneous rates would be at an extreme.

One of the important conclusions from our manuscript is that all features are an inherent consequence of the helicase motion - therefore there is a distribution of first-passage times, rather than a single repeating one. We then show the motion can be objectively described by a relatively simple model. One must use objective kinetic models rather than one's subjective eye as this can focus biasedly on certain features.

Now when we consider the CMG signals they were not obtained with the same experimental process: they are obtained using the unpeeling configuration which is OK, but the authors first load the CMG with ATP γ S and then trigger the unwinding by flushing ATP after rinsing any unbound CMG.

These experimental details are correct

What is really strange is why stopping the recording after 90 min when the helicase has not yet finished unwinding.

It is not "strange" to record for 90 minutes. Attempting to perform longer measurements is unreasonable. This is due to the following practical reasons.

1. While conducting the experiment, we cannot observe the unwinding trajectory of every DNA molecule simultaneously, we can observe a maximum of 4 in real-time. The remaining ~ 100 molecules are analysed after the experiment has finished using the saved images (as we explain in our methods section). Thus, we cannot know when a particular unwinding trace has reached completion in order to stop the experiment at the 'correct' time.
2. With helicases unwinding at hundreds of base pairs per second, such as *E. coli* DnaB or T7 gp4, the unwinding can be completed in seconds, resulting in perhaps 300 images of data to analyse. Unwinding at fractions of a base pair per second, as in the case of CMG, means that even after 90 mins (5400 secs) the DNA is not fully unwound and we collect over 300,000 images that are saved to a hard disk for later processing.

We must limit the time span of our experiment as even with 90 mins of data it takes approximately 3 days to analyse the images to obtain helicase trajectories; and further time to perform the final analysis.

For these reasons, 90 mins is a practical time limit on our experiment, not a strange place to stop.

In single molecule conditions, ATP consumption is extremely small and the helicase velocity does not decrease in time after 24 hours, study of slow helicase (1 bp/s) have been done with long recordings, where unwinding bursts last more than one hour so

that the beginning and the end of the signal is clearly seen. Thus all the traces analysed are peculiar since they are always going up, no helicase start nor unbinding are visible which really help identify single molecule traces.

As we discussed earlier, CMG is a heterohexamer locked shut with Cdc45 and GINS, and cannot open or disassemble without additional factors. This makes it highly unlikely to dissociate from the DNA, to conserve genome stability, and so why no “unbinding” is observed. Together with our gold standard of controls, no unwinding without ATP or CMG, we and the other reviewers are convinced that we are measuring DNA unwinding by single CMG molecules.

As per above we now include the following in the introduction to clarify

“Eukaryotic replisome **construction and disassembly** is tightly regulated [^{14,15}]; an AAA+ ATPase hetero-hexameric minichromosome maintenance complex (Mcm) 2-7 loads as a double hexamer on double-stranded DNA (dsDNA) [¹⁶] at origins of replication before additional firing factors aid formation of the active replicative helicase Cdc45/Mcm2-7/GINS (CMG) [^{8,14,17,18}].”

We have also added

“, **preventing the opening of the ring without additional factors** [¹¹]. **Notably, other replicative helicases, including SV40 large T antigen, can load from a monomeric state and open the hexameric ring dynamically** [^{12,19}].”

We do have trajectories that demonstrate the start of helicase unwinding but it predominantly starts during ATP addition. As standard in the field, tracking of the motor protein position is lost during this addition due to flow in the sample chamber, so we cannot display it for all traces. We now include in supplementary Fig. 7 an example trajectory demonstrating that after ATP addition there is a period of no unwinding before the helicase begins unwinding. This figure is reference in the manuscript as;

“Following the protocol outlined in Fig. 1d, we observe typical CMG single-molecule unwinding trajectories as shown in Fig. 1e, with no unwinding observed in the absence of CMG or ATP (Supplementary Figures 6a and 7).”

and shown below.

Using the simple process of having the buffer containing the helicase and ATP where the helicase loads randomly and unbinds and falls has revealed extremely simple and efficient providing nice bursts why the authors did not try this simple approach?

Supplementary Fig. 7 | A plot showing the start of CMG unwinding a linear DNA substrate with 3' flap after addition of ATP before time zero. Unwinding predominantly starts during ATP addition. As standard in the field, tracking of the motor protein position is lost during this addition due to flow in the sample chamber, so we cannot display it for all traces. However, we display here an example trajectory, at 20 pN and 4mM ATP, demonstrating that after ATP addition there is a period of no unwinding before the helicase begins unwinding.

Firstly, as carefully explained in our response to the reviewer's first report, we perform loading of CMG with ATP γ S and then remove free CMG from the sample chamber, before unwinding commences with addition of ATP. This was to remove any ambiguity as to whether single-molecules are performing the unwinding because in this case only single CMG molecules are loaded and free CMG has been removed. In the manuscript it reads;

“CMG is drawn into the sample chamber in loading buffer, containing ATP γ S, and given time to bind the 3' flap. Next ~4-5 chamber volumes of temperature-equilibrated CMG-free running buffer, containing ATP, is drawn **through** the chamber. This ensures only previously loaded CMG remains in the sample chamber, preventing multiple helicases acting on single DNA substrates.”

From this statement it is clear we take a more careful approach to guarantee single-molecule trajectories, by pre-loading with ATPγS.

The authors explain that the CMG helicase did not display any activity on the hairpin substrate (whereas SV40 did), this sounds extremely difficult to explain, this assay is typically six times more sensitive than the unpeeling configuration, having no signal here is a real issue what model can explain this?

We can observe SV40 large T-antigen unwinding both hairpin and linear substrates but cannot observe CMG unwinding a hairpin. The reviewer states “this sounds extremely difficult to explain”. On the contrary, there is a clear explanation for this which we described in our revision. The reviewer misunderstands the reason for not seeing a signal or activity. It has nothing to do with the sensitivity of the assay, but everything to do with the nature of the helicase loading mechanism and DNA template geometry.

We have added the following text in the supplementary notes and the following figure as supplementary Fig. 4.

“We can observe SV40 large T-antigen unwinding both hairpin and linear substrates but cannot observe CMG unwinding a hairpin. This is not due to the sensitivity of the assay, but stems from the nature of the helicase loading mechanism and DNA template geometry.

SV40 large T antigen can, demonstrably, load onto DNA without a free end and onto origin DNA without additional factors [12]. This is likely due to the homohexameric construction and simpler DNA loading mechanism for this helicase, for example, the binding of monomers to DNA to form the hexamer (Supplementary Figure 4, top panels) [19]. CMG loading is much more tightly regulated [9], and pre-formed CMG complex cannot load onto DNA without a free 3' end unless Mcm10 is present [11]. Indeed, bulk biochemical assays are performed on fork substrates or substrates with a 3' free end to allow loading [25]. We believe that in our assay we can observe unwinding on the linear 2.7 kb duplex DNA substrate because the closed CMG ring can thread onto the 3' free end (Supplementary Figure 4, bottom right panel) as is observed in structural work [26]. We do not observe unwinding when using a hairpin because it has no free end for CMG to thread onto, thus preventing loading and subsequently unwinding (Supplementary Figure 4, bottom left panel). Without the tightly regulated loading pathway and associated additional factors CMG is not loaded onto DNA without a free end.”

Supplementary Fig. 4 | Differences between SV40 large T antigen and CMG loading on specific DNA constructs. Top panels) SV40 large T antigen can bind to and form a hexamer on both a hairpin and linear DNA template, where a 3' flap is and is not available, respectively. Bottom panels) Due to the tightly regulated process of eukaryotic helicase CMG loading, Cdc45 and GINS effectively close the ring structure of the heterohexameric Mcm2-7. This prevents CMG binding around a hairpin strand as there is no free 3' end but it can ‘thread’ onto a free 3' end contained within the linear substrate.

We explicitly stated that we don't observe unwinding on the hairpin with CMG because CMG likely needs a 3' free end to thread on to. We have now altered this sentence to also include a reference to the supplementary information. It now reads;

“The hairpin was not used in CMG experiments as in our hands no unwinding was observed, likely as CMG requires a free 3' ssDNA tail to bind the DNA construct [11] (Supplementary Figure 4 and Supplementary Note 3).”

Finally, using noisy traces with no indication of helicase unbinding, the authors fit a model with 9 parameters! Having done a lot of data analysis, I would not dare trying to publish such a model: there is a joke which is unfortunately true saying that with 5 parameters you can fit an elephant, nine are really a too much.

Firstly, the reviewer is no doubt aware that all data, here in particular time series traces of objects undergoing Brownian motion, is noisy. It is the magnitude and colour of that noise that varies.

On recommendation from the first review round we included a state of the art noise analysis that demonstrates both the magnitude and colour of the noise. We have a sufficiently small enough noise level to observe what we claim. Indeed, neither this reviewer nor the others have not raised any further comments regarding this having seen our additional analysis.

Secondly, we do not use these traces to fit the model. The reviewer seems to have misinterpreted our analysis. The traces are processed to extract first-passage times, to which we fit the model. Even if the traces were “noisy” (for example the variance was twice as large) the objectively calculated passage interval (supplementary Fig. 4) will be calculated to be equally larger, thus maintaining the robustness of our assay. We demonstrated such robustness for Reviewer 1 in our previous replies to comments in section “(4) Are the kinetic parameters extracted from the FPT distribution resistant to changes in the passage interval?”, which they have not questioned.

With regard to our model we have several points to make

As authors, we have performed a great deal of analysis also and have gone to great lengths to perform this analysis carefully and objectively.

We believe the joke the reviewer is referring to is a quote attributed to John von Neumann by Enrico Fermi in the 1950s as told by Freeman Dyson [27]. The greater context of this quote was in a conversation discussing the merits of Dyson's calculations exploring the strong force as measured by Fermi. In the same article, Dyson states Fermi told him “There are two ways of doing calculations in theoretical physics...One way, and this is the way I prefer, is to have a clear physical picture of the process that you are calculating. The other way is to have a precise and self-consistent mathematical formalism. You have neither.” Dyson had included four arbitrary “cut-off” parameters in their theoretical work; this stimulated Fermi's recollection of von Neumann's infamous elephant comment. The key aspect here is that in our model we **do** have a clear physical picture of the process and a self-consistent mathematical formalism - we include no arbitrary parameters. Quotes taken out of context can be misleading.

Fortunately, we do not have to rely on jokes to decide on our approach and demonstrate the scientific merits of our model. We can take advantage of advances in knowledge since the 1950s and use statistical methods to ensure the validity of our work. Placing an arbitrary value of 5 on the maximum number of parameters in a model is unscientific, subjective, and unreasonable. A quadratic equation uses 3 parameters ($ax^2 + bx + c$) and a simple double exponential decay with background offset has 5 parameters to fit ($ae^{-t/\tau} + be^{-t/\tau'} + c$), and these are both ubiquitous in nature. Rather than submit to an arbitrary rule of thumb, we have used quantitative statistics to take the choice of model out of our subjective hands. We also report the uncertainty of all our parameters are reported in the manuscript to allow each individual reader to decide on the merits of the fits, and to interpret the conclusions. If the reviewer was truly concerned they might have commented on the magnitude of the uncertainty for each parameter, the mathematical formalism, or questioned the physics of our model. They have done none of these.

The reasons our model is not only sufficient but more powerful and reliable than usual is as follows;

1. We use the Bayesian Information Criterion (BIC) [7] to tell us which of the potential models best fits the data. The Bayesian Information Criterion is relatively simple to use. When fitting several models, the one with the lowest calculated BIC is the model that best describes the data. It is appropriate to use here as it penalises for model complexity and so prefers simpler models with fewer parameters. For example, in supplementary Fig. 5d we demonstrate that fitting our data with a simpler unidirectional model with a single pause fits worse than the model we propose. This can first be observed visually from the fit overlaid on the histogram. However, we do not rely on such subjective assessment. The BIC objectively tells us that our proposed model fits better than a unidirectional model (Supplementary Figure 5d). Furthermore, for this particular case, the BIC has told us that the unidirectional model fits best with 1 pause and our proposed model fits best with 3 pauses. So, despite the BIC penalising our preferred model for using more pauses and hence being more complex, it still outperforms the simpler unidirectional one pause model.
2. The standard procedure in much single-molecule work is to histogram the data regarding dwell times or pauses (first-passage time equivalents) and to then fit a Gaussian model *a priori*. We do not fit to the histogram as it is well known that alterations of the bin size can lead to different fits and thus kinetic parameters. We start from a biophysical model, make no prior assumptions regarding the functional form of the fit, and objectively eliminate different models and include the requisite number of pauses.
3. Furthermore, similar mathematical fitting models, used in similar experiments, already published in literature confidently use up to 15 parameters [28-31], and when trying to understand complex kinetic pathways far greater than 5 parameters are required [5,32,33]. Careful fitting of multi parameter models is established practice in certain fields with a robust basis [6,34]. Furthermore, we note that for certain values of ATP concentration and force our model uses only 7 parameters, which is objectively discovered as described.

We have explained these points throughout the manuscript. However, to clarify the details, we add reference ⁵ in the sentence that reads;

"Instead we employ first-passage time (FPT) analysis [35], extended from dwell time distributions [5,28,35-37], which also negates the analysis problems described."

which describes fitting with models that have 18 free parameters.

We have now also added the following to the methods section;

"The model first-passage time probability distribution (eq. 2) was used to find maximum likelihood estimators for the model parameters. The maximum likelihood estimation is performed using *fmincon* in MATLAB (Mathworks, 2014a-2017a). **The choice of model, unidirectional walker or biased random walk, as well as the number of pauses to include was made by calculating the Bayesian Information Criterion (BIC) [7] for each and selecting the model with the minimum BIC as the best fit. This criterion penalises for model complexity, i.e. the more parameters, the larger the BIC. Greater detail on fitting multi parameter models can be found in references 5,6 and those within.**"

We would challenge the reviewer to find an experimental single-molecule journal article with a more robust method of objectively fitting models to data than that presented by ourselves, or the references cited above.

To be more quantitative, the authors say that "Naively calculating the mean unwinding rate over the trajectory with a linear fit (Fig. 2a) we observe a larger distribution in linear unwinding rates and an increase in mean from 30pN to 40pN ($t(86) = 5.1$, $p = 2.2 \times 10^{-6}$, Welch's t-test statistics given in supplementary fig. 6). This is consistent with the trends observed for other hexameric helicases [58,59,69] and the expected force dependence from theory [69,70]." This statement is right but the authors consider only the helicase rate change between 30 and 39 pN to establish their statement, while they have measured a decrease in rate going from 20 pN to 30 pN which ought to contradict their statement completely (with a solid statistical test).

We are pleased to see the reviewer agrees that our statement is correct. The fact that there is a decrease in speed between 20 and 30 pN does not contradict our statement. The trends observed in other studies (references 58, 59, 69) are exponential in nature allowing for a flatter start to the change before a larger rate of increase for increasing force. If one looks at the data in these studies it can be seen that the data has a large amount of variability from the model with some adjacent points changing in value in the opposite direction as expected. It is an inherent aspect of experimental biophysics that the data does not perfectly fit the expected model. Indeed, we would be more worried if the data matched perfectly with an exponential over these three populations.

We have perhaps been too careful in choosing our statement, giving the impression that we have not considered this fact. We have fixed this simply in the manuscript by re-writing it to read

"Naively calculating the linear unwinding rate (Fig. 2a) we observed a wider distribution and increase in mean from 30pN to 40pN, **yet a decrease in mean and narrower distribution from 20pN to 30pN** ($t(86) = 5.1$, $p = 2.2 \times 10^{-6}$ and $t(78) = 2.9$, $p = 0.004$, respectively. Welch's t-test statistics given in Supplementary Figure 8). **Despite the small decreases in mean between 20 pN and 30 pN, these results are broadly consistent with the trends observed for other hexameric helicases [1-3] and the expected force dependence from theory [3,4]."**

In the same spirit, most helicases display Michaelis-Menten behaviour for their unwinding rate versus ATP concentration which is not the case here.

The reviewer has not fit a Michaelis-Menten type curve to our data – how can it be said objectively/scientifically whether or not the unwinding rate follows this behaviour?

We have not performed a fit to this data, attempting to do so would;

Not add anything to the narrative of the manuscript.

Not be appropriate over the range of ATP concentrations used here.

Be inappropriate because we have demonstrated the naivety of obtaining velocities in this manner.

Also we note that in our original response to the comments of reviewer 2 we state

"Kinetic parameters of replication machinery measured in bulk [38] and at the single-molecule level have been shown to reach a maximum or minimum at certain regions of force and ATP concentration [39]."

This shows that these data points do not necessarily have to follow a simple linear increasing trend. We have presented the data we observed and measured, included the uncertainties, described it, and we find that it is more complex than a simple linear increase.

For me, the authors are far too confident in their helicase traces quality which, in my opinion, are not sufficient, I recommend having better traces and better statistics before publishing.

We have demonstrated our traces are sufficient to observe helicase unwinding by both analysing the noise and recapitulating literature data with SV40 large T antigen. Indeed, the reviewer also agrees with us saying that our SV40 large T-antigen data is "perfect" and no longer has concerns with the noise.

We have provided the gold standard of controls to show the activity is true by demonstrating unwinding occurs only in the presence of both ATP and CMG.

We are unsure as to what "better" traces would be. If the reviewer is referring to demonstrating CMG unwinding a hairpin substrate as we have done for SV40 large T antigen then we must again point out that due to CMG being a tightly closed helicase it cannot bind to a piece of DNA without a free end. Thus, it is impossible to provide the kind of data the reviewer requests. Although the reviewer must be happy with the noise level of our system, as they have not mentioned this again after the previous rebuttal, we want to note that improving the precision (noise) of the traces is a practical impossibility without rebuilding the entire microscope and software, a huge task. Also the model fitting is robust to noise size as we discuss above and in the previous response to reviewers.

If by "having better statistics" the reviewer means repeating the experiments we note it will not alter the conclusions of the paper. We have already performed experimental replicates for the previous response to comments and no substantial change has occurred.

1. Ribeck, N. & Saleh, O. A. DNA Unwinding by Ring-Shaped T4 Helicase gp41 Is Hindered by Tension on the Occluded Strand. *PLoS ONE* **8**, e79237 (2013).
2. Ribeck, N., Kaplan, D. L., Bruck, I. & Saleh, O. A. DnaB Helicase Activity Is Modulated by DNA Geometry and Force. *Biophysical Journal* **99**, 2170–2179 (2010).
3. Manosas, M., Xi, X. G., Bensimon, D. & Croquette, V. Active and passive mechanisms of helicases. *Nucleic Acids Research* **38**, 5518–5526 (2010).
4. Pincus, D. L., Chakrabarti, S. & Thirumalai, D. Helicase Processivity and Not the Unwinding Velocity Exhibits Universal Increase with Force. *Biophysical Journal* **109**, 220–230 (2015).
5. Lape, R., Plested, A. J. R., Moroni, M., Colquhoun, D. & Sivilotti, L. G. The 1K276E Startle Disease Mutation Reveals Multiple Intermediate States in the Gating of Glycine Receptors. *Journal of Neuroscience* **32**, 1336–1352 (2012).
6. *Single-channel recording*. (Plenum Press, 1995).
7. Neath, A. A. & Cavanaugh, J. E. The Bayesian information criterion: background, derivation, and applications. *Wiley Interdisciplinary Reviews: Computational Statistics* **4**, 199–203 (2012).
8. Douglas, M. E., Ali, F. A., Costa, A. & Diffley, J. F. X. The mechanism of eukaryotic CMG helicase activation. *Nature* (2018). doi:10.1038/nature25787
9. Priego Moreno, S., Bailey, R., Campion, N., Herron, S. & Gambus, A. Polyubiquitylation drives replisome disassembly at the termination of DNA replication. *Science* **346**, 477–481 (2014).
10. Maric, M., Maculins, T., De Piccoli, G. & Labib, K. Cdc48 and a ubiquitin ligase drive disassembly of the CMG helicase at the end of DNA replication. *Science* **346**, 1253596 (2014).
11. Wasserman, M. R., Schauer, G. D., O'Donnell, M. E. & Liu, S. Replisome preservation by a single-stranded DNA gate in the CMG helicase. *bioRxiv* (2018). doi:10.1101/368472
12. Yardimci, H. *et al.* Bypass of a protein barrier by a replicative DNA helicase. *Nature* **492**, 205–209 (2012).
13. Abid Ali, F. *et al.* Cryo-EM structures of the eukaryotic replicative helicase bound to a translocation substrate. *Nature Communications* **7**, 10708 (2016).
14. Yeeles, J. T. P., Deegan, T. D., Janska, A., Early, A. & Diffley, J. F. X. Regulated eukaryotic DNA replication origin firing with purified proteins. *Nature* **519**, 431–435 (2015).
15. Bell, S. P. & Labib, K. Chromosome Duplication in *Saccharomyces cerevisiae*. *Genetics* **203**, 1027–1067 (2016).
16. Remus, D. *et al.* Concerted Loading of Mcm2–7 Double Hexamers around DNA during DNA Replication Origin Licensing. *Cell* **139**, 719–730 (2009).
17. Moyer, S. E., Lewis, P. W. & Botchan, M. R. Isolation of the Cdc45/Mcm2–7/GINS (CMG) complex, a candidate for the eukaryotic DNA replication fork helicase. *Proceedings of the National Academy of Sciences* **103**, 10236–10241 (2006).
18. Ilves, I., Petojevic, T., Pesavento, J. J. & Botchan, M. R. Activation of the MCM2-7 Helicase by Association with Cdc45 and GINS Proteins. *Molecular Cell* **37**, 247–258 (2010).
19. Dean, F. B., Borowiec, J. A., Eki, T. & Hurwitz, J. The simian virus 40 T antigen double hexamer assembles around the DNA at the replication origin. *J. Biol. Chem.* **267**, 14129–14137 (1992).
20. Klaue, D. DNA Unwinding by Helicases Investigated on the Single Molecule Level. (2012).
21. Berghuis, B. A., Köber, M., van Laar, T. & Dekker, N. H. High-throughput, high-force probing of DNA-protein interactions with magnetic tweezers. *Methods* **105**, 90–98 (2016).
22. Schermerhorn, K. M., Tanner, N., Kelman, Z. & Gardner, A. F. High-temperature single-molecule kinetic analysis of thermophilic archaeal MCM helicases. *Nucleic Acids Research* **44**, 8764–8771 (2016).
23. Dumont, S. *et al.* RNA translocation and unwinding mechanism of HCV NS3 helicase and its coordination by ATP. *Nature* **439**, 105–108 (2006).
24. Ibarra, B. *et al.* Proofreading dynamics of a processive DNA polymerase. *EMBO J.* **28**, 2794–2802 (2009).
25. Langston, L. D. & O'Donnell, M. E. Action of CMG with strand-specific DNA blocks supports an internal unwinding mode for the eukaryotic replicative helicase. *eLife* **6**, e23449 (2017).
26. Dyson, F. A meeting with Enrico Fermi. *Nature* **427**, 297–297 (2004).
27. Dulin, D. *et al.* Elongation-Competent Pauses Govern the Fidelity of a Viral RNA-Dependent RNA Polymerase. *Cell Reports* **10**, 983–992 (2015).

28. Dulin, D., Berghuis, B. A., Depken, M. & Dekker, N. H. Untangling reaction pathways through modern approaches to high-throughput single-molecule force-spectroscopy experiments. *Current Opinion in Structural Biology* **34**, 116–122 (2015).
29. Dulin, D. *et al.* Backtracking behavior in viral RNA-dependent RNA polymerase provides the basis for a second initiation site. *Nucleic Acids Research* gkv1098 (2015). doi:10.1093/nar/gkv1098
30. Dulin, D. *et al.* Pausing controls branching between productive and non-productive pathways during initial transcription in bacteria. *Nature Communications* **9**, (2018).
31. Lape, R., Colquhoun, D. & Sivilotti, L. G. On the nature of partial agonism in the nicotinic receptor superfamily. *Nature* **454**, 722–727 (2008).
32. Burzomato, V., Beato, M., Groot-Kormelink, P. J., Colquhoun, D. & Sivilotti, L. G. Single-channel behavior of heteromeric alpha1beta glycine receptors: an attempt to detect a conformational change before the channel opens. *J. Neurosci.* **24**, 10924–10940 (2004).
33. Colquhoun, D. & Hawkes, A. G. Relaxation and Fluctuations of Membrane Currents that Flow through Drug-Operated Channels. *Proceedings of the Royal Society B: Biological Sciences* **199**, 231–262 (1977).
34. Redner, S. *A guide to first-passage processes*. (Cambridge University Press, 2001).
35. Mejia, Y. X., Mao, H., Forde, N. R. & Bustamante, C. Thermal Probing of E. coli RNA Polymerase Off-Pathway Mechanisms. *Journal of Molecular Biology* **382**, 628–637 (2008).
36. Galburt, E. A., Grill, S. W. & Bustamante, C. Single molecule transcription elongation. *Methods* **48**, 323–332 (2009).
37. Floyd, D. L., Harrison, S. C. & van Oijen, A. M. Analysis of Kinetic Intermediates in Single-Particle Dwell-Time Distributions. *Biophysical Journal* **99**, 360–366 (2010).
38. Kang, Y.-H., Galal, W. C., Farina, A., Tappin, I. & Hurwitz, J. Properties of the human Cdc45/Mcm2-7/GINS helicase complex and its action with DNA polymerase in rolling circle DNA synthesis. *Proceedings of the National Academy of Sciences* **109**, 6042–6047 (2012).
39. Manosas, M., Spiering, M. M., Ding, F., Croquette, V. & Benkovic, S. J. Collaborative coupling between polymerase and helicase for leading-strand synthesis. *Nucleic Acids Research* **40**, 6187–6198 (2012).